# How Reliable is Your Regression Model's Uncertainty Under Real-World Distribution Shifts?

**Fredrik K. Gustafsson**
*Department of Information Technology*
*Uppsala University, Sweden*

**Martin Danelljan**
*Computer Vision Lab*
*ETH Zürich, Switzerland*

**Thomas B. Schön**
*Department of Information Technology*
*Uppsala University, Sweden*

**Reviewed on OpenReview:** *https://openreview.net/forum?id=WJt2Pc3qtI*

## Abstract

Many important computer vision applications are naturally formulated as regression problems. Within medical imaging, accurate regression models have the potential to automate various tasks, helping to lower costs and improve patient outcomes. Such safety-critical deployment does however require reliable estimation of model uncertainty, also under the wide variety of distribution shifts that might be encountered in practice. Motivated by this, we set out to investigate the reliability of regression uncertainty estimation methods under various real-world distribution shifts. To that end, we propose an extensive benchmark of 8 image-based regression datasets with different types of challenging distribution shifts. We then employ our benchmark to evaluate many of the most common uncertainty estimation methods, as well as two state-of-the-art uncertainty scores from the task of out-of-distribution detection. We find that while methods are well calibrated when there is no distribution shift, they all become highly overconfident on many of the benchmark datasets. This uncovers important limitations of current uncertainty estimation methods, and the proposed benchmark therefore serves as a challenge to the research community. We hope that our benchmark will spur more work on how to develop truly reliable regression uncertainty estimation methods. Code is available at `https://github.com/fregu856/regression_uncertainty`.

## 1 Introduction

Regression is a fundamental machine learning problem with many important computer vision applications (Rothe et al., 2016; Law & Deng, 2018; Xiao et al., 2018; Zhu et al., 2018; Shi et al., 2019; Lathuilière et al., 2019). In general, it entails predicting continuous targets $y$ from given inputs $x$. Within medical imaging, a number of tasks are naturally formulated as regression problems, including brain age estimation (Cole et al., 2017; Jónsson et al., 2019; Shi et al., 2020), prediction of cardiovascular volumes and risk factors (Xue et al., 2017; Poplin et al., 2018) and body composition analysis (Langner et al., 2020; 2021). If machine learning models could be deployed to automatically regress various such properties within real-world clinical practice, this would ultimately help lower costs and improve patient outcomes across the medical system (Topol, 2019).

Real-world deployment in medical applications, and within other safety-critical domains, does however put very high requirements on such regression models. In particular, the common approach of training a deep neural network (DNN) to directly output a predicted regression target $\hat{y} = f(x)$ is *not* sufficient, as it fails to capture any measure of uncertainty in the predictions $\hat{y}$. The model is thus unable to e.g. detect inputs $x$

which are out-of-distribution (OOD) compared to its training data. Since the predictive accuracy of DNNs typically degrades significantly on OOD inputs (Hendrycks & Dietterich, 2019; Koh et al., 2021), this could have potentially catastrophic consequences. Much research effort has therefore been invested into various approaches for training uncertainty-aware DNN models (Blundell et al., 2015; Gal, 2016; Kendall & Gal, 2017; Lakshminarayanan et al., 2017; Romano et al., 2019), to explicitly estimate the uncertainty in the predictions.

These uncertainty estimates must however be accurate and reliable. Otherwise, if the model occasionally becomes overconfident and outputs highly confident yet incorrect predictions, providing uncertainty estimates might just instill a false sense of security – arguably making the model even less suitable for safety-critical deployment. Specifically, the uncertainty estimates must be *well calibrated* and properly align with the prediction errors (Gneiting et al., 2007; Guo et al., 2017). Moreover, the uncertainty must remain well calibrated also under the wide variety of *distribution shifts* that might be encountered during practical deployment (Quionero-Candela et al., 2009; Finlayson et al., 2021). For example in medical applications, a model trained on data collected solely at a large urban hospital in the year 2020, for instance, should output well-calibrated predictions also in 2023, for patients both from urban and rural areas. While uncertainty calibration, as well as general DNN robustness (Hendrycks et al., 2021a), has been evaluated under distribution shifts for classification tasks (Ovadia et al., 2019), this important problem is not well-studied for *regression*.

Motivated by this, we set out to investigate the reliability of regression uncertainty estimation methods under various real-world distribution shifts. To that end, we propose an extensive benchmark consisting of 8 image-based regression datasets (see Figure 1) with different types of distribution shifts. These are all publicly available and relatively large-scale datasets (6 592 - 20 614 training images), yet convenient to store and train models on ($64 \times 64$ images with 1D regression targets). Four of the datasets are also taken from medical applications, with clinically relevant distribution shifts. We evaluate some of the most commonly used regression uncertainty estimation methods, including conformal prediction, quantile regression and what is often considered the state-of-the-art – ensembling (Ovadia et al., 2019; Gustafsson et al., 2020b). We also consider the approach of selective prediction (Geifman & El-Yaniv, 2017), in which the regression model can abstain from outputting predictions for certain inputs. This enables us to evaluate uncertainty scores from the rich literature on OOD detection (Salehi et al., 2022). Specifically, we evaluate two recent scores based on feature-space density (Mukhoti et al., 2021a; Pleiss et al., 2020; Sun et al., 2022; Kuan & Mueller, 2022).

In total, we evaluate 10 different methods. Among them, we find that not a single one is close to being perfectly calibrated across all datasets. While the methods are well calibrated on baseline variants with no distribution shifts, they all become highly overconfident on many of our benchmark datasets. Also the conformal prediction methods suffer from this issue, despite their commonly promoted theoretical guarantees. This highlights the importance of always being aware of underlying assumptions, assessing whether or not they are likely to hold in practice. Methods based on the state-of-the-art OOD uncertainty scores perform well *relative* to other methods, but are also overconfident in many cases – the *absolute* performance is arguably still not sufficient. Our proposed benchmark thus serves as a challenge to the research community, and we hope that it will spur more work on how to develop truly reliable regression uncertainty estimation methods.

**Summary of Contributions** We collect a set of 8 large-scale yet convenient image-based regression datasets with different types of challenging distribution shifts. Utilizing this, we propose a benchmark for testing the reliability of regression uncertainty estimation methods under real-world distribution shifts. We then employ our benchmark to evaluate many of the most common uncertainty estimation methods, as well as two state-of-the-art uncertainty scores from OOD detection. We find that all methods become highly overconfident on many of the benchmark datasets, thus uncovering limitations of current uncertainty estimation methods.

## 2 Background

In a regression problem, the task is to predict a target $y^\star \in \mathcal{Y}$ for any given input $x^\star \in \mathcal{X}$. To solve this, we are also given a train set of i.i.d. input-target pairs, $\mathcal{D}_{\text{train}} = \{(x_i, y_i)\}_{i=1}^N$, $(x_i, y_i) \sim p(x, y)$. What separates regression from classification is that the target space $\mathcal{Y}$ is continuous, $\mathcal{Y} = \mathbb{R}^K$. In this work, we only consider the 1D case, i.e. when $\mathcal{Y} = \mathbb{R}$. Moreover, the input space $\mathcal{X}$ here always corresponds to the space of images.

**Prediction Intervals, Coverage & Calibration** Given a desired miscoverage rate $\alpha$, a *prediction interval* $C_\alpha(x^\star) = [L_\alpha(x^\star),\ U_\alpha(x^\star)] \subseteq \mathbb{R}$ is a function that maps the input $x^\star$ onto an interval that should cover the true regression target $y^\star$ with probability $1 - \alpha$. For any set $\{(x_i^\star, y_i^\star)\}_{i=1}^{N^\star}$ of $N^\star$ examples, the empirical interval *coverage* is the proportion of inputs for which the prediction interval covers the corresponding target,

$$\text{Coverage}(C_\alpha) = \frac{1}{N^\star} \sum_{i=1}^{N^\star} \mathbb{I}\{y_i^\star \in C_\alpha(x_i^\star)\}. \tag{1}$$

If the coverage equals $1 - \alpha$, we say that the prediction intervals are perfectly *calibrated*. Unless stated otherwise, we set $\alpha = 0.1$ in this work. The prediction intervals should thus obtain a coverage of 90%.

## 2.1 Regression Uncertainty Estimation Methods

The most common approach to image-based regression is to train a DNN $f_\theta : \mathcal{X} \to \mathbb{R}$ that outputs a predicted target $\hat{y} = f_\theta(x)$ for any input $x$, using e.g. the L2 or L1 loss (Lathuilière et al., 2019). We are interested in methods which extend this standard direct regression approach to also provide uncertainty estimates for the predictions. Specifically, we consider methods which output a prediction interval $C_\alpha(x)$ and a predicted target $\hat{y}(x) \in C_\alpha(x)$ for each input $x$. The uncertainty in the prediction $\hat{y}(x)$ is then quantified as the length of the interval $C_\alpha(x)$ (larger interval - higher uncertainty). Some of the most commonly used regression uncertainty estimation methods fall under this category, as described in more detail below.

**Conformal Prediction** The standard regression approach can be extended by utilizing the framework of split conformal prediction (Papadopoulos et al., 2002; Papadopoulos, 2008; Romano et al., 2019). This entails splitting the train set $\{(x_i, y_i)\}_{i=1}^N$ into a proper train set $\mathcal{I}_1$ and a calibration set $\mathcal{I}_2$. The DNN $f_\theta$ is trained on $\mathcal{I}_1$, and absolute residuals $R = \{|y_i - f_\theta(x_i)| : i \in \mathcal{I}_2\}$ are computed on the calibration set $\mathcal{I}_2$. Given a new input $x^\star$, a prediction interval $C_\alpha(x^\star)$ is then constructed from the prediction $f_\theta(x^\star)$ as,

$$C_\alpha(x^\star) = [f_\theta(x^\star) - Q_{1-\alpha}(R, \mathcal{I}_2),\ f_\theta(x^\star) + Q_{1-\alpha}(R, \mathcal{I}_2)], \tag{2}$$

where $Q_{1-\alpha}(R, \mathcal{I}_2)$ is the $(1-\alpha)$-th quantile of the absolute residuals $R$. Under the assumption of exchangeably drawn train and test data, this prediction interval is guaranteed to satisfy $\mathbb{P}\{y^\star \in C_\alpha(x^\star)\} \geq 1 - \alpha$ (marginal coverage guarantee). The interval $C_\alpha(x^\star)$ has a fixed length of $2Q_{1-\alpha}(R, \mathcal{I}_2)$ for all inputs $x^\star$.

**Quantile Regression** A DNN can also be trained to directly output prediction intervals of input-dependent length, utilizing the quantile regression approach (Koenker & Bassett Jr, 1978; Romano et al., 2019; Jensen et al., 2022). This entails estimating the conditional quantile function $q^\alpha(x) = \inf\{y \in \mathbb{R} : F_{Y|X}(y|x) \geq \alpha\}$, where $F_{Y|X}$ is the conditional cumulative distribution function. Specifically, a DNN is trained to output estimates of the lower and upper quantiles $q^{\alpha_{\text{lo}}}(x)$, $q^{\alpha_{\text{up}}}(x)$ at $\alpha_{\text{lo}} = \alpha/2$ and $\alpha_{\text{up}} = 1 - \alpha/2$. Given a new input $x^\star$, a prediction interval $C_\alpha(x^\star)$ can then be directly formed as,

$$C_\alpha(x^\star) = [q_\theta^{\alpha_{\text{lo}}}(x^\star),\ q_\theta^{\alpha_{\text{up}}}(x^\star)]. \tag{3}$$

The estimated quantiles $q_\theta^{\alpha_{\text{lo}}}(x^\star)$, $q_\theta^{\alpha_{\text{up}}}(x^\star)$ can be output by a single DNN $f_\theta$, trained using the pinball loss (Steinwart & Christmann, 2011). A prediction $\hat{y}(x^\star)$ can also be extracted as the center point of $C_\alpha(x^\star)$.

**Probabilistic Regression** Another approach is to explicitly model the conditional distribution $p(y|x)$, for example using a Gaussian model $p_\theta(y|x) = \mathcal{N}\big(y; \mu_\theta(x), \sigma_\theta^2(x)\big)$ (Chua et al., 2018; Gustafsson et al., 2020a). A single DNN $f_\theta$ can be trained to output both the mean $\mu_\theta(x)$ and variance $\sigma_\theta^2(x)$ by minimizing the corresponding negative log-likelihood. For a given input $x^\star$, a prediction interval can then be constructed as,

$$C_\alpha(x^\star) = [\mu_\theta(x^\star) - \sigma_\theta(x^\star)\Phi^{-1}(1 - \alpha/2),\ \mu_\theta(x^\star) + \sigma_\theta(x^\star)\Phi^{-1}(1 - \alpha/2)], \tag{4}$$

where $\Phi$ is the CDF of the standard normal distribution. The mean $\mu_\theta(x^\star)$ is also taken as a prediction $\hat{y}$.

**Epistemic Uncertainty** From the Bayesian perspective, quantile regression and Gaussian models capture aleatoric (inherent data noise) but not epistemic uncertainty, which accounts for uncertainty in the model parameters (Gal, 2016; Kendall & Gal, 2017). This can be estimated in a principled manner via Bayesian

| | Cells-Tails | ChairAngle-Gap | AssetWealth | Ventricular Volume | BrainTumour Pixels | SkinLesion Pixels | Histology NucleiPixels | AerialBuilding Pixels |
|---|---|---|---|---|---|---|---|---|
| Train | | | | | | | | |
| | y = 100 | y = 80.5 | y = 1.594 | y = 40.38 | y = 252 | y = 500 | y = 1257 | y = 1097 |
| Test | | | | | | | | |
| | y = 198 | y = 43.8 | y = 1.314 | y = 85.93 | y = 273 | y = 516 | y = 1156 | y = 433 |

Figure 1: We propose a benchmark consisting of 8 image-based regression datasets, testing the reliability of regression uncertainty estimation methods under real-world distribution shifts. Example train (top row) and test inputs $x$, along with the corresponding ground truth targets $y$, are here shown for each of the 8 datasets.

inference, and various approximate methods have been explored (Neal, 1995; Ma et al., 2015; Blundell et al., 2015; Gal & Ghahramani, 2016). In practice, it has been shown difficult to beat the simple approach of ensembling (Lakshminarayanan et al., 2017; Ovadia et al., 2019), which entails training $M$ models $\{f_{\theta_i}\}_{i=1}^{M}$ and combining their predictions. For Gaussian models, a single mean $\hat{\mu}$ and variance $\hat{\sigma}^2$ can be computed as,

$$\hat{\mu}(x^\star) = \frac{1}{M} \sum_{i=1}^{M} \mu_{\theta_i}(x^\star), \qquad \hat{\sigma}^2(x^\star) = \frac{1}{M} \sum_{i=1}^{M} \left( \left( \hat{\mu}(x^\star) - \mu_{\theta_i}(x^\star) \right)^2 + \sigma_{\theta_i}^2(x^\star) \right), \tag{5}$$

and then plugged into (4) to construct a prediction interval $C_\alpha(x^\star)$ for a given input $x^\star$.

## 2.2 Selective Prediction

The framework of selective prediction has been applied both to classification (Herbei & Wegkamp, 2006; Geifman & El-Yaniv, 2017) and regression problems (Jiang et al., 2020). The general idea is to give a model the option to abstain from outputting predictions for some inputs. This is achieved by combining the prediction model $f_\theta$ with an uncertainty function $\kappa_f : \mathcal{X} \to \mathbb{R}$. Given an input $x^\star$, the prediction $f_\theta(x^\star)$ is output if the uncertainty $\kappa_f(x^\star) \leq \tau$ (for some user-specified threshold $\tau$), otherwise $x^\star$ is rejected and no prediction is made. The *prediction rate* is the proportion of inputs for which a prediction is output,

$$\text{Predition Rate} = \frac{1}{N^\star} \sum_{i=1}^{N^\star} \mathbb{I}\{\kappa_f(x_i^\star) \leq \tau\}. \tag{6}$$

In principle, if high uncertainty $\kappa_f(x^\star)$ corresponds to a large prediction error $|y^\star - \hat{y}(x^\star)|$ and vice versa, small errors will be achieved for all predictions which are actually output by the model. Specifically in this work, we combine selective prediction with the regression methods from Section 2.1. A prediction interval $C_\alpha(x^\star)$ and predicted target $\hat{y}(x^\star)$ are thus output if and only if (iff) $\kappa_f(x^\star) \leq \tau$. Our aim is for this to improve the calibration (interval coverage closer to $1 - \alpha$) of the output prediction intervals.

For the uncertainty function $\kappa_f(x)$, the variance $\hat{\sigma}^2(x)$ of a Gaussian ensemble (5) could be used, for example. One could also use some of the various uncertainty scores employed in the rich OOD detection literature (Salehi et al., 2022). In OOD detection, the task is to distinguish in-distribution inputs $x$, inputs which are similar to those of the train set $\{(x_i, y_i)\}_{i=1}^{N}$, from out-of-distribution inputs. A principled approach to OOD detection would be to fit a model of $p(x)$ on the train set. Inputs $x$ for which $p(x)$ is small are then deemed OOD (Serrà et al., 2020). In our considered case where inputs $x$ are images, modelling $p(x)$ can however be quite challenging. To mitigate this, a feature extractor $g : \mathcal{X} \to \mathbb{R}^{D_x}$ can be utilized, modelling $p(x)$ indirectly by fitting a simple model to the feature vectors $g(x)$. In the classification setting, Mukhoti et al. (2021a) fit a

Gaussian mixture model (GMM) to the feature vectors $\{g(x_i)\}_{i=1}^{N}$ of the train set. Given an input $x^{\star}$, it is then deemed OOD if the GMM density $\text{GMM}(g(x^{\star}))$ is small. Pleiss et al. (2020) apply this approach also to regression problems. Instead of fitting a GMM to the feature vectors and evaluating its density, (Sun et al., 2022; Kuan & Mueller, 2022) compute the distance $\text{kNN}(g(x^{\star}))$ between $g(x^{\star})$ and its $k$ nearest neighbors in the train set $\{g(x_i)\}_{i=1}^{N}$. The input $x^{\star}$ is then deemed OOD if this kNN distance is large.

## 3 Proposed Benchmark

We propose an extensive benchmark for testing the reliability of regression uncertainty estimation methods under real-world distribution shifts. The benchmark consists of 8 publicly available image-based regression datasets, which are described in detail in Section 3.1. Our complete evaluation procedure, evaluating uncertainty estimation methods mainly in terms of prediction interval coverage, is then described in Section 3.2.

### 3.1 Datasets

In an attempt to create a standard benchmark for image-based regression under distribution shifts, we collect and modify 8 datasets from the literature. Two of them contain synthetic images while the remaining six are real-world datasets, four of which are taken from medical applications. Examples from each of the 8 datasets are shown in Figure 1. We create two additional variants of each of the synthetic datasets, thus resulting in 12 datasets in total. They are all relatively large-scale (6 592 - 20 614 train images) and contain input images $x$ of size $64 \times 64$ along with 1D regression targets $y$. Descriptions of all 12 datasets are given below (further details are also provided in Appendix A), starting with the two synthetic datasets and their variants.

**Cells** Given a synthetic fluorescence microscopy image $x$, the task is to predict the number of cells $y$ in the image. We utilize the Cell-200 dataset from Ding et al. (2021; 2020), consisting of 200 000 grayscale images of size $64 \times 64$. We randomly draw 10 000 train images, 2 000 val images and 10 000 test images. Thus, there is no distribution shift between train/val and test. We therefore use this as a baseline dataset.

**Cells-Tails** A variant of CELLS with a clear distribution shift between train/val and test. For train/val, the regression targets $y$ are limited to $]50, 150]$. For test, the targets instead lie in the original range $[1, 200]$.

**Cells-Gap** Another variant of CELLS with a clear distribution shift between train/val and test. For train/val, the regression targets $y$ are limited to $[1, 50[\cup]150, 200]$. For test, the targets instead lie in the original $[1, 200]$.

**ChairAngle** Given a synthetic image $x$ of a chair, the task is to predict the yaw angle $y$ of the chair. We utilize the RC-49 dataset (Ding et al., 2021; 2020), which contains $64 \times 64$ images of different chair models rendered at yaw angles ranging from $0.1°$ to $89.9°$. We randomly split their training set and obtain 17 640 train images and 4 410 val images. By sub-sampling their test set we also get 11 225 test images. There is no clear distribution shift between train/val and test, and we therefore use this as a second baseline dataset.

**ChairAngle-Tails** A variant of CHAIRANGLE with a clear distribution shift between train/val and test. For train/val, we limit the regression targets $y$ to $]15, 75[$. For test, the targets instead lie in the original $]0, 90[$.

**ChairAngle-Gap** Another variant of CHAIRANGLE with a clear distribution shift. For train/val, the regression targets $y$ are limited to $]0, 30[\cup]60, 90[$. For test, the targets instead lie in the original $]0, 90[$.

**AssetWealth** Given a satellite image $x$, the task is to predict the asset wealth index $y$ of the region. We utilize the PovertyMap-Wilds dataset from (Koh et al., 2021), which is a variant of the dataset collected by Yeh et al. (2020). We use the training, validation-ID and test-OOD subsets of the data, giving us 9 797 train images, 1 000 val images and 3 963 test images. We resize the images from size $224 \times 224$ to $64 \times 64$. Train/val and test contain satellite images from disjoint sets of African countries, creating a distribution shift.

**VentricularVolume** Given an echocardiogram image $x$ of a human heart, the task is to predict the volume $y$ of the left ventricle. We utilize the EchoNet-Dynamic dataset (Ouyang et al., 2020), which contains 10 030 echocardiogram videos. Each video captures a complete cardiac cycle and is labeled with the left ventricular volume at two separate time points, representing end-systole (at the end of contraction) and end-diastole (just before contraction). For each video, we extract just one of these volume measurements along with the corresponding video frame. To create a clear distribution shift between train/val and test, we select the end

systolic volume (smaller volume) for train and val, but the end diastolic volume (larger volume) for test. We utilize the provided dataset splits, giving us 7 460 train images, 1 288 val images and 1 276 test images.

**BrainTumourPixels**  Given an image slice $x$ of a brain MRI scan, the task is to predict the number of pixels $y$ in the image which are labeled as brain tumour. We utilize the brain tumour dataset of the medical segmentation decathlon (Simpson et al., 2019; Antonelli et al., 2022), which is a subset of the data used in the 2016 and 2017 BraTS challenges (Bakas et al., 2018; 2017; Menze et al., 2014). The dataset contains 484 brain MRI scans with corresponding tumour segmentation masks. We split these scans 80%/20%/20% into train, val and test sets. The scans are 3D volumes of size $240 \times 240 \times 155$. We convert each scan into 155 image slices of size $240 \times 240$, and create a regression target for each image by counting the number of labeled brain tumour pixels. This gives us 20 614 train images, 6 116 val images and 6 252 test images.

**SkinLesionPixels**  Given a dermatoscopic image $x$ of a pigmented skin lesion, the task is to predict the number of pixels $y$ in the image which are labeled as lesion. We utilize the HAM10000 dataset by Tschandl et al. (2018), which contains 10 015 dermatoscopic images with corresponding skin lesion segmentation masks. HAM10000 consists of four different sub-datasets, three of which were collected in Austria, while the fourth sub-dataset was collected in Australia. To create a clear distribution shift between train/val and test, we use the Australian sub-dataset as our test set. After randomly splitting the remaining images 85%/15% into train and val sets, we obtain 6 592 train images, 1 164 val images and 2 259 test images.

**HistologyNucleiPixels**  Given an H&E stained histology image $x$, the task is to predict the number of pixels $y$ in the image which are labeled as nuclei. We utilize the CoNSeP dataset by Graham et al. (2019), along with the pre-processed versions they provide of the Kumar (Kumar et al., 2017) and TNBC (Naylor et al., 2018) datasets. The datasets contain large H&E stained image tiles, with corresponding nuclear segmentation masks. The three datasets were collected at different hospitals/institutions, with differing procedures for specimen preservation and staining. By using CoNSeP and Kumar for train/val and TNBC for test, we thus obtain a clear distribution shift. From the large image tiles, we extract $64 \times 64$ patches via regular gridding, and create a regression target for each image patch by counting the number of labeled nuclei pixels. In the end, we obtain 10 808 train images, 2 702 val images and 2 267 test images.

**AerialBuildingPixels**  Given an aerial image $x$, the task is to predict the number of pixels $y$ in the image which are labeled as building. We utilize the Inria aerial image labeling dataset (Maggiori et al., 2017), which contains 180 large aerial images with corresponding building segmentation masks. The images are captured at five different geographical locations. We use the images from two densely populated American cities for train/val, and the images from a more rural European area for test, thus obtaining a clear distribution shift. After preprocessing, we obtain 11 184 train images, 2 797 val images and 3 890 test images.

Constructing these custom datasets is one of our main contributions. This is what enables us to propose an extensive benchmark of large-scale yet convenient datasets (which are all publicly available), containing different types of challenging distribution shifts, specifically for image-based regression.

### 3.2  Evaluation

We propose to evaluate regression uncertainty estimation methods mainly in terms of prediction interval coverage (1): if a method outputs a prediction $\hat{y}(x)$ and a 90% prediction interval $C_{0.1}(x)$ for each input $x$, does the method actually achieve 90% coverage on the *test* set? I.e., are the prediction intervals calibrated?

**Motivation**  Regression uncertainty estimation methods can be evaluated using various approaches. One alternative is sparsification (Ilg et al., 2018), measuring how well the uncertainty can be used to sort predictions from worst to best. Perfect sparsification can however be achieved even if the absolute scale of the uncertainty is consistently underestimated. Therefore, a lot of previous work has instead evaluated methods in terms of calibration (Gneiting et al., 2007; Guo et al., 2017). Specifically for regression, a common form of calibration is based on quantiles (Kuleshov et al., 2018; Cui et al., 2020; Chung et al., 2021). Essentially, a model is there said to be calibrated if the interval coverage (1) equals $1 - \alpha$ for *all* miscoverage rates $\alpha \in ]0, 1[$. This is measured by the expected calibration error, $\text{ECE} = \frac{1}{m} \sum_{j=1}^{m} \left| \text{Coverage}(C_{\alpha_j}) - (1 - \alpha_j) \right|$, $\alpha_j \sim \text{U}(0, 1)$. Our proposed evaluation metric is thus a special case of this approach, considering just one specific miscoverage rate $\alpha = 0.1$. We argue that this results in a simpler and more interpretable metric, which also is motivated

by how prediction intervals actually are used in real-world applications. There, one particular value of $\alpha$ is selected ($\alpha = 0.1$ is a common choice), and the corresponding intervals $C_\alpha$ are then expected to achieve a coverage of $1 - \alpha$ on unseen test data. Recent alternative calibration metrics directly measure how well the uncertainty aligns with the prediction errors (Levi et al., 2022; Pickering & Sapsis, 2022). While these enable relative comparisons of different methods, they are not easily interpretable in terms of absolute performance.

**Implementation Details** For each dataset and method, we first train a DNN on the train set $\mathcal{D}_\text{train}$. Then, we run the method on the *val* set $\mathcal{D}_\text{val}$, resulting in a prediction interval $C_\alpha(x) = [L_\alpha(x), \, U_\alpha(x)]$ for each input $x$ ($\alpha = 0.1$). Importantly, we then calibrate these prediction intervals on *val* using the procedure in (Romano et al., 2019). This gives calibrated prediction intervals $\tilde{C}_\alpha(x)$, for which the interval coverage on *val* exactly equals $1 - \alpha$. Specifically, $\tilde{C}_\alpha(x)$ is constructed from the original interval $C_\alpha(x) = [L_\alpha(x), \, U_\alpha(x)]$,

$$\tilde{C}_\alpha(x) = [L_\alpha(x) - Q_{1-\alpha}(E, \mathcal{D}_\text{val}), \, U_\alpha(x) + Q_{1-\alpha}(E, \mathcal{D}_\text{val})], \tag{7}$$

where $E = \{\max(L_\alpha(x_i) - y_i, \, y_i - U_\alpha(x_i)) : i \in \mathcal{D}_\text{val}\}$ are conformity scores computed on $\mathcal{D}_\text{val}$, and $Q_{1-\alpha}(E, \mathcal{D}_\text{val})$ is the $(1 - \alpha)$-th quantile of these scores. Finally, we then run the method on the test set $\mathcal{D}_\text{test}$, outputting a calibrated prediction interval $\tilde{C}_\alpha(x^\star)$ for each input $x^\star$. Ideally, the interval coverage of $\tilde{C}_\alpha(x^\star)$ does not change from val to test, i.e. Coverage($\tilde{C}_\alpha$) $= 1 - \alpha$ should be true also on test. If Coverage($\tilde{C}_\alpha$) $\neq 1 - \alpha$, a conservative method ($> 1 - \alpha$) is preferred compared to an *overconfident* method ($< 1 - \alpha$). For methods based on selective prediction (Section 2.2), the only difference is that prediction intervals $\tilde{C}_\alpha(x^\star)$ are output only for some test inputs $x^\star$ (iff $\kappa_f(x^\star) \leq \tau$). The interval coverage is thus computed only on this subset of test. This is similar to the notion of "selective calibration" discussed for the classification setting by Fisch et al. (2022). We set $\alpha = 0.1$ since this is a commonly used miscoverage rate in practice.

**Secondary Metrics** We also evaluate methods in terms of mean absolute error (MAE) and average interval length on the *val* set. This measures the quality of the prediction $\hat{y}(x)$ and the prediction interval $C_\alpha(x)$, respectively (Gneiting et al., 2007). The average interval length is a natural secondary metric, since a method that achieves a coverage close to $1 - \alpha$ but outputs extremely large intervals for all inputs $x$, not would be particularly useful in practice. Moreover, if two different methods both are perfectly calibrated, i.e. Coverage($C_\alpha$) $= 1 - \alpha$, the method producing smaller prediction intervals would be preferred. For methods based on selective prediction (which output predictions only for certain inputs $x$), the proportion of inputs for which a prediction actually is output is another natural secondary metric. For these methods, we thus also evaluate in terms of the prediction rate (6) on test. If a coverage close to $1 - \alpha$ is achieved with a very low prediction rate, the method might still not be practically useful in certain applications. For two perfectly calibrated methods, one with a higher prediction rate would be preferred.

## 4 Evaluated Methods

We evaluate five common regression uncertainty estimation methods from Section 2.1, which all output a prediction interval $C_\alpha(x)$ and a predicted target $\hat{y}(x) \in C_\alpha(x)$ for each input $x$. Two of these we also combine with selective prediction (Section 2.2), utilizing four different uncertainty functions $\kappa_f(x)$. In total, we evaluate 10 different methods. For all methods, we train models based on a ResNet34 (He et al., 2016) backbone DNN. This architecture is chosen because of its simplicity and widespread use. The ResNet takes an image $x$ as input and outputs a feature vector $g(x) \in \mathbb{R}^{512}$. Below we specify and provide implementation details for each of the 10 evaluated methods, while we refer back to Section 2 for more general descriptions.

**Conformal Prediction** We create a standard direct regression model by feeding the ResNet feature vector $g(x)$ into a network head of two fully-connected layers, outputting a scalar prediction $f_\theta(x)$. The model is trained using the L2 loss. We then utilize conformal prediction to create prediction intervals $C_\alpha(x)$ according to (2). Instead of splitting the train images into $\mathcal{I}_1$ and $\mathcal{I}_2$, we use the val images as the calibration set $\mathcal{I}_2$.

**Ensemble** We train an ensemble $\{f_{\theta_1}, \ldots, f_{\theta_M}\}$ of $M = 5$ direct regression models and compute the ensemble mean $\hat{\mu}(x) = \frac{1}{M} \sum_{i=1}^{M} f_{\theta_i}(x)$ and ensemble variance $\hat{\sigma}^2(x) = \frac{1}{M} \sum_{i=1}^{M} (\hat{\mu}(x) - f_{\theta_i}(x))^2$. By inserting these into equation 4, prediction intervals $C_\alpha(x)$ of input-dependent length are then constructed.

**Gaussian**  We create a Gaussian model $p_\theta(y|x) = \mathcal{N}\big(y; \mu_\theta(x), \sigma_\theta^2(x)\big)$ by feeding the ResNet feature vector $g(x)$ into two separate network heads of two fully-connected layers. These output the mean $\mu_\theta(x)$ and variance $\sigma_\theta^2(x)$, respectively. Prediction intervals $C_\alpha(x)$ are then constructed according to (4).

**Gaussian Ensemble**  We train an ensemble of $M = 5$ Gaussian models, compute a single mean $\hat{\mu}(x)$ and variance $\hat{\sigma}^2(x)$ according to (5), and plug these into (4) to construct prediction intervals $C_\alpha(x)$.

**Quantile Regression**  We create a quantile regression model by feeding the ResNet feature vector $g(x)$ into two separate network heads. These output the quantiles $q_\theta^{\alpha_{\text{lo}}}(x), q_\theta^{\alpha_{\text{up}}}(x)$, directly forming prediction intervals $C_\alpha(x)$ according to (3). The model is trained by minimizing the average pinball loss of $q_\theta^{\alpha_{\text{lo}}}(x)$ and $q_\theta^{\alpha_{\text{up}}}(x)$.

**Gaussian + Selective GMM**  We combine the GAUSSIAN method with a selective prediction mechanism. After training a Gaussian model, we run it on each image in train to extract ResNet feature vectors $\{g(x_i)\}_{i=1}^N$. We then utilize scikit-learn (Pedregosa et al., 2011) to fit a GMM (4 components, full covariance) to these train feature vectors. To compute an uncertainty score $\kappa_f(x)$ for a given input $x$, we extract $g(x)$ and evaluate its likelihood according to the fitted GMM, $\kappa_f(x) = -\text{GMM}\big(g(x)\big)$. The prediction $\mu_\theta(x)$ and corresponding prediction interval $C_\alpha(x)$ of the Gaussian model are then output iff $\kappa_f(x) \leq \tau$. To set the user-specified threshold $\tau$, we compute $\kappa_f(x)$ on all images in val and pick the 95% quantile. This choice of $\tau$ is motivated by the commonly reported FPR95 OOD detection metric, but $\tau$ could be set using other approaches.

**Gaussian + Selective kNN**  Identical to GAUSSIAN + SELECTIVE GMM, but $\kappa_f(x) = \text{kNN}\big(g(x)\big)$. Specifically, the uncertainty score $\kappa_f(x)$ is computed by extracting $g(x)$ and computing the average distance to its $k = 10$ nearest neighbors among the train feature vectors $\{g(x_i)\}_{i=1}^N$. Following Kuan & Mueller (2022), we utilize the Annoy[1] approximate neighbors library, with cosine similarity as the distance metric.

**Gaussian + Selective Variance**  Identical to GAUSSIAN + SELECTIVE GMM, but $\kappa_f(x) = \sigma_\theta^2(x)$ (the variance of the Gaussian model). This is used as a simple baseline for the two previous methods.

**Gaussian Ensemble + Selective GMM**  We combine GAUSSIAN ENSEMBLE with a selective prediction mechanism. After training an ensemble of $M = 5$ Gaussian models, we run each model on each image in train to extract $M$ sets of ResNet feature vectors. For each model, we then fit a GMM to its set of train feature vectors. I.e., we fit $M$ different GMMs. To compute an uncertainty score $\kappa_f(x)$ for a given input $x$, we extract a feature vector and evaluate its likelihood according to the corresponding GMM, for each of the $M$ models. Finally, we compute the mean of the GMM likelihoods, $\kappa_f(x) = \frac{1}{M}\sum_{i=1}^M -\text{GMM}_i\big(g_i(x)\big)$.

**Gaussian Ensemble + Selective Ensemble Variance**  Identical to GAUSSIAN ENSEMBLE + SELECTIVE GMM, but $\kappa_f(x) = \frac{1}{M}\sum_{i=1}^M \big(\hat{\mu}(x) - \mu_{\theta_i}(x)\big)^2$, where $\hat{\mu}(x) = \frac{1}{M}\sum_{i=1}^M \mu_{\theta_i}(x)$. Hence, the variance of the ensemble means is used as the uncertainty score. This constitutes a simple baseline for the previous method.

All models are trained for 75 epochs using the ADAM optimizer (Kingma & Ba, 2014). The same hyperparameters are used for all datasets, and neither the training procedure nor the models are specifically tuned for any particular dataset. All experiments are implemented using PyTorch (Paszke et al., 2019), and our complete implementation is made publicly available. All models were trained on individual NVIDIA TITAN Xp GPUs. On one such GPU, training 20 models on one dataset took approximately 24 hours. We chose to train all models based on a ResNet34 backbone because it is widely used across various applications, yet simple to implement and quite computationally inexpensive. The proposed benchmark and the evaluated uncertainty estimation methods are however entirely independent of this specific choice of DNN backbone architecture. Exploring the use of other more powerful models and evaluating how this affects the reliability of uncertainty estimation methods is an interesting direction which we leave for future work.

## 5   Related Work

Out-of-distribution robustness of DNNs is an active area of research (Hendrycks & Dietterich, 2019; Hendrycks et al., 2021b;a; Izmailov et al., 2021; Yao et al., 2022; Schwinn et al., 2022; Wiles et al., 2022; Zhang et al., 2022). All these previous works do however focus exclusively on *classification* tasks. Moreover, they consider no uncertainty measures but instead evaluate only in terms of accuracy. While (Zaidi et al., 2021; Hendrycks

---

[1] https://github.com/spotify/annoy

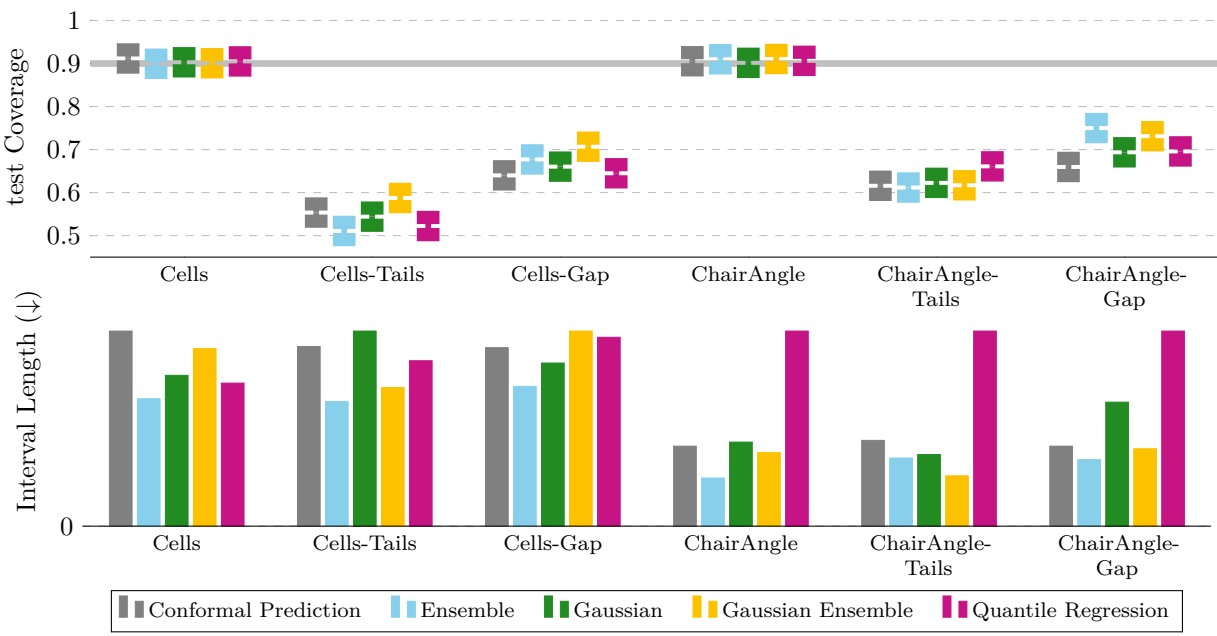

Figure 2: Results for the five common regression uncertainty estimation methods (which output predictions and corresponding 90% prediction intervals for all inputs), on the six *synthetic* datasets. **Top:** Results in terms of our main metric test coverage. A perfectly calibrated method would achieve a test coverage of exactly 90%, as indicated by the solid line. **Bottom:** Results in terms of average val interval length.

et al., 2022) evaluate uncertainty calibration, they also just consider the classification setting. In contrast, evaluation of uncertainty estimation methods is our main focus, and we do this specifically for *regression*.

The main sources of inspiration for our work are (Koh et al., 2021) and (Ovadia et al., 2019). While Koh et al. (2021) propose an extensive benchmark with various real-world distribution shifts, it only contains a single regression dataset. Moreover, methods are evaluated solely in terms of predictive performance. Ovadia et al. (2019) perform a comprehensive evaluation of uncertainty estimation methods under distribution shifts, but only consider classification tasks. Inspired by this, we thus propose our benchmark for evaluating reliability of *uncertainty estimation* methods under *real-world distribution shifts* in the *regression* setting. Most similar to our work is that of Malinin et al. (2021). However, their benchmark contains just two regression datasets (tabular weather prediction and a complex vehicle motion prediction task), they only evaluate ensemble-based uncertainty estimation methods, and these methods are not evaluated in terms of calibration.

## 6 Results

We evaluate the 10 methods specified in Section 4 on all 12 datasets from Section 3.1, according to the evaluation procedure described in Section 3.2. For each method we train 20 models, randomly select 5 of them for evaluation and report the averaged metrics. For the ensemble methods, we construct an ensemble by randomly selecting $M = 5$ out of the 20 trained models, evaluate the ensemble and then repeat this 5 times in total. To ensure that the results do not depend on our specific choice of $\alpha = 0.1$, we also evaluate methods with two alternative miscoverage rates. While the main paper only contains results for $\alpha = 0.1$, we repeat most of the evaluation for $\alpha = 0.2$ and $\alpha = 0.05$ in Appendix B, observing very similar trends overall.

### 6.1 Common Uncertainty Estimation Methods

We start by evaluating the first five methods from Section 4, those which output predictions and corresponding 90% prediction intervals for all inputs. The results in terms of our main metric test coverage are presented in the upper part of Figure 2 for the synthetic datasets, and in Figure 3 for the six real-world datasets. In the

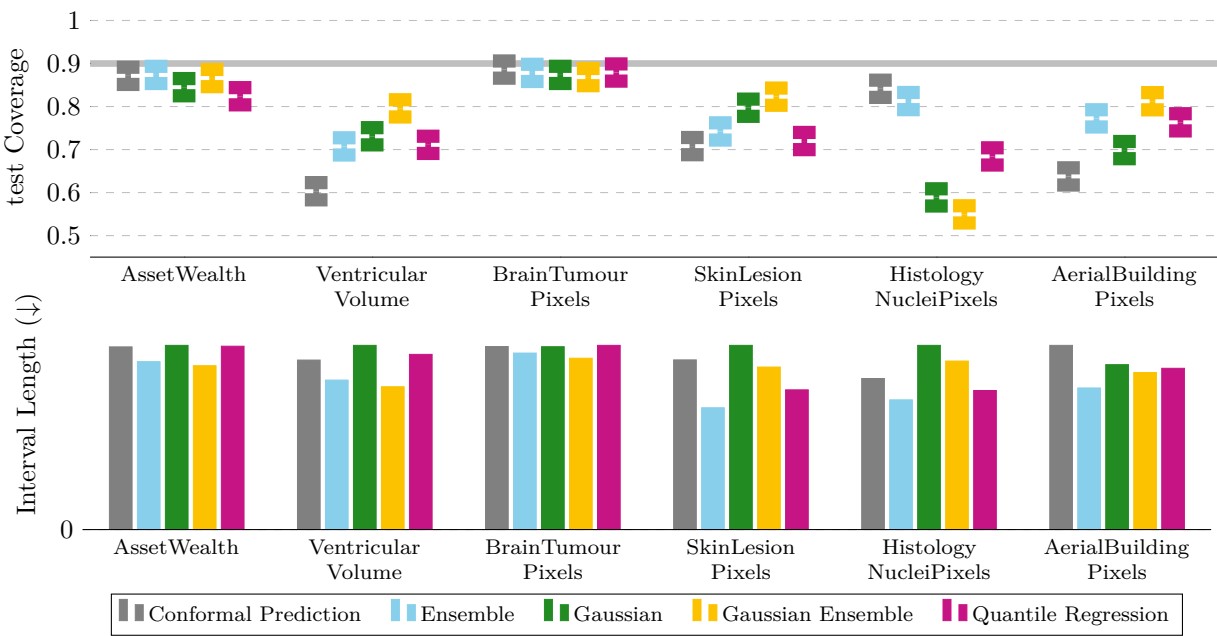

Figure 3: Results for the five common regression uncertainty estimation methods (which output predictions and corresponding 90% prediction intervals for all inputs), on the six *real-world* datasets. **Top:** Results in terms of our main metric test coverage. **Bottom:** Results in terms of average val interval length.

lower parts of Figure 2 & 3, results in terms of average val interval length are presented. The complete results, including our other secondary metric val MAE, are provided in Table A1 - Table A12 in the appendix. Please note that, because we utilize a new benchmark consisting of custom datasets, we are not able to directly compare the MAE of our models with that of any previous work from the literature.

In Figure 2, the test coverage results on the first synthetic dataset Cells are found in the upper-left. As there is no distribution shift between train/val and test for this dataset, we use it as a baseline. We observe that all five methods have almost perfectly calibrated prediction intervals, i.e. they all obtain a test coverage very close to 90%. This is exactly the desired behaviour. On Cells-Tails however, on which we introduced a clear distribution shift, we observe in Figure 2 that the test coverage drops dramatically from the desired 90% for all five methods. Even the state-of-the-art uncertainty estimation method Gaussian Ensemble here becomes highly overconfident, as its test coverage drops down to ≈ 59%. On Cells-Gap, the test coverages are slightly closer to 90%, but all five methods are still highly overconfident. On the other synthetic dataset ChairAngle, we observe in Figure 2 that all five methods have almost perfectly calibrated prediction intervals. However, as we introduce clear distribution shifts on ChairAngle-Tails and ChairAngle-Gap, we can observe that the test coverage once again drops dramatically for all methods.

The results on the six real-world datasets are found in Figure 3. In the upper part, we observe that all five methods have quite well-calibrated prediction intervals on AssetWealth and BrainTumourPixels, even though they all are consistently somewhat overconfident (test coverages of 82%-89%). On the four remaining datasets, the methods are in general more significantly overconfident. On VentricularVolume, we observe test coverages of 60%-80% for all methods, and on SkinLesionPixels the very best coverage is ≈ 82%. On HistologyNucleiPixels, most methods only obtain test coverages of 55%-70%, and on AerialBuildingPixels the very best coverage is ≈ 81%. In fact, not a single method actually reaches the desired 90% test coverage on any of these real-world datasets.

In terms of average val interval length, we observe in the lower parts of Figure 2 & 3 that Ensemble consistently produces smaller prediction intervals than Conformal Prediction. Moreover, the intervals of Gaussian Ensemble are usually smaller than those of Gaussian. When comparing the interval lengths of Quantile Regression and Gaussian, we observe no clear trend that is consistent across all datasets.

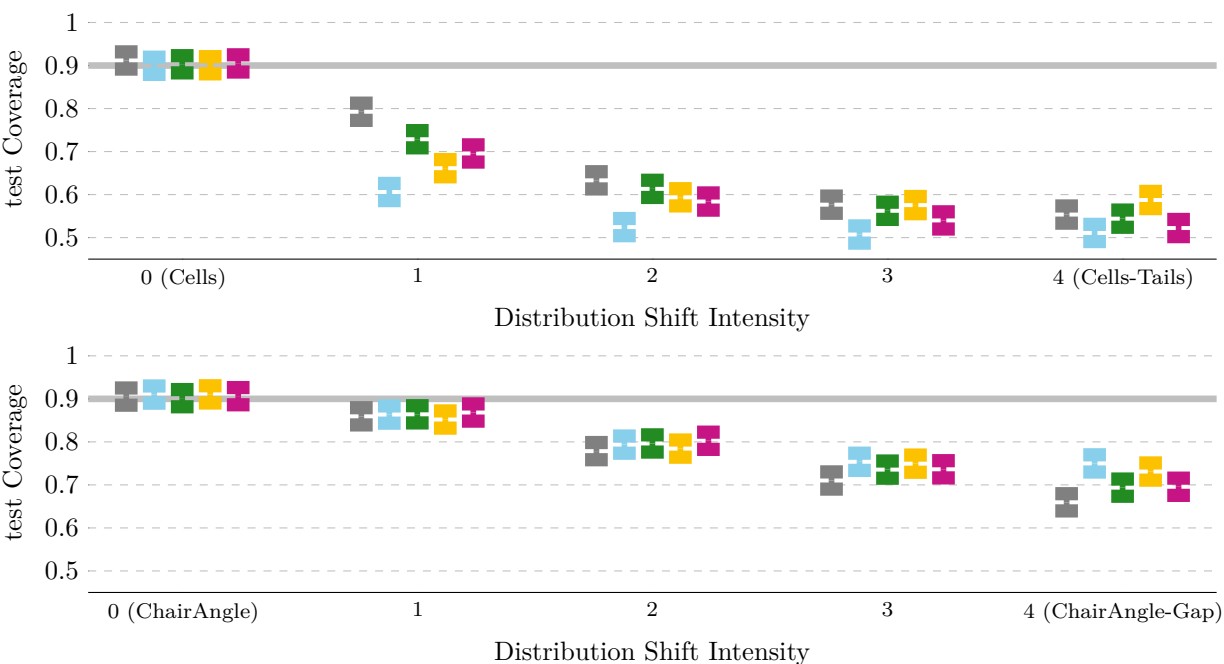

Figure 4: Test coverage results for the five common regression uncertainty estimation methods, on synthetic datasets with *increasing degrees of distribution shifts*. **Top:** From CELLS (no distribution shift) to CELLS-TAILS (maximum distribution shift). **Bottom:** From CHAIRANGLE to CHAIRANGLE-GAP.

Since the average interval lengths vary a lot between different datasets, Figure 2 & 3 only show relative comparisons of the methods. For absolute numerical scales, see Table A1 - Table A12.

To further study how the test coverage performance is affected by distribution shifts, we also apply the five methods to three additional variants of the CELLS dataset. CELLS has no difference in regression target range between train/val and test, whereas for CELLS-TAILS the target range is $]50, 150]$ for train/val and $[1, 200]$ for test. By creating three variants with intermediate target ranges, we thus obtain a sequence of five datasets with increasing degrees of distribution shifts, starting with CELLS (no distribution shift) and ending with CELLS-TAILS (maximum distribution shift). The test coverage results on this sequence of datasets are presented in the upper part of Figure 4. We observe that as the degree of distribution shift is increased step-by-step, the test coverage also drops accordingly. The lower part of Figure 4 presents the results of a similar experiment, in which we construct a sequence of five datasets starting with CHAIRANGLE (no distribution shift) and ending with CHAIRANGLE-GAP (maximum distribution shift). Also in this case, we observe that the test coverage drops step-by-step along with the increased degree of distribution shift.

A study of the relative performance of the five methods on the real-world datasets, in terms of all three metrics (test coverage, average val interval length, val MAE), is finally presented in Figure A1 - Figure A3 in the appendix. One can clearly observe that ENSEMBLE and GAUSSIAN ENSEMBLE achieve the best performance, thus indicating that ensembling multiple models indeed helps to improve the performance.

## 6.2 Selective Prediction Methods

Next, we evaluate the methods with an added selective prediction mechanism. We start with the three methods based on GAUSSIAN. The results in terms of test coverage and test prediction rate are available in Figure 5 for the synthetic datasets, and in Figure 6 for the six real-world datasets. While a complete evaluation of these methods also should include the average val interval length, we note that the selective prediction mechanism does not modify the intervals of the underlying GAUSSIAN method (which already has been evaluated in terms of interval length in Section 6.1). Here, we therefore focus on the test coverage and test prediction rate. Complete numerical results are provided in Table A1 - Table A12 in the appendix.

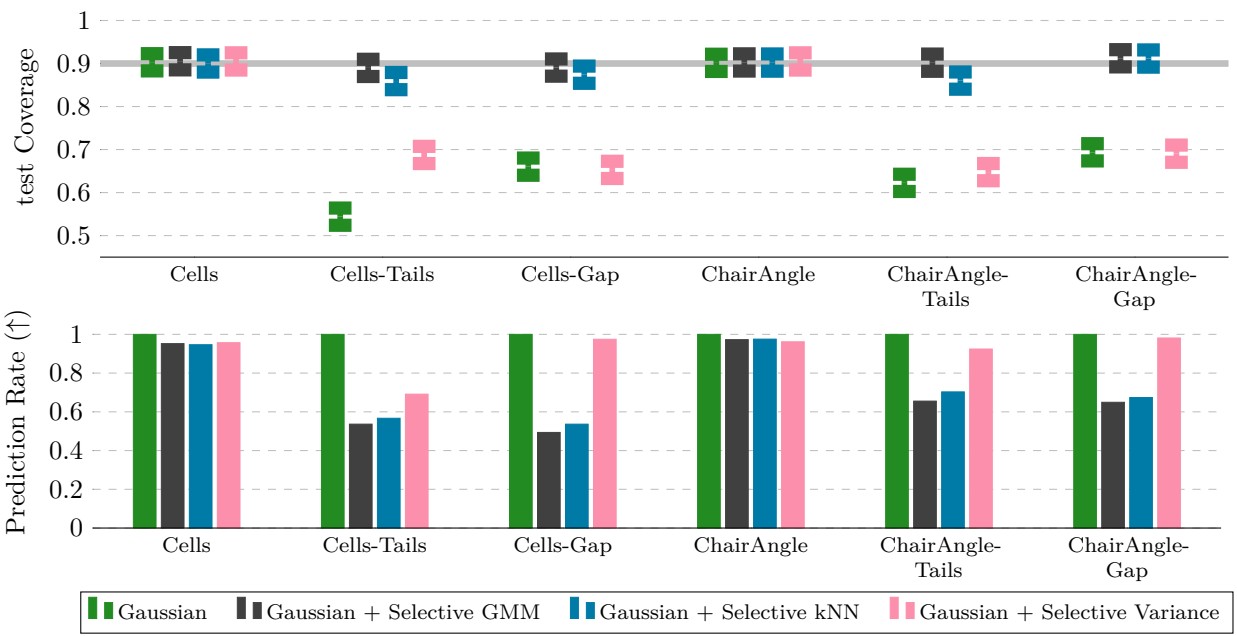

Figure 5: Results for the three selective prediction methods based on GAUSSIAN, on the six *synthetic* datasets. **Top:** Results in terms of our main metric test coverage. **Bottom:** Results in terms of test prediction rate (the proportion of test inputs for which a prediction actually is output).

In the upper part of Figure 5, we observe that selective prediction based on feature-space density significantly improves the test coverage of GAUSSIAN on the synthetic datasets. While GAUSSIAN has well-calibrated prediction intervals only on CELLS and CHAIRANGLE, which are baseline datasets without any distribution shift, GAUSSIAN + SELECTIVE GMM is almost perfectly calibrated across all six datasets. On CELLS-TAILS, for example, it improves the test coverage from $\approx 54\%$ up to $\approx 89\%$. GAUSSIAN + SELECTIVE kNN also significantly improves the test coverages, but not quite to the same extent. In the lower part of Figure 5, we can observe that when GAUSSIAN + SELECTIVE GMM significantly improves the test coverage, there is also a clear drop in its test prediction rate. For example, the prediction rate drops from $\approx 0.95$ on CELLS down to $\approx 0.54$ on CELLS-TAILS. By rejecting nearly 50% of all inputs as OOD in this case, GAUSSIAN + SELECTIVE GMM can thus remain well-calibrated on the subset of test it actually outputs predictions for. In Figure 5, we also observe that GAUSSIAN + SELECTIVE VARIANCE only marginally improves the test coverage.

While GAUSSIAN + SELECTIVE GMM significantly improves the test coverage of GAUSSIAN and has well-calibrated prediction intervals across the synthetic datasets, we observe in Figure 6 that this is not true for the six real-world datasets. GAUSSIAN + SELECTIVE GMM does consistently improve the test coverage, but only marginally, and it still suffers from significant overconfidence in many cases. On VENTRICULARVOLUME, for example, the test prediction rate of GAUSSIAN + SELECTIVE GMM is as low as $\approx 0.71$, but the test coverage only improves from $\approx 73\%$ to $\approx 75\%$ compared to GAUSSIAN.

For the two methods based on GAUSSIAN ENSEMBLE, the results are presented in Figure A4 & A5 in the appendix. Overall, we observe very similar trends. GAUSSIAN ENSEMBLE + SELECTIVE GMM significantly improves the test coverage of GAUSSIAN ENSEMBLE and is almost perfectly calibrated across the synthetic datasets. However, when it comes to the real-world datasets, it often remains significantly overconfident.

Finally, Figure A6 presents a relative comparison of the five selective prediction methods across all 12 datasets, in terms of average test coverage error (absolute difference between empirical and expected interval coverage) and average test prediction rate. We observe that GAUSSIAN ENSEMBLE + SELECTIVE GMM achieves the best test coverage error, but also has the lowest test prediction rate. In fact, each improvement in terms of test coverage error also corresponds to a decrease in test prediction rate for these five methods, meaning that there seems to be an inherent trade-off between the two metrics.

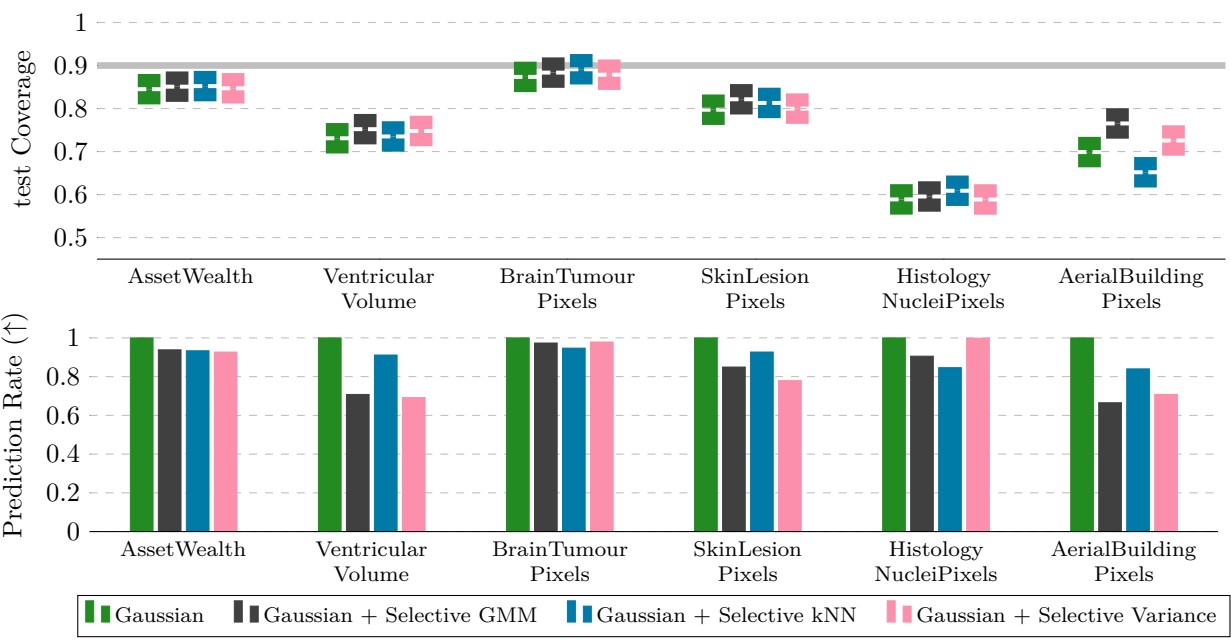

Figure 6: Results for the three selective prediction methods based on Gaussian, on the six *real-world* datasets. **Top:** Results in terms of test coverage. **Bottom:** Results in terms of test prediction rate.

## 7 Discussion

Let us now analyze the results from Section 6 in more detail, and discuss what we consider the most important findings and insights. First of all, we can observe that among the 10 considered methods, not a single one was close to producing perfectly calibrated prediction intervals across all 12 datasets. We thus conclude that our proposed benchmark indeed is challenging and interesting. Moreover, the results in Figure 2 & 3 demonstrate that while common uncertainty estimation methods are well calibrated when there is no distribution shift (Cells and ChairAngle), they can all break down and become highly overconfident in many realistic scenarios. This highlights the importance of employing sufficiently realistic and thus challenging benchmarks when evaluating uncertainty estimation methods. Otherwise, we might be lead to believe that methods will be more reliable during practical deployment than they actually are.

**Coverage Guarantees Might Instill a False Sense of Security**  We also want to emphasize that Conformal Prediction and Quantile Regression[2] have theoretical coverage guarantees, but still are observed to become highly overconfident for many datasets in Figure 2 & 3. Since the guarantees depend on the assumption that all data points are exchangeable (true for i.i.d. data, for instance), which generally does not hold under distribution shifts, these results should actually not be surprising. The results are however a good reminder that we always need to be aware of the underlying assumptions, and whether or not they are likely to hold in common practical applications. Otherwise, such theoretical guarantees might just instill a false sense of security, making us trust methods more than we actually should.

**Clear Performance Differences between Synthetic and Real-World Datasets**  We find it interesting that selective prediction based on feature-space density, in particular Gaussian + Selective GMM, works almost perfectly in terms of test coverage across the synthetic datasets (Figure 5), but fails to give significant improvements on the real-world datasets (Figure 6). The results on VentricularVolume are particularly interesting, as the prediction rate drops quite a lot without significantly improving the test coverage. This means that while a relatively large proportion of the test inputs are deemed OOD and thus rejected by the method, the test coverage is barely improved. On the synthetic datasets, there is a corresponding improvement in test coverage whenever the prediction rate drops significantly (Figure 5). It is not clear

---

[2]Since all prediction intervals are calibrated on val, we are using *Conformalized Quantile Regression* (Romano et al., 2019).

why such an obvious performance difference between synthetic and real-world datasets is observed. One possible explanation is that real-world data requires better models for $p(x)$, i.e. that the relatively simple approaches based on feature-space density not are sufficient. Properly explaining this performance difference is an important problem, but we will here leave this as an interesting direction for future work.

**OOD Uncertainty Scores Perform Well, but Not Well Enough**  Comparing the selective prediction methods, we observe that GAUSSIAN + SELECTIVE GMM consistently outperforms GAUSSIAN + SELECTIVE VARIANCE (Figure 5 & 6) and that GAUSSIAN ENSEMBLE + SELECTIVE GMM outperforms GAUSSIAN ENSEMBLE + SELECTIVE ENSEMBLE VARIANCE in most cases (Figure A4 & A5). Relative to common uncertainty estimation baselines, methods based on feature-space density thus achieve very strong performance. This is in line with the state-of-the-art OOD detection performance that has been demonstrated recently. In our results, we can however clearly observe that while feature-space methods perform well *relative* to common baselines, the resulting selective prediction methods are still overconfident in many cases – the *absolute* performance is still far from perfect. Using our benchmark, we are thus able to not only compare the relative performance of different OOD uncertainty scores, but also evaluate their performance in an absolute sense.

**Performance Differences among Real-World Datasets are Mostly Logical**  When we compare the performance on the different real-world datasets in Figure 3, all methods are relatively well-calibrated on BRAINTUMOURPIXELS and ASSETWEALTH. For BRAINTUMOURPIXELS, the train, val and test splits were created by randomly splitting the original set of MRI scans. The distribution shift between train/val and test is thus also fairly limited. For ASSETWEALTH (satellite images from different African countries), the shift is likely quite limited at least compared to AERIALBUILDINGPIXELS (satellite images from two different continents). Finally, the results for HISTOLOGYNUCLEIPIXELS are interesting, as this is the only dataset where CONFORMAL PREDICTION clearly obtains the best test coverage. It is not clear why the methods which output prediction intervals of input-dependent length struggle on this particular dataset.

Finally, we should emphasize that while test coverage is our main metric, this by itself is not sufficient for a method to be said to "perform well" in a general sense. For example, a perfectly calibrated method with a low test prediction rate might not be particularly useful in practice. While even very low test prediction rates likely would be tolerated in many medical applications and other safety-critical domains (as long as the method stays perfectly calibrated), one can imagine more low-risk settings where this calibration versus prediction rate trade-off is a lot less clear. Since not a single one of the evaluated methods was close to being perfectly calibrated across all 12 datasets, we did however mainly focus on analyzing the test coverage in this paper. If multiple methods had performed well in terms of test coverage, a more detailed analysis and discussion of the secondary metrics performance would have been necessary.

**The main actionable takeaways from our work can be summarized as follows:**

- All methods are well calibrated on baseline datasets with no distribution shift, but become highly overconfident in many realistic scenarios. Uncertainty estimation methods must therefore be evaluated using sufficiently challenging benchmarks. Otherwise, one might be lead to believe that methods will be more reliable during real-world deployment than they actually are.

- Conformal prediction methods have commonly promoted theoretical coverage guarantees, but these depend on an assumption that is unlikely to hold in many practical applications. Consequently, also these methods can become highly overconfident in realistic scenarios. If the underlying assumptions are not examined critically by practitioners, such theoretical guarantees risk instilling a false sense of security – making these models even less suitable for safety-critical deployment.

- The clear performance difference between synthetic and real-world datasets observed for selective prediction methods based on feature-space density is a very interesting direction for future work. If the reasons for this performance gap can be understood, an uncertainty estimation method that stays well calibrated across all datasets could potentially be developed.

- Selective prediction methods based on feature-space density perform well relative to other methods (as expected based on their state-of-the-art OOD detection performance), but are also overconfident in many cases. Only comparing the relative performance of different methods is therefore not sufficient.

> To track if actual progress is being made towards the ultimate goal of truly reliable uncertainty estimation methods, benchmarks must also evaluate method performance in an absolute sense.

## 8 Conclusion

We proposed an extensive benchmark for testing the reliability of regression uncertainty estimation methods under real-world distribution shifts. The benchmark consists of 8 publicly available image-based regression datasets with different types of challenging distribution shifts. We employed our benchmark to evaluate many of the most common uncertainty estimation methods, as well as two state-of-the-art uncertainty scores from OOD detection. We found that while all methods are well calibrated when there is no distribution shift, they become highly overconfident on many of the benchmark datasets. Methods based on the OOD uncertainty scores performed well relative to other methods, but the absolute performance is still far from perfect. This uncovers important limitations of current regression uncertainty estimation methods. Our work thus serves as a challenge to the research community, to develop methods which actually produce calibrated prediction intervals across all benchmark datasets. To that end, future directions include exploring the use of more sophisticated models for $p(x)$ within selective prediction – hopefully closing the performance gap between synthetic and real-world datasets – and employing alternative DNN backbone architectures. We hope that our benchmark will spur more work on how to develop truly reliable regression uncertainty estimation methods.

### Acknowledgements

This research was supported by the Swedish Research Council via the projects *NewLEADS – New Directions in Learning Dynamical Systems* (contract number: 621-2016-06079) and *Deep Probabilistic Regression – New Models and Learning Algorithms* (contract number: 2021-04301), and by the *Kjell & Märta Beijer Foundation*. We also thank Ludvig Hult and Dave Zachariah for providing helpful feedback on an early manuscript.

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

## A  Dataset Details

More detailed descriptions of the 12 datasets from Section 3.1 are provided below.

**Cells**  Given a synthetic fluorescence microscopy image $x$, the task is to predict the number of cells $y$ in the image. We utilize the Cell-200 dataset from Ding et al. (2021; 2020), consisting of $200\,000$ grayscale images of size $64 \times 64$. We randomly draw $10\,000$ train images, $2\,000$ val images and $10\,000$ test images. Thus, there is no distribution shift between train/val and test. We therefore use this as a baseline dataset.

**Cells-Tails**  We create a variant of Cells with a clear distribution shift between train/val and test. For the $10\,000$ train images and $2\,000$ val images, the regression targets $y$ are limited to the range $]50, 150]$. For the $10\,000$ test images, the targets instead lie in the full original range $[1, 200]$.

**Cells-Gap**  Another variant of Cells with a clear distribution shift between train/val and test. For the $10\,000$ train images and $2\,000$ val images, the regression targets $y$ are limited to $[1, 50[\cup]150, 200]$. For the $10\,000$ test images, the targets instead lie in the full original range $[1, 200]$.

**ChairAngle**  Given a synthetic image $x$ of a rendered chair model, the task is to predict the yaw angle $y$ of the chair. We utilize the RC-49 dataset from Ding et al. (2021; 2020), which contains $64 \times 64$ RGB images of different chair models rendered at 899 yaw angles ranging from $0.1°$ to $89.9°$, with step size $0.1°$. We randomly split their training set and obtain $17\,640$ train images and $4\,410$ val images. By sub-sampling their test set we also get $11\,225$ test images. There is no obvious distribution shift between train/val and test, and we therefore use this as a second baseline dataset.

**ChairAngle-Tails**  We create a variant of ChairAngle with a clear distribution shift between train/val and test. For train and val, we limit the regression targets $y$ to the range $]15, 75[$. For test, the targets instead lie in the full original range $]0, 90[$. We obtain $11\,760$ train images, $2\,940$ val images and $11\,225$ test images.

**ChairAngle-Gap**  Another variant of ChairAngle with a clear distribution shift between train/val and test. For the $11\,760$ train images and $2\,940$ val images, the regression targets $y$ are limited to $]0, 30[\cup]60, 90[$. For the $11\,225$ test images, the targets instead lie in the full original range $]0, 90[$.

**AssetWealth**  Given a satellite image $x$ (8 image channels), the task is to predict the asset wealth index $y$ of the region. We utilize the PovertyMap-Wilds dataset (Koh et al., 2021), which is a variant of the dataset collected by Yeh et al. (2020). We use the training, validation-ID and test-OOD subsets of the data, giving us $9\,797$ train images, $1\,000$ val images and $3\,963$ test images. We resize the images from size $224 \times 224$ to $64 \times 64$. Train/val and test contain satellite images from disjoint sets of African countries, creating a distribution shift.

**VentricularVolume**  Given an echocardiogram image $x$ of a human heart, the task is to predict the volume $y$ of the left ventricle. We utilize the EchoNet-Dynamic dataset by Ouyang et al. (2020), which contains $10\,030$ echocardiogram videos. Each video captures a complete cardiac cycle and is labeled with measurements of the left ventricular volume at two separate time points, representing end-systole (at the end of contraction - smaller volume) and end-diastole (just before contraction - larger volume). For each video, we extract just one of these volume measurements along with the corresponding video frame. To create a clear distribution shift between train/val and test, we select the end systolic volume (smaller volume) for train and val, but the end diastolic volume (larger volume) for test. We utilize the provided dataset splits, giving us $7\,460$ train images, $1\,288$ val images and $1\,276$ test images. We resize the images from size $112 \times 112$ to $64 \times 64$.

**BrainTumourPixels**  Given an image slice $x$ of a brain MRI scan, the task is to predict the number of pixels $y$ in the image which are labeled as brain tumour. We utilize the brain tumour dataset of the medical segmentation decathlon (Simpson et al., 2019; Antonelli et al., 2022), which is a subset of the data used in the 2016 and 2017 BraTS challenges (Bakas et al., 2018; 2017; Menze et al., 2014). The dataset contains 484 brain MRI scans with corresponding tumour segmentation masks. We split these scans $80\%/20\%/20\%$ into train, val and test sets. The scans are 3D volumes of size $240 \times 240 \times 155$. We convert each scan into 155 image slices of size $240 \times 240$, and create a regression target for each image by counting the number of labeled brain tumour pixels. We then also remove all images which contain no tumour pixels. The original image slices have 4 channels (FLAIR, T1w, T1gd, T2w), but we only use the first three and convert the slices into standard RGB images. This gives us $20\,614$ train images, $6\,116$ val images and $6\,252$ test images. We also resize the images from size $240 \times 240$ to $64 \times 64$.

**SkinLesionPixels**  Given a dermatoscopic image $x$ of a pigmented skin lesion, the task is to predict the number of pixels $y$ in the image which are labeled as lesion. We utilize the HAM10000 dataset by Tschandl et al. (2018), which contains 10 015 dermatoscopic images with corresponding skin lesion segmentation masks. HAM10000 consists of four different sub-datasets, three of which (ViDIR Legacy, ViDIR Current and ViDIR MoleMax) were collected in Austria, while the fourth sub-dataset (Rosendahl) was collected in Australia. To create a clear distribution shift between train/val and test, we use the Australian sub-dataset (Rosendahl) as our test set. After randomly splitting the remaining images 85%/15% into train and val sets, we obtain 6 592 train images, 1 164 val images and 2 259 test images. We then create a regression target for each image by counting the number of labeled skin lesion pixels. We also resize the images from size $450 \times 600$ to $64 \times 64$.

**HistologyNucleiPixels**  Given an H&E stained histology image $x$, the task is to predict the number of pixels $y$ in the image which are labeled as nuclei. We utilize the CoNSeP dataset by Graham et al. (2019), along with the pre-processed versions they provide of the Kumar (Kumar et al., 2017) and TNBC (Naylor et al., 2018) datasets. The datasets contain large H&E stained image tiles (of size $1\,000 \times 1\,000$ or $512 \times 512$) at 40× objective magnification, with corresponding nuclear segmentation masks. The three datasets were collected at different hospitals/institutions, with differing procedures for specimen preservation and staining. By using CoNSeP and Kumar for train/val and TNBC for test, we thus obtain a clear distribution shift. From the large image tiles, we extract $64 \times 64$ patches via regular gridding, and create a regression target for each image patch by counting the number of labeled nuclei pixels. We then also remove all images which contain no nuclei pixels. In the end, we obtain 10 808 train images, 2 702 val images and 2 267 test images.

**AerialBuildingPixels**  Given an aerial image $x$, the task is to predict the number of pixels $y$ in the image which are labeled as building. We utilize the Inria aerial image labeling dataset (Maggiori et al., 2017), which contains 180 large aerial images with corresponding building segmentation masks. The images are of size $5\,000 \times 5\,000$, and are captured at five different geographical locations. We use the images from two densely populated American cities (Austin and Chicago) for train/val, and the images from a more rural European area (West Tyrol, Austria) for test, thus obtaining a clear distribution shift. We first resize the images to size $1\,000 \times 1\,000$, and then extract $64 \times 64$ patches via regular gridding. We also create a regression target for each image patch by counting the number of labeled building pixels. After removal of all images which contain no building pixels, we finally obtain 11 184 train images, 2 797 val images and 3 890 test images.

The additional variants of CELLS and CHAIRANGLE in Figure 4 are specified as follows. CELLS: no difference in regression target range between train/val and test. CELLS-TAILS: target range $]50, 150]$ for train/val, $[1, 200]$ for test. We create three versions with intermediate target ranges (1: $[37.5, 163.5]$, 2: $[25, 176]$, 3: $[12.5, 188.5]$) for test. CHAIRANGLE: no difference in target range between train/val and test. CHAIRANGLE-GAP: target range $]0, 30[\cup]60, 90[$ for train/val, $]0, 90[$ for test. We create three versions with intermediate target ranges (1: $]0, 33.75[\cup]56.25, 90[$, 2: $]0, 37.5[\cup]52.5, 90[$, 3: $]0, 41.25[\cup]48.75, 90[$) for test.

## B  Additional Results & Method Variations

*Please note that the results in Table A1 - Table A42 not are rounded/truncated to only significant digits.*

To ensure that the test coverage results do not depend on our specific choice of studying 90% prediction intervals ($\alpha = 0.1$), we repeat most of the evaluation for two alternative miscoverage rates $\alpha$. Specifically, we redo the evaluation of 6/10 methods on 9/12 datasets with 80% ($\alpha = 0.2$) and 95% ($\alpha = 0.05$) prediction intervals. The results for 80% prediction intervals are given in Figure A7 - Figure A10 and Table A13 - Table A21, while the results for 95% prediction intervals are given in Figure A11 - Figure A14 and Table A22 - Table A30. We observe very similar trends overall. For example, all methods still have almost perfectly calibrated prediction intervals on CELLS and CHAIRANGLE, i.e. they all obtain a test coverage very close to 80%/95%, but only GAUSSIAN + SELECTIVE GMM remains well-calibrated on CELLS-TAILS and CHAIRANGLE-GAP. With the exception of BRAINTUMOURPIXELS, all methods are also significantly overconfident on all the real-world datasets.

Figure A15 shows test coverage results for the five common regression uncertainty estimation methods on the AERIALBUILDINGPIXELS dataset, and on two versions with different test sets. For all three datasets, train/val contains images from Austin and Chicago. For AERIALBUILDINGPIXELS, test contains images from

West Tyrol, Austria. For AERIALBUILDINGPIXELS-KITSAP, test instead contains images from Kitsap County, WA. For AERIALBUILDINGPIXELS-VIENNA, test contains images from Vienna, Austria. Intuitively, the distribution shift between train/val and test could potentially be smaller for AERIALBUILDINGPIXELS-KITSAP and AERIALBUILDINGPIXELS-VIENNA than for the original AERIALBUILDINGPIXELS, but we observe no clear trends in Figure A15.

Figure A16 & A17 present a study in which we aim to relate the test coverage performance to a quantitative measure of distribution shift ("distance" between the distributions of train/val and test), complementing our qualitative discussion in the *Performance Differences among Real-World Datasets are Mostly Logical* paragraph of Section 7. How to quantify the level of distribution shift in real-world datasets is however far from obvious, see e.g. Appendix E.1 in (Wenzel et al., 2022). We here explore if the difference in regression accuracy (MAE) on val and test can be adopted as such a measure, extending the approach by Wenzel et al. (2022) to our regression setting. We compute the average test coverage error for the five common regression uncertainty estimation methods, whereas the val/test MAE is for the CONFORMAL PREDICTION method (standard direct regression models). The results for the six synthetic datasets are presented in Figure A16, and for the six real-world datasets in Figure A17. We observe that, in general, a larger distribution shift measure does indeed seem to correspond to worse test coverage performance. Among the 12 datasets, HISTOLOGYNUCLEIPIXELS is the only one that quite clearly breaks this general trend.

Apart from the main comparison of the 10 methods specified in Section 4, we also evaluate a few method variations. The results for these experiments are provided in Table A31 - Table A42. For GAUSSIAN + SELECTIVE GMM, we vary the number of GMM mixture components from the standard $k = 4$ to $k = 2$ and $k = 8$, but observe no particularly consistent or significant trends in the results. Similarly, we vary the number of neighbors from the standard $k = 10$ to $k = 5$ and $k = 20$ for GAUSSIAN + SELECTIVE KNN, but observe no clear trends here either. For GAUSSIAN + SELECTIVE KNN, we also explore replacing the cosine similarity distance metric with L2 distance, again obtaining very similar results. Following Mukhoti et al. (2021b), we finally add spectral normalization (Miyato et al., 2018) for further feature-space regularization, but observe no significant improvements for GAUSSIAN + SELECTIVE GMM.

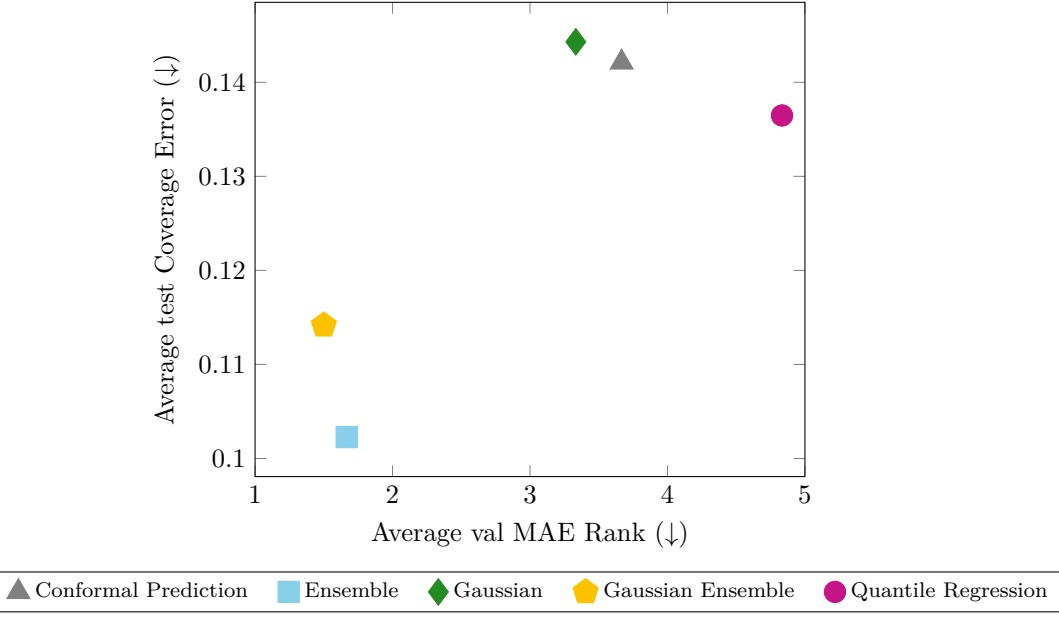

Figure A1: Performance comparison of the five common regression uncertainty estimation methods on the six *real-world* datasets, in terms of average test coverage error and average val MAE rank (the five methods are ranked 1 - 5 in terms of val MAE on each dataset, and then the average rank is computed). ENSEMBLE and GAUSSIAN ENSEMBLE achieve the best performance.

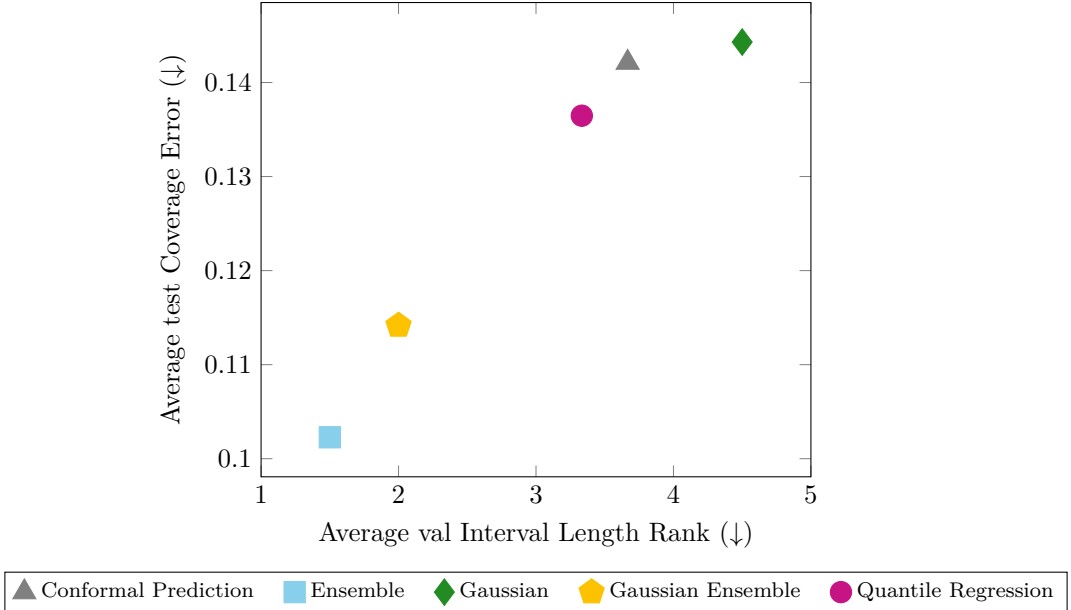

Figure A2: Performance comparison of the five common regression uncertainty estimation methods on the six *real-world* datasets, in terms of average test coverage error and average val interval length rank (the five methods are ranked 1 - 5 in terms of val interval length on each dataset, and then the average rank is computed). ENSEMBLE and GAUSSIAN ENSEMBLE achieve the best performance.

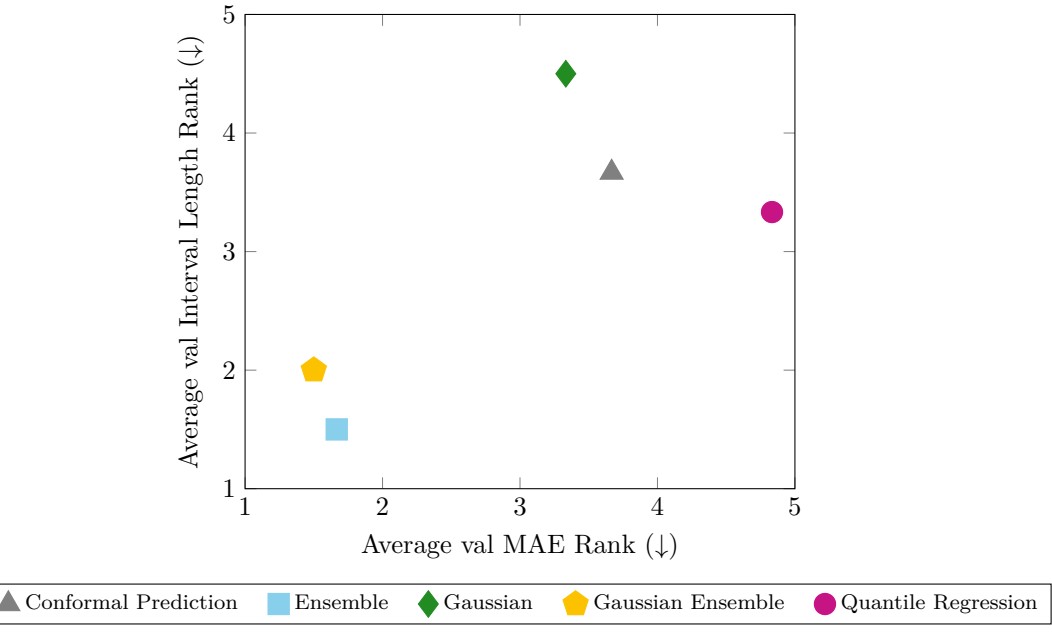

Figure A3: Performance comparison of the five common regression uncertainty estimation methods on the six *real-world* datasets, in terms of average val interval length rank and average val MAE rank. ENSEMBLE and GAUSSIAN ENSEMBLE achieve the best performance.

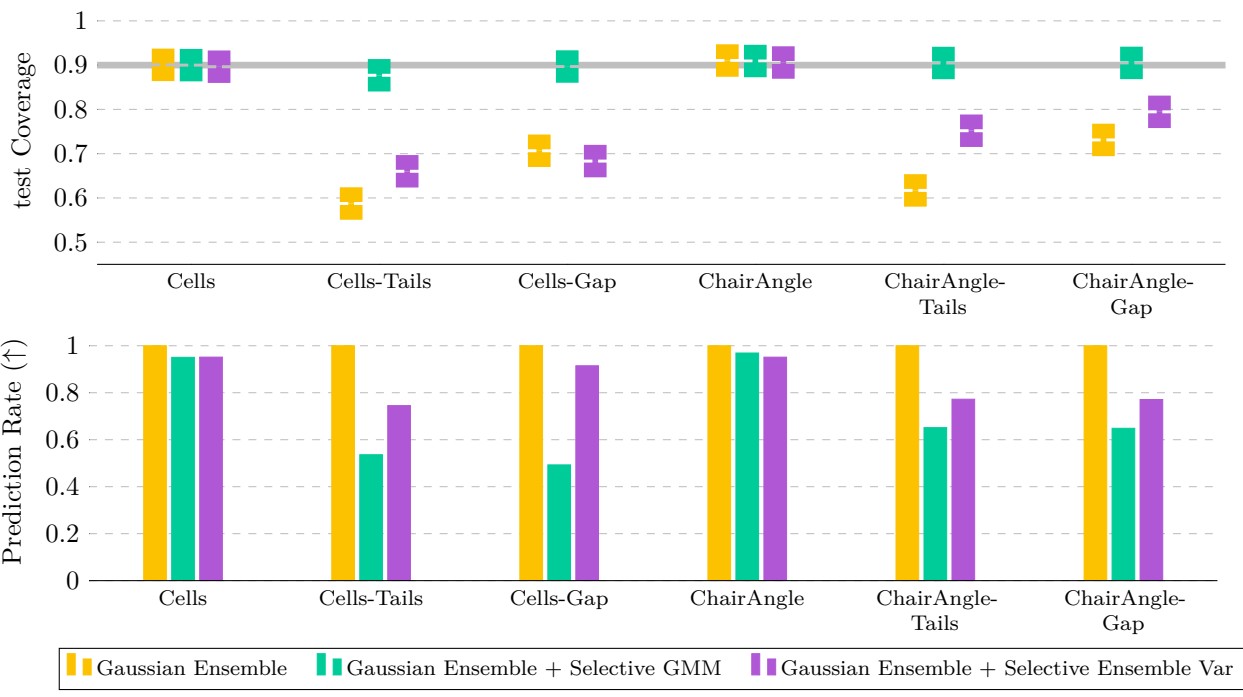

Figure A4: Results for the two selective prediction methods based on GAUSSIAN ENSEMBLE, on the six *synthetic* datasets. **Top:** Results in terms of test coverage. **Bottom:** Results in terms of test prediction rate.

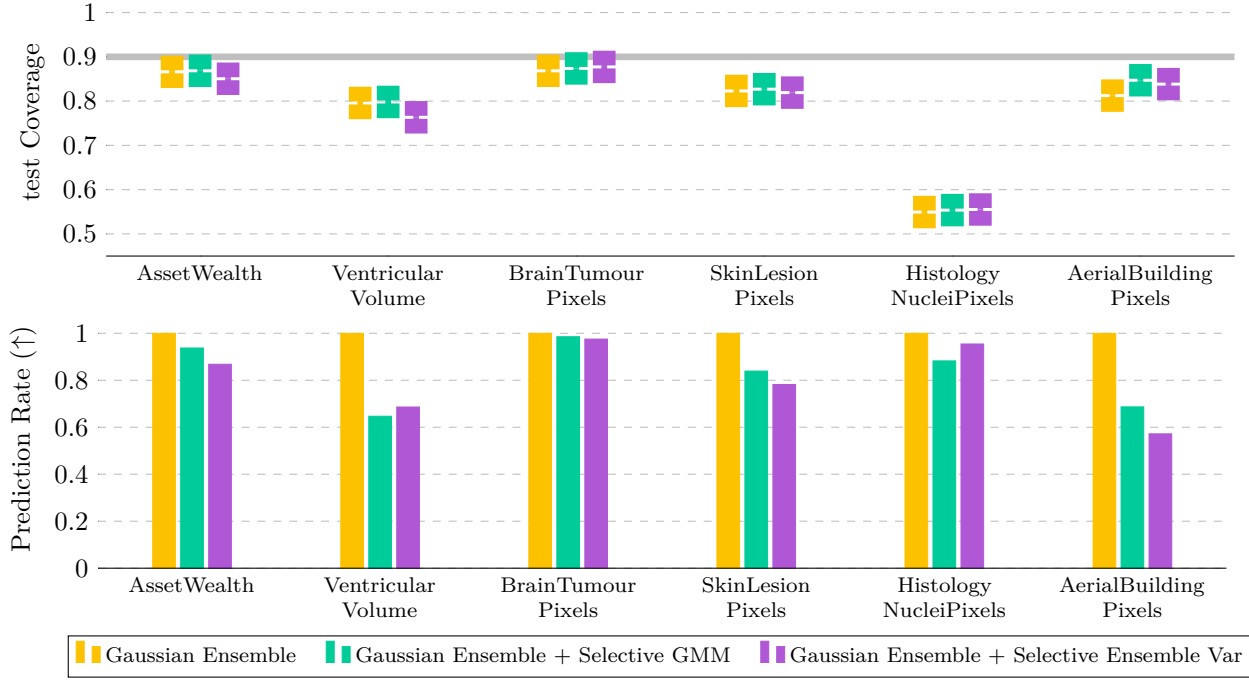

Figure A5: Results for the two selective prediction methods based on GAUSSIAN ENSEMBLE, on the *real-world* datasets. **Top:** Results in terms of test coverage. **Bottom:** Results in terms of test prediction rate.

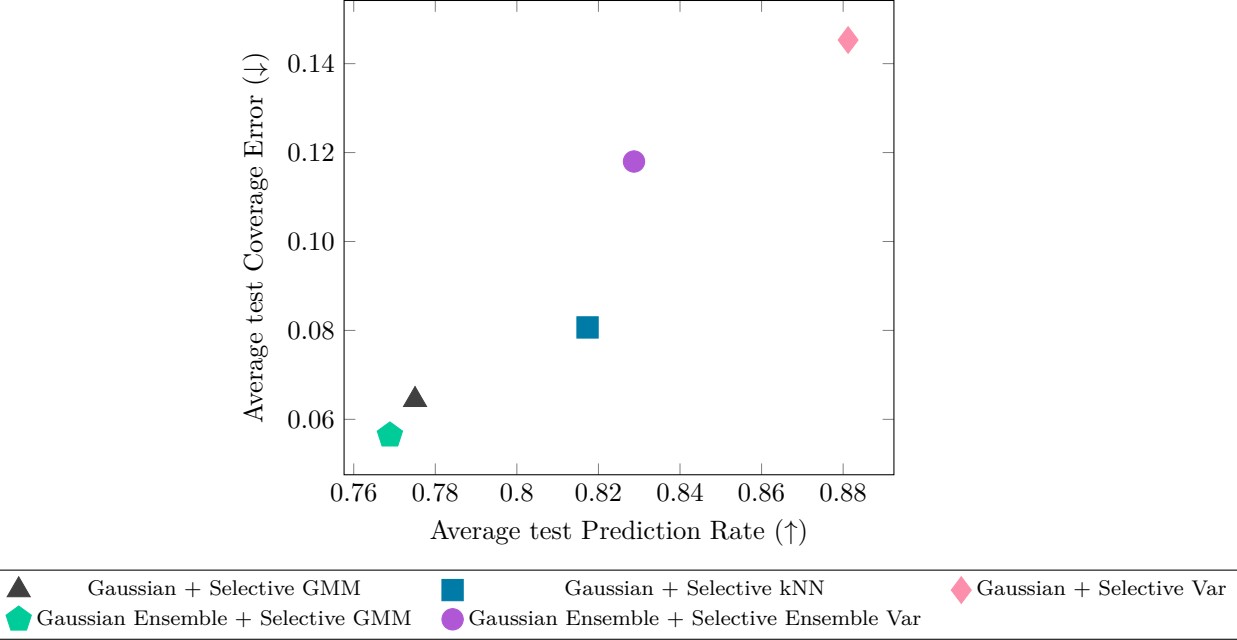

Figure A6: Performance comparison of the selective prediction methods across all 12 datasets, in terms of average test coverage error and test prediction rate. GAUSSIAN ENSEMBLE + SELECTIVE GMM achieves the best coverage error, but also has the lowest prediction rate. For these five methods, each improvement in terms of coverage error also corresponds to a decrease in prediction rate.

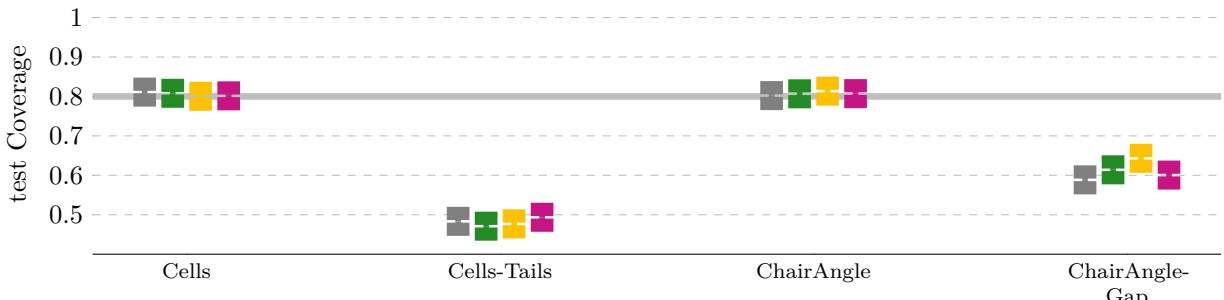

Figure A7: Miscoverage rate $\alpha = 0.2$: Results in terms of test coverage for four of the common regression uncertainty estimation methods (Conformal Prediction, Gaussian, Gaussian Ensemble, Quantile Regression), on four of the *synthetic* datasets. See Table A13 - Table A16 for other metrics.

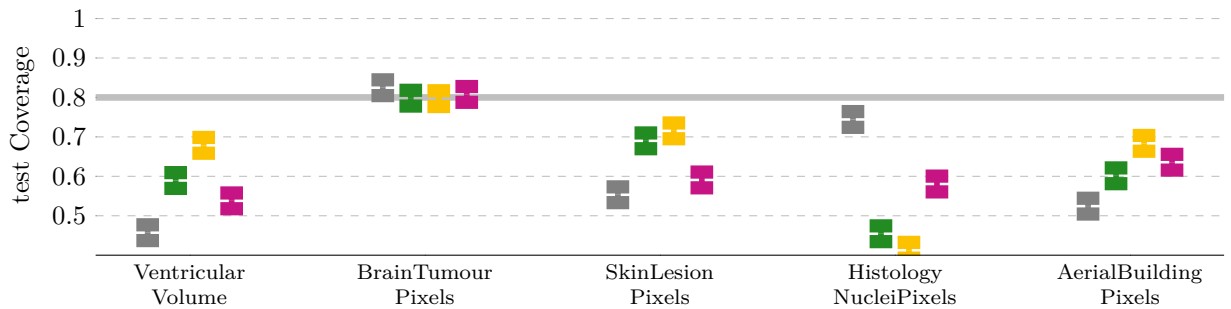

Figure A8: Miscoverage rate $\alpha = 0.2$: Results in terms of test coverage for four of the common regression uncertainty estimation methods (Conformal Prediction, Gaussian, Gaussian Ensemble, Quantile Regression), on five of the *real-world* datasets. See Table A17 - Table A21 for other metrics.

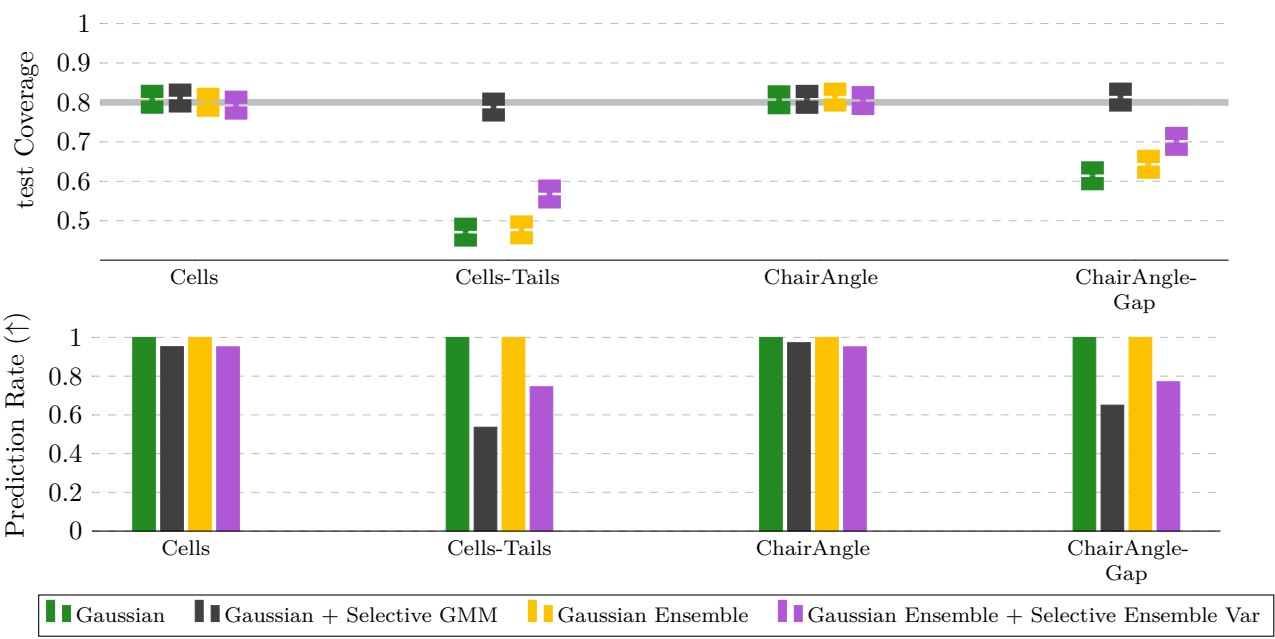

Figure A9: Miscoverage rate $\alpha = 0.2$: Results for two of the selective prediction methods, on four of the *synthetic* datasets. See Table A13 - Table A16 for other metrics.

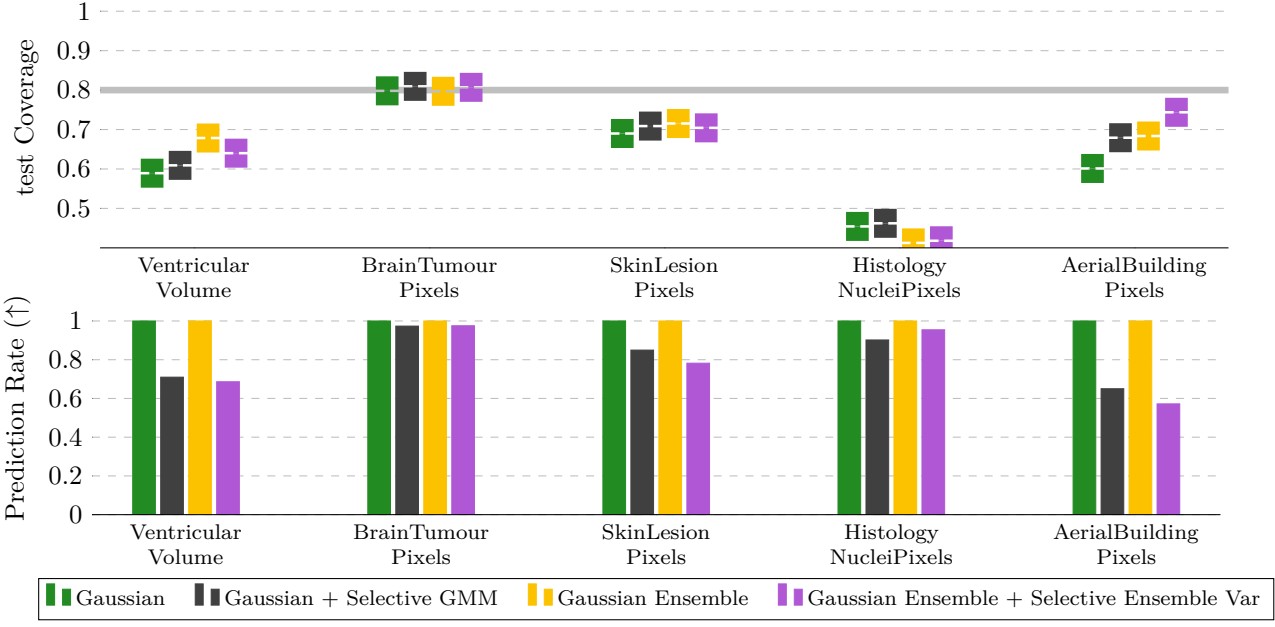

Figure A10: Miscoverage rate $\alpha = 0.2$: Results for two of the selective prediction methods, on five of the *real-world* datasets. See Table A17 - Table A21 for other metrics.

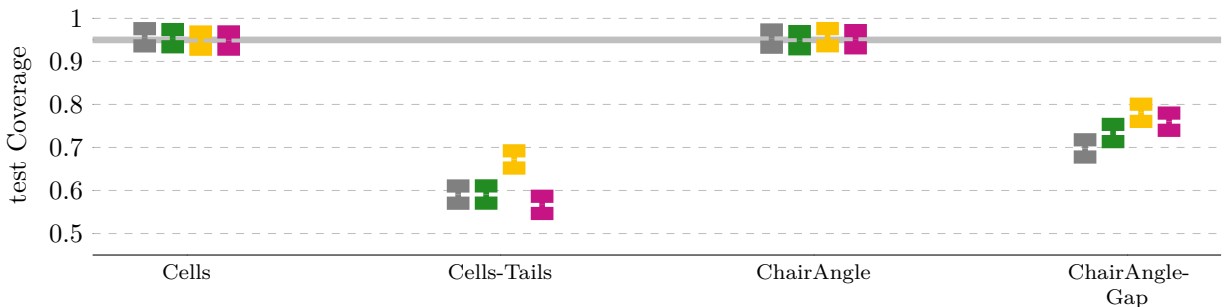

Figure A11: Miscoverage rate $\alpha = 0.05$: Results in terms of test coverage for four of the common regression uncertainty estimation methods (Conformal Prediction, Gaussian, Gaussian Ensemble, Quantile Regression), on four of the *synthetic* datasets. See Table A22 - Table A25 for other metrics.

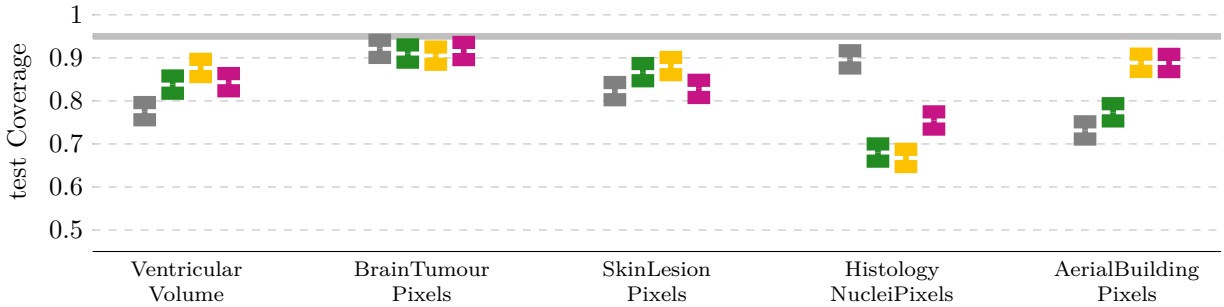

Figure A12: Miscoverage rate $\alpha = 0.05$: Results in terms of test coverage for four of the common regression uncertainty estimation methods (Conformal Prediction, Gaussian, Gaussian Ensemble, Quantile Regression), on five of the *real-world* datasets. See Table A26 - Table A30 for other metrics.

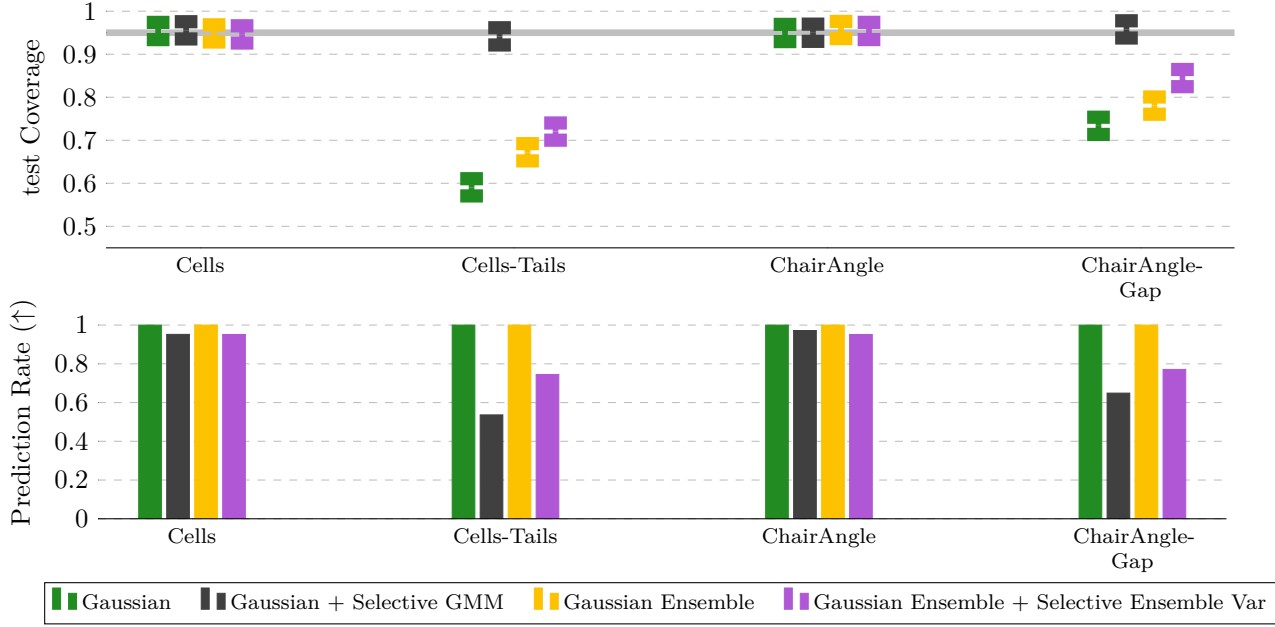

Figure A13: Miscoverage rate $\alpha = 0.05$: Results for two of the selective prediction methods, on four of the *synthetic* datasets. See Table A22 - Table A25 for other metrics.

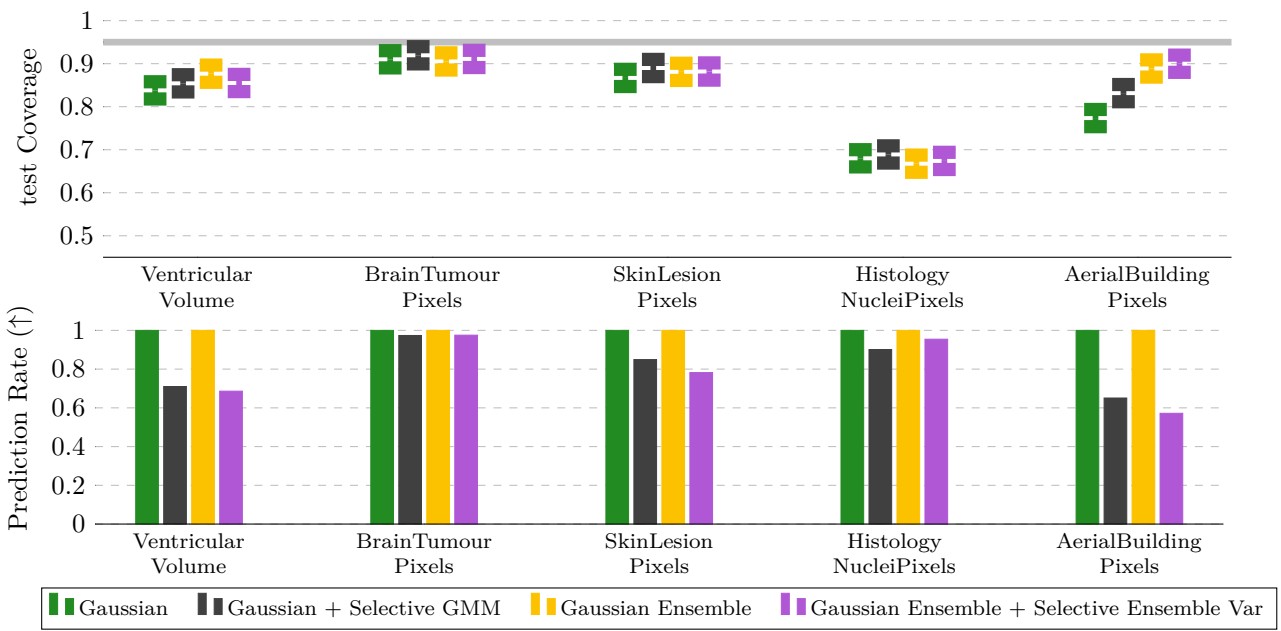

Figure A14: Miscoverage rate $\alpha = 0.05$: Results for two of the selective prediction methods, on five of the *real-world* datasets. See Table A26 - Table A30 for other metrics.

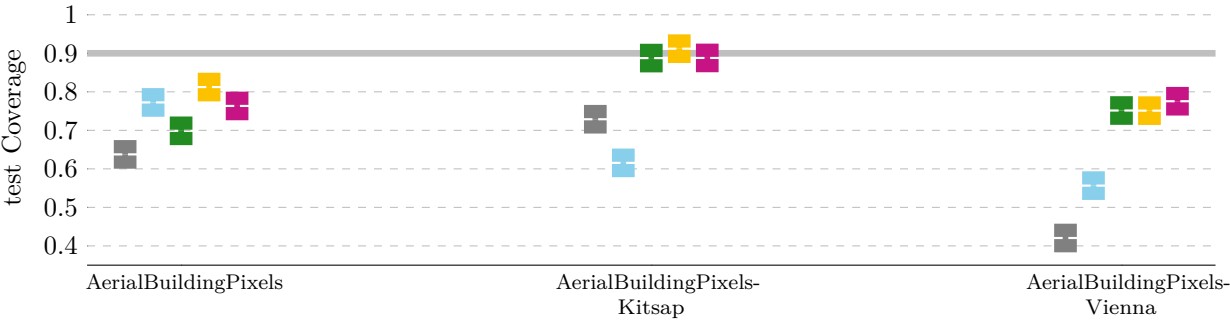

Figure A15: Results for the five common regression uncertainty estimation methods (Conformal Prediction, Ensemble, Gaussian, Gaussian Ensemble, Quantile Regression) on the AERIALBUILDINGPIXELS dataset, and on two versions with different test sets. For all three datasets, train/val contains images from Austin and Chicago. For AERIALBUILDINGPIXELS, test contains images from West Tyrol, Austria. For AERIALBUILDINGPIXELS-KITSAP, test instead contains images from Kitsap County, WA. For AERIALBUILDINGPIXELS-VIENNA, test contains images from Vienna, Austria.

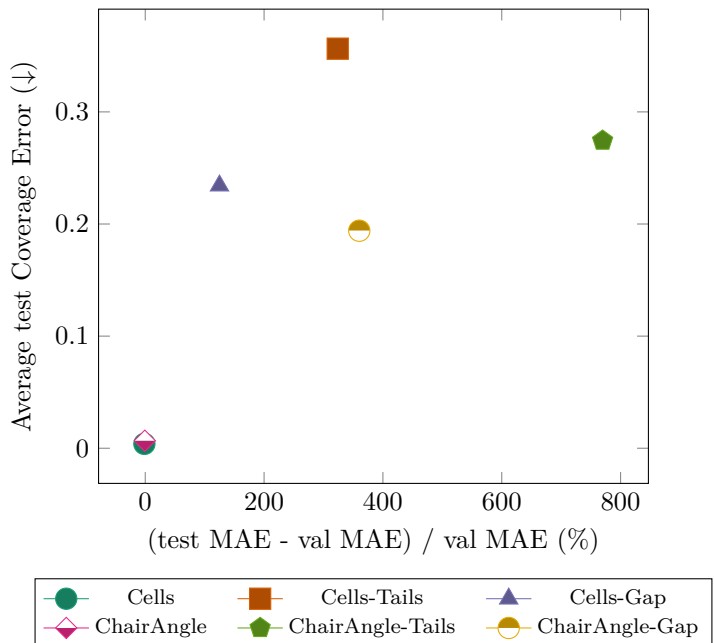

Figure A16: Using the difference in regression accuracy (MAE) on val and test as a quantitative measure of distribution shift in each dataset (inspired by Wenzel et al. (2022), extended to our regression setting), and comparing this to the test coverage performance. The average test coverage error is computed for the five common regression uncertainty estimation methods. The val/test MAE is for the Conformal Prediction method (standard direct regression models). Results for the six *synthetic* datasets. In general, a larger distribution shift measure corresponds to worse test coverage performance.

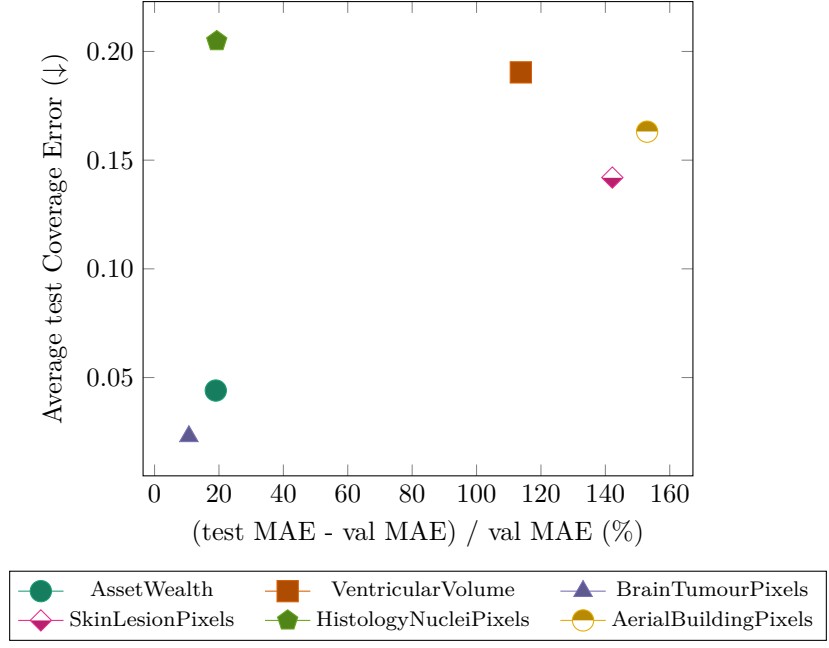

Figure A17: Exactly the same comparison as in Figure A16, but for the six *real-world* datasets. A larger distribution shift measure generally corresponds to worse test coverage performance also in this case. The HISTOLOGYNUCLEIPIXELS dataset is somewhat of an outlier.

Table A1: Complete results on the Cells dataset.

| Method | val MAE (↓) | val Interval Length (↓) | test Coverage (≥ 0.90) | test Prediction Rate (↑) |
|---|---|---|---|---|
| Conformal Prediction | 4.03121 ± 2.78375 | 18.8216 ± 11.8302 | 0.9117 ± 0.00275173 | 1.0 |
| Ensemble | 2.22525 ± 0.471309 | 12.2881 ± 2.78795 | 0.8994 ± 0.00689986 | 1.0 |
| Gaussian | 3.61704 ± 1.13624 | 14.5492 ± 4.44927 | 0.90286 ± 0.00594629 | 1.0 |
| Gaussian Ensemble | 2.78757 ± 1.16951 | 17.1285 ± 4.33367 | 0.90062 ± 0.0027571 | 1.0 |
| Quantile Regression | 3.70405 ± 0.81647 | 13.8023 ± 2.19085 | 0.90486 ± 0.006771 | 1.0 |
| Gaussian + Selective GMM | 3.61704 ± 1.13624 | 14.5492 ± 4.44927 | 0.905241 ± 0.00448315 | 0.95216 ± 0.00290558 |
| Gaussian + Selective kNN | 3.61704 ± 1.13624 | 14.5492 ± 4.44927 | 0.900069 ± 0.00677637 | 0.94656 ± 0.00470472 |
| Gaussian + Selective Variance | 3.61704 ± 1.13624 | 14.5492 ± 4.44927 | 0.904839 ± 0.00795449 | 0.95712 ± 0.00200938 |
| Gaussian Ensemble + Selective GMM | 2.78757 ± 1.16951 | 17.1285 ± 4.33367 | 0.899999 ± 0.00311328 | 0.95066 ± 0.0023105 |
| Gaussian Ensemble + Selective Ens Var | 2.78757 ± 1.16951 | 17.1285 ± 4.33367 | 0.896465 ± 0.0015705 | 0.95156 ± 0.00215277 |

Table A2: Complete results on the Cells-Tails dataset.

| Method | val MAE (↓) | val Interval Length (↓) | test Coverage (≥ 0.90) | test Prediction Rate (↑) |
|---|---|---|---|---|
| Conformal Prediction | 3.56733 ± 1.0818 | 14.2128 ± 4.23582 | 0.55356 ± 0.0299096 | 1.0 |
| Ensemble | 1.83461 ± 0.178937 | 9.8697 ± 1.6672 | 0.50078 ± 0.007608 | 1.0 |
| Gaussian | 4.05446 ± 1.33153 | 15.4321 ± 4.98796 | 0.54402 ± 0.0347525 | 1.0 |
| Gaussian Ensemble | 2.40691 ± 0.580524 | 10.9696 ± 1.70051 | 0.5874 ± 0.0479263 | 1.0 |
| Quantile Regression | 3.32571 ± 1.24578 | 13.0848 ± 3.54239 | 0.52222 ± 0.0284102 | 1.0 |
| Gaussian + Selective GMM | 4.05446 ± 1.33153 | 15.4321 ± 4.98796 | 0.889825 ± 0.0193021 | 0.53654 ± 0.0101012 |
| Gaussian + Selective kNN | 4.05446 ± 1.33153 | 15.4321 ± 4.98796 | 0.859179 ± 0.0255173 | 0.56692 ± 0.0127107 |
| Gaussian + Selective Variance | 4.05446 ± 1.33153 | 15.4321 ± 4.98796 | 0.687475 ± 0.0659807 | 0.6914 ± 0.0651196 |
| Gaussian Ensemble + Selective GMM | 2.40691 ± 0.580524 | 10.9696 ± 1.70051 | 0.877068 ± 0.0222502 | 0.53604 ± 0.0125903 |
| Gaussian Ensemble + Selective Ens Var | 2.40691 ± 0.580524 | 10.9696 ± 1.70051 | 0.660212 ± 0.0325832 | 0.74512 ± 0.0438128 |

Table A3: Complete results on the Cells-Gap dataset.

| Method | val MAE (↓) | val Interval Length (↓) | test Coverage (≥ 0.90) | test Prediction Rate (↑) |
|---|---|---|---|---|
| Conformal Prediction | 3.67702 ± 1.33587 | 17.1152 ± 6.49211 | 0.64002 ± 0.0620287 | 1.0 |
| Ensemble | 2.7261 ± 0.705274 | 13.4061 ± 2.40513 | 0.6771 ± 0.0544788 | 1.0 |
| Gaussian | 3.53089 ± 1.0619 | 15.6396 ± 6.23458 | 0.66028 ± 0.114703 | 1.0 |
| Gaussian Ensemble | 3.46118 ± 0.95429 | 18.707 ± 4.13287 | 0.7066 ± 0.0317638 | 1.0 |
| Quantile Regression | 4.75328 ± 1.73499 | 18.108 ± 4.21716 | 0.64488 ± 0.12875 | 1.0 |
| Gaussian + Selective GMM | 3.53089 ± 1.0619 | 15.6396 ± 6.23458 | 0.890569 ± 0.00953089 | 0.49372 ± 0.0025926 |
| Gaussian + Selective kNN | 3.53089 ± 1.0619 | 15.6396 ± 6.23458 | 0.874032 ± 0.0432364 | 0.53646 ± 0.0139864 |
| Gaussian + Selective Variance | 3.53089 ± 1.0619 | 15.6396 ± 6.23458 | 0.652766 ± 0.117192 | 0.9748 ± 0.000940213 |
| Gaussian Ensemble + Selective GMM | 3.46118 ± 0.95429 | 18.707 ± 4.13287 | 0.896848 ± 0.0127879 | 0.49278 ± 0.00192914 |
| Gaussian Ensemble + Selective Ens Var | 3.46118 ± 0.95429 | 18.707 ± 4.13287 | 0.68326 ± 0.0299799 | 0.9147 ± 0.0182427 |

Table A4: Complete results on the ChairAngle dataset.

| Method | val MAE (↓) | val Interval Length (↓) | test Coverage (≥ 0.90) | test Prediction Rate (↑) |
|---|---|---|---|---|
| Conformal Prediction | 0.289127 ± 0.081643 | 1.08752 ± 0.202247 | 0.905265 ± 0.00339915 | 1.0 |
| Ensemble | 0.189858 ± 0.0548452 | 0.788401 ± 0.137465 | 0.909915 ± 0.00399101 | 1.0 |
| Gaussian | 0.376692 ± 0.171928 | 1.37757 ± 0.382191 | 0.901577 ± 0.00308472 | 1.0 |
| Gaussian Ensemble | 0.361834 ± 0.165348 | 1.2044 ± 0.442422 | 0.910414 ± 0.00316257 | 1.0 |
| Quantile Regression | 0.851253 ± 0.544717 | 3.19741 ± 1.9751 | 0.906209 ± 0.00204038 | 1.0 |
| Gaussian + Selective GMM | 0.376692 ± 0.171928 | 1.37757 ± 0.382191 | 0.902482 ± 0.00436054 | 0.972739 ± 0.00191733 |
| Gaussian + Selective kNN | 0.376692 ± 0.171928 | 1.37757 ± 0.382191 | 0.90222 ± 0.00459694 | 0.975465 ± 0.00201785 |
| Gaussian + Selective Variance | 0.376692 ± 0.171928 | 1.37757 ± 0.382191 | 0.904805 ± 0.00765098 | 0.961782 ± 0.0193133 |
| Gaussian Ensemble + Selective GMM | 0.361834 ± 0.165348 | 1.2044 ± 0.442422 | 0.9093 ± 0.00297263 | 0.969033 ± 0.000940447 |
| Gaussian Ensemble + Selective Ens Var | 0.361834 ± 0.165348 | 1.2044 ± 0.442422 | 0.905905 ± 0.00336481 | 0.951359 ± 0.00453556 |

Table A5: Complete results on the CHAIRANGLE-TAILS dataset.

| Method | val MAE (↓) | val Interval Length (↓) | test Coverage (≥ 0.90) | test Prediction Rate (↑) |
|---|---|---|---|---|
| Conformal Prediction | $0.358162 \pm 0.168257$ | $1.31102 \pm 0.531401$ | $0.615804 \pm 0.00527695$ | 1.0 |
| Ensemble | $0.21931 \pm 0.0596705$ | $1.04074 \pm 0.245846$ | $0.611706 \pm 0.00345611$ | 1.0 |
| Gaussian | $0.241214 \pm 0.091736$ | $1.09417 \pm 0.382907$ | $0.622592 \pm 0.00714065$ | 1.0 |
| Gaussian Ensemble | $0.13365 \pm 0.0189933$ | $0.769172 \pm 0.0814039$ | $0.617016 \pm 0.0063698$ | 1.0 |
| Quantile Regression | $0.820934 \pm 0.653268$ | $2.9815 \pm 2.15473$ | $0.660953 \pm 0.0379624$ | 1.0 |
| Gaussian + Selective GMM | $0.241214 \pm 0.091736$ | $1.09417 \pm 0.382907$ | $0.901946 \pm 0.00382993$ | $0.655448 \pm 0.001058$ |
| Gaussian + Selective kNN | $0.241214 \pm 0.091736$ | $1.09417 \pm 0.382907$ | $0.860311 \pm 0.00617031$ | $0.703038 \pm 0.00691227$ |
| Gaussian + Selective Variance | $0.241214 \pm 0.091736$ | $1.09417 \pm 0.382907$ | $0.647559 \pm 0.0360179$ | $0.924383 \pm 0.0751205$ |
| Gaussian Ensemble + Selective GMM | $0.13365 \pm 0.0189933$ | $0.769172 \pm 0.0814039$ | $0.904807 \pm 0.00394199$ | $0.651314 \pm 0.0013893$ |
| Gaussian Ensemble + Selective Ens Var | $0.13365 \pm 0.0189933$ | $0.769172 \pm 0.0814039$ | $0.751845 \pm 0.0131765$ | $0.772401 \pm 0.00878839$ |

Table A6: Complete results on the CHAIRANGLE-GAP dataset.

| Method | val MAE (↓) | val Interval Length (↓) | test Coverage (≥ 0.90) | test Prediction Rate (↑) |
|---|---|---|---|---|
| Conformal Prediction | $0.35034 \pm 0.161819$ | $1.35 \pm 0.547089$ | $0.659546 \pm 0.00916108$ | 1.0 |
| Ensemble | $0.22898 \pm 0.0853825$ | $1.1222 \pm 0.174147$ | $0.749951 \pm 0.0155327$ | 1.0 |
| Gaussian | $0.454516 \pm 0.280174$ | $2.09212 \pm 0.933756$ | $0.69363 \pm 0.0345876$ | 1.0 |
| Gaussian Ensemble | $0.226352 \pm 0.0677413$ | $1.30588 \pm 0.136235$ | $0.731065 \pm 0.0126216$ | 1.0 |
| Quantile Regression | $0.639151 \pm 0.296536$ | $3.29137 \pm 1.53269$ | $0.695697 \pm 0.0413613$ | 1.0 |
| Gaussian + Selective GMM | $0.454516 \pm 0.280174$ | $2.09212 \pm 0.933756$ | $0.91215 \pm 0.00604745$ | $0.649372 \pm 0.00334919$ |
| Gaussian + Selective kNN | $0.454516 \pm 0.280174$ | $2.09212 \pm 0.933756$ | $0.911574 \pm 0.00376608$ | $0.673764 \pm 0.0113432$ |
| Gaussian + Selective Variance | $0.454516 \pm 0.280174$ | $2.09212 \pm 0.933756$ | $0.690436 \pm 0.0366025$ | $0.981292 \pm 0.0152406$ |
| Gaussian Ensemble + Selective GMM | $0.226352 \pm 0.0677413$ | $1.30588 \pm 0.136235$ | $0.905039 \pm 0.00229837$ | $0.648624 \pm 0.00094348$ |
| Gaussian Ensemble + Selective Ens Var | $0.226352 \pm 0.0677413$ | $1.30588 \pm 0.136235$ | $0.794531 \pm 0.014876$ | $0.7711 \pm 0.0191172$ |

Table A7: Complete results on the ASSETWEALTH dataset.

| Method | val MAE (↓) | val Interval Length (↓) | test Coverage (≥ 0.90) | test Prediction Rate (↑) |
|---|---|---|---|---|
| Conformal Prediction | $0.346532 \pm 0.00578306$ | $1.5838 \pm 0.0318086$ | $0.87136 \pm 0.0090944$ | 1.0 |
| Ensemble | $0.320002 \pm 0.00264202$ | $1.45568 \pm 0.0129312$ | $0.87348 \pm 0.00598963$ | 1.0 |
| Gaussian | $0.367501 \pm 0.0416437$ | $1.597 \pm 0.207599$ | $0.844966 \pm 0.0456831$ | 1.0 |
| Gaussian Ensemble | $0.3295 \pm 0.00783906$ | $1.42071 \pm 0.0485437$ | $0.866162 \pm 0.00711672$ | 1.0 |
| Quantile Regression | $0.404279 \pm 0.0683226$ | $1.58957 \pm 0.161689$ | $0.823921 \pm 0.0313343$ | 1.0 |
| Gaussian + Selective GMM | $0.367501 \pm 0.0416437$ | $1.597 \pm 0.207599$ | $0.850824 \pm 0.047533$ | $0.93838 \pm 0.0170443$ |
| Gaussian + Selective kNN | $0.367501 \pm 0.0416437$ | $1.597 \pm 0.207599$ | $0.852107 \pm 0.0474734$ | $0.933586 \pm 0.0203541$ |
| Gaussian + Selective Variance | $0.367501 \pm 0.0416437$ | $1.597 \pm 0.207599$ | $0.846981 \pm 0.0430247$ | $0.926874 \pm 0.0159098$ |
| Gaussian Ensemble + Selective GMM | $0.3295 \pm 0.00783906$ | $1.42071 \pm 0.0485437$ | $0.868444 \pm 0.00727312$ | $0.937371 \pm 0.0138213$ |
| Gaussian Ensemble + Selective Ens Var | $0.3295 \pm 0.00783906$ | $1.42071 \pm 0.0485437$ | $0.85047 \pm 0.00397082$ | $0.868282 \pm 0.0299149$ |

Table A8: Complete results on the VENTRICULARVOLUME dataset.

| Method | val MAE (↓) | val Interval Length (↓) | test Coverage (≥ 0.90) | test Prediction Rate (↑) |
|---|---|---|---|---|
| Conformal Prediction | $11.2471 \pm 0.201399$ | $47.4505 \pm 1.556$ | $0.603135 \pm 0.0125842$ | 1.0 |
| Ensemble | $10.2476 \pm 0.113707$ | $41.8445 \pm 0.750822$ | $0.707367 \pm 0.00877743$ | 1.0 |
| Gaussian | $12.7238 \pm 1.52197$ | $51.566 \pm 3.52739$ | $0.730878 \pm 0.0619045$ | 1.0 |
| Gaussian Ensemble | $10.1141 \pm 0.180661$ | $39.9817 \pm 1.91308$ | $0.795768 \pm 0.0231243$ | 1.0 |
| Quantile Regression | $12.4944 \pm 0.676265$ | $49.0448 \pm 3.33314$ | $0.710972 \pm 0.045171$ | 1.0 |
| Gaussian + Selective GMM | $12.7238 \pm 1.52197$ | $51.566 \pm 3.52739$ | $0.752046 \pm 0.0529087$ | $0.707994 \pm 0.0208741$ |
| Gaussian + Selective kNN | $12.7238 \pm 1.52197$ | $51.566 \pm 3.52739$ | $0.735105 \pm 0.0612413$ | $0.911599 \pm 0.0447475$ |
| Gaussian + Selective Variance | $12.7238 \pm 1.52197$ | $51.566 \pm 3.52739$ | $0.747868 \pm 0.0569984$ | $0.691693 \pm 0.0341016$ |
| Gaussian Ensemble + Selective GMM | $10.1141 \pm 0.180661$ | $39.9817 \pm 1.91308$ | $0.798094 \pm 0.0166674$ | $0.646865 \pm 0.0366201$ |
| Gaussian Ensemble + Selective Ens Var | $10.1141 \pm 0.180661$ | $39.9817 \pm 1.91308$ | $0.763412 \pm 0.0286772$ | $0.686207 \pm 0.0431835$ |

Table A9: Complete results on the BRAINTUMOURPIXELS dataset.

| Method | val MAE (↓) | val Interval Length (↓) | test Coverage (≥ 0.90) | test Prediction Rate (↑) |
|---|---|---|---|---|
| Conformal Prediction | $21.3163 \pm 0.45997$ | $93.7169 \pm 2.99061$ | $0.885925 \pm 0.00395743$ | 1.0 |
| Ensemble | $21.1133 \pm 0.210209$ | $90.3825 \pm 0.825858$ | $0.878183 \pm 0.00318872$ | 1.0 |
| Gaussian | $21.0625 \pm 0.358012$ | $93.6284 \pm 2.29916$ | $0.873544 \pm 0.00871896$ | 1.0 |
| Gaussian Ensemble | $20.5336 \pm 0.211421$ | $87.7414 \pm 0.818979$ | $0.868426 \pm 0.00328047$ | 1.0 |
| Quantile Regression | $22.0348 \pm 0.697606$ | $94.3249 \pm 3.07265$ | $0.879079 \pm 0.00380396$ | 1.0 |
| Gaussian + Selective GMM | $21.0625 \pm 0.358012$ | $93.6284 \pm 2.29916$ | $0.883515 \pm 0.0138279$ | $0.973576 \pm 0.0187252$ |
| Gaussian + Selective kNN | $21.0625 \pm 0.358012$ | $93.6284 \pm 2.29916$ | $0.891264 \pm 0.00734602$ | $0.947185 \pm 0.0178434$ |
| Gaussian + Selective Variance | $21.0625 \pm 0.358012$ | $93.6284 \pm 2.29916$ | $0.878666 \pm 0.00735197$ | $0.978791 \pm 0.00694672$ |
| Gaussian Ensemble + Selective GMM | $20.5336 \pm 0.211421$ | $87.7414 \pm 0.818979$ | $0.873824 \pm 0.00466519$ | $0.985349 \pm 0.00461984$ |
| Gaussian Ensemble + Selective Ens Var | $20.5336 \pm 0.211421$ | $87.7414 \pm 0.818979$ | $0.877211 \pm 0.00302057$ | $0.975368 \pm 0.00323397$ |

Table A10: Complete results on the SkinLesionPixels dataset.

| Method | val MAE (↓) | val Interval Length (↓) | test Coverage (≥ 0.90) | test Prediction Rate (↑) |
|---|---|---|---|---|
| Conformal Prediction | $107.514 \pm 1.87464$ | $492.922 \pm 14.1865$ | $0.708012 \pm 0.00917777$ | 1.0 |
| Ensemble | $99.3156 \pm 0.997867$ | $353.842 \pm 4.73577$ | $0.742098 \pm 0.00852236$ | 1.0 |
| Gaussian | $105.417 \pm 1.15178$ | $535.139 \pm 215.446$ | $0.797255 \pm 0.0270734$ | 1.0 |
| Gaussian Ensemble | $100.639 \pm 0.464183$ | $472.14 \pm 85.2654$ | $0.822931 \pm 0.0134328$ | 1.0 |
| Quantile Regression | $113.076 \pm 3.20875$ | $405.904 \pm 4.75158$ | $0.719788 \pm 0.0213826$ | 1.0 |
| Gaussian + Selective GMM | $105.417 \pm 1.15178$ | $535.139 \pm 215.446$ | $0.821515 \pm 0.0137705$ | $0.849579 \pm 0.0206181$ |
| Gaussian + Selective kNN | $105.417 \pm 1.15178$ | $535.139 \pm 215.446$ | $0.813027 \pm 0.0306586$ | $0.927047 \pm 0.00830718$ |
| Gaussian + Selective Variance | $105.417 \pm 1.15178$ | $535.139 \pm 215.446$ | $0.799821 \pm 0.0150855$ | $0.77946 \pm 0.059169$ |
| Gaussian Ensemble + Selective GMM | $100.639 \pm 0.464183$ | $472.14 \pm 85.2654$ | $0.826921 \pm 0.00719649$ | $0.839132 \pm 0.0108815$ |
| Gaussian Ensemble + Selective Ens Var | $100.639 \pm 0.464183$ | $472.14 \pm 85.2654$ | $0.819129 \pm 0.00910128$ | $0.782116 \pm 0.0109776$ |

Table A11: Complete results on the HistologyNucleiPixels dataset.

| Method | val MAE (↓) | val Interval Length (↓) | test Coverage (≥ 0.90) | test Prediction Rate (↑) |
|---|---|---|---|---|
| Conformal Prediction | $217.887 \pm 3.71766$ | $993.665 \pm 18.3282$ | $0.841288 \pm 0.0128314$ | 1.0 |
| Ensemble | $197.078 \pm 1.99489$ | $853.361 \pm 14.1904$ | $0.812704 \pm 0.00664074$ | 1.0 |
| Gaussian | $211.795 \pm 10.0239$ | $1211.83 \pm 396.946$ | $0.588796 \pm 0.0912794$ | 1.0 |
| Gaussian Ensemble | $196.785 \pm 2.14454$ | $1108.53 \pm 145.034$ | $0.54936 \pm 0.05995$ | 1.0 |
| Quantile Regression | $227.895 \pm 9.37415$ | $914.909 \pm 45.995$ | $0.684076 \pm 0.0428704$ | 1.0 |
| Gaussian + Selective GMM | $211.795 \pm 10.0239$ | $1211.83 \pm 396.946$ | $0.59554 \pm 0.0956742$ | $0.90569 \pm 0.0453555$ |
| Gaussian + Selective kNN | $211.795 \pm 10.0239$ | $1211.83 \pm 396.946$ | $0.608929 \pm 0.0805954$ | $0.846228 \pm 0.0482832$ |
| Gaussian + Selective Variance | $211.795 \pm 10.0239$ | $1211.83 \pm 396.946$ | $0.588598 \pm 0.0913593$ | $0.998765 \pm 0.00134954$ |
| Gaussian Ensemble + Selective GMM | $196.785 \pm 2.14454$ | $1108.53 \pm 145.034$ | $0.553725 \pm 0.0642274$ | $0.882488 \pm 0.0156395$ |
| Gaussian Ensemble + Selective Ens Var | $196.785 \pm 2.14454$ | $1108.53 \pm 145.034$ | $0.555149 \pm 0.0620052$ | $0.954477 \pm 0.0256246$ |

Table A12: Complete results on the AerialBuildingPixels dataset.

| Method | val MAE (↓) | val Interval Length (↓) | test Coverage (≥ 0.90) | test Prediction Rate (↑) |
|---|---|---|---|---|
| Conformal Prediction | $235.417 \pm 7.16096$ | $1038.45 \pm 51.5565$ | $0.637584 \pm 0.0691102$ | 1.0 |
| Ensemble | $199.206 \pm 2.76543$ | $798.312 \pm 6.80986$ | $0.772494 \pm 0.0227288$ | 1.0 |
| Gaussian | $217.877 \pm 1.72493$ | $929.562 \pm 47.6606$ | $0.698766 \pm 0.0662263$ | 1.0 |
| Gaussian Ensemble | $208.487 \pm 1.03581$ | $885.349 \pm 27.8584$ | $0.812339 \pm 0.0617386$ | 1.0 |
| Quantile Regression | $242.284 \pm 6.0122$ | $909.191 \pm 24.9528$ | $0.763342 \pm 0.0902358$ | 1.0 |
| Gaussian + Selective GMM | $217.877 \pm 1.72493$ | $929.562 \pm 47.6606$ | $0.76535 \pm 0.0388677$ | $0.652082 \pm 0.0990489$ |
| Gaussian + Selective kNN | $217.877 \pm 1.72493$ | $929.562 \pm 47.6606$ | $0.651867 \pm 0.0771154$ | $0.840103 \pm 0.0463989$ |
| Gaussian + Selective Variance | $217.877 \pm 1.72493$ | $929.562 \pm 47.6606$ | $0.725714 \pm 0.0779575$ | $0.708226 \pm 0.103803$ |
| Gaussian Ensemble + Selective GMM | $208.487 \pm 1.03581$ | $885.349 \pm 27.8584$ | $0.847101 \pm 0.038266$ | $0.686787 \pm 0.0278702$ |
| Gaussian Ensemble + Selective Ens Var | $208.487 \pm 1.03581$ | $885.349 \pm 27.8584$ | $0.838352 \pm 0.0312236$ | $0.571928 \pm 0.0402727$ |

Table A13: Miscoverage rate $\alpha = 0.2$: Results on the Cells dataset.

| Method | val MAE (↓) | val Interval Length (↓) | test Coverage (≥ 0.80) | test Prediction Rate (↑) |
|---|---|---|---|---|
| Conformal Prediction | $4.03121 \pm 2.78375$ | $14.042 \pm 10.2791$ | $0.811 \pm 0.0100485$ | 1.0 |
| Gaussian | $3.61704 \pm 1.13624$ | $11.544 \pm 3.59513$ | $0.808 \pm 0.00859046$ | 1.0 |
| Gaussian Ensemble | $2.78757 \pm 1.16951$ | $13.1298 \pm 3.34997$ | $0.80044 \pm 0.00316582$ | 1.0 |
| Quantile Regression | $4.40758 \pm 1.47411$ | $13.3314 \pm 4.39511$ | $0.80152 \pm 0.00667005$ | 1.0 |
| Gaussian + Selective GMM | $3.61704 \pm 1.13624$ | $11.544 \pm 3.59513$ | $0.811076 \pm 0.00507358$ | $0.95216 \pm 0.00280043$ |
| Gaussian Ensemble + Selective Ens Var | $2.78757 \pm 1.16951$ | $13.1298 \pm 3.34997$ | $0.793124 \pm 0.00576659$ | $0.95156 \pm 0.00215277$ |

Table A14: Miscoverage rate $\alpha = 0.2$: Results on the Cells-Tails dataset.

| Method | val MAE (↓) | val Interval Length (↓) | test Coverage (≥ 0.80) | test Prediction Rate (↑) |
|---|---|---|---|---|
| Conformal Prediction | $3.56733 \pm 1.0818$ | $11.3841 \pm 3.54242$ | $0.48332 \pm 0.0337429$ | 1.0 |
| Gaussian | $4.05446 \pm 1.33153$ | $12.7086 \pm 4.39552$ | $0.47106 \pm 0.0319074$ | 1.0 |
| Gaussian Ensemble | $2.40691 \pm 0.580524$ | $8.62187 \pm 1.5177$ | $0.4768 \pm 0.0461347$ | 1.0 |
| Quantile Regression | $4.20398 \pm 1.61554$ | $13.1241 \pm 4.61204$ | $0.49358 \pm 0.0341114$ | 1.0 |
| Gaussian + Selective GMM | $4.05446 \pm 1.33153$ | $12.7086 \pm 4.39552$ | $0.788228 \pm 0.0239896$ | $0.5364 \pm 0.0100584$ |
| Gaussian Ensemble + Selective Ens Var | $2.40691 \pm 0.580524$ | $8.62187 \pm 1.5177$ | $0.567734 \pm 0.0270648$ | $0.74512 \pm 0.0438128$ |

Table A15: Miscoverage rate $\alpha = 0.2$: Results on the CHAIRANGLE dataset.

| Method | val MAE (↓) | val Interval Length (↓) | test Coverage (≥ 0.80) | test Prediction Rate (↑) |
|---|---|---|---|---|
| Conformal Prediction | 0.289127 ± 0.081643 | 0.872808 ± 0.205426 | 0.80212 ± 0.00337798 | 1.0 |
| Gaussian | 0.376692 ± 0.171928 | 1.14193 ± 0.40832 | 0.806646 ± 0.00565869 | 1.0 |
| Gaussian Ensemble | 0.361834 ± 0.165348 | 1.01037 ± 0.399644 | 0.813452 ± 0.00329742 | 1.0 |
| Quantile Regression | 0.648062 ± 0.181932 | 1.87322 ± 0.428951 | 0.807127 ± 0.00380433 | 1.0 |
| Gaussian + Selective GMM | 0.376692 ± 0.171928 | 1.14193 ± 0.40832 | 0.807801 ± 0.0066405 | 0.972829 ± 0.0019256 |
| Gaussian Ensemble + Selective Ens Var | 0.361834 ± 0.165348 | 1.01037 ± 0.399644 | 0.804719 ± 0.00318856 | 0.951359 ± 0.00453556 |

Table A16: Miscoverage rate $\alpha = 0.2$: Results on the CHAIRANGLE-GAP dataset.

| Method | val MAE (↓) | val Interval Length (↓) | test Coverage (≥ 0.80) | test Prediction Rate (↑) |
|---|---|---|---|---|
| Conformal Prediction | 0.35034 ± 0.161819 | 1.09877 ± 0.497591 | 0.588472 ± 0.0119766 | 1.0 |
| Gaussian | 0.454516 ± 0.280174 | 1.69802 ± 0.840083 | 0.613951 ± 0.0358797 | 1.0 |
| Gaussian Ensemble | 0.226352 ± 0.0677413 | 0.972076 ± 0.118477 | 0.6431 ± 0.0136145 | 1.0 |
| Quantile Regression | 0.692597 ± 0.269476 | 2.29958 ± 0.738934 | 0.600606 ± 0.0136385 | 1.0 |
| Gaussian + Selective GMM | 0.454516 ± 0.280174 | 1.69802 ± 0.840083 | 0.813442 ± 0.00720691 | 0.650227 ± 0.00138861 |
| Gaussian Ensemble + Selective Ens Var | 0.226352 ± 0.0677413 | 0.972076 ± 0.118477 | 0.701301 ± 0.0146886 | 0.7711 ± 0.0191172 |

Table A17: Miscoverage rate $\alpha = 0.2$: Results on the VENTRICULARVOLUME dataset.

| Method | val MAE (↓) | val Interval Length (↓) | test Coverage (≥ 0.80) | test Prediction Rate (↑) |
|---|---|---|---|---|
| Conformal Prediction | 11.2471 ± 0.201399 | 33.0996 ± 0.991594 | 0.457053 ± 0.00583106 | 1.0 |
| Gaussian | 12.7238 ± 1.52197 | 37.3172 ± 4.91536 | 0.589342 ± 0.0793156 | 1.0 |
| Gaussian Ensemble | 10.1141 ± 0.180661 | 29.9914 ± 1.12007 | 0.678527 ± 0.0277009 | 1.0 |
| Quantile Regression | 12.5465 ± 0.941364 | 35.8073 ± 1.5139 | 0.537774 ± 0.0444887 | 1.0 |
| Gaussian + Selective GMM | 12.7238 ± 1.52197 | 37.3172 ± 4.91536 | 0.609177 ± 0.0682058 | 0.709875 ± 0.0232873 |
| Gaussian Ensemble + Selective Ens Var | 10.1141 ± 0.180661 | 29.9914 ± 1.12007 | 0.640078 ± 0.0277429 | 0.686207 ± 0.0431835 |

Table A18: Miscoverage rate $\alpha = 0.2$: Results on the BRAINTUMOURPIXELS dataset.

| Method | val MAE (↓) | val Interval Length (↓) | test Coverage (≥ 0.80) | test Prediction Rate (↑) |
|---|---|---|---|---|
| Conformal Prediction | 21.3163 ± 0.45997 | 66.146 ± 2.75637 | 0.824632 ± 0.0128289 | 1.0 |
| Gaussian | 21.0625 ± 0.358012 | 64.2423 ± 1.9975 | 0.798113 ± 0.0133438 | 1.0 |
| Gaussian Ensemble | 20.5336 ± 0.211421 | 62.5706 ± 0.713494 | 0.796961 ± 0.00658989 | 1.0 |
| Quantile Regression | 21.9545 ± 0.149787 | 66.2654 ± 1.33091 | 0.807646 ± 0.0165337 | 1.0 |
| Gaussian + Selective GMM | 21.0625 ± 0.358012 | 64.2423 ± 1.9975 | 0.809477 ± 0.0139729 | 0.973576 ± 0.0187252 |
| Gaussian Ensemble + Selective Ens Var | 20.5336 ± 0.211421 | 62.5706 ± 0.713494 | 0.806998 ± 0.00641546 | 0.975368 ± 0.00323397 |

Table A19: Miscoverage rate $\alpha = 0.2$: Results on the SKINLESIONPIXELS dataset.

| Method | val MAE (↓) | val Interval Length (↓) | test Coverage (≥ 0.80) | test Prediction Rate (↑) |
|---|---|---|---|---|
| Conformal Prediction | 107.514 ± 1.87464 | 275.494 ± 5.60123 | 0.553254 ± 0.0122434 | 1.0 |
| Gaussian | 105.417 ± 1.15178 | 395.661 ± 174.897 | 0.689951 ± 0.0406148 | 1.0 |
| Gaussian Ensemble | 100.639 ± 0.464183 | 351.961 ± 70.4308 | 0.715272 ± 0.0278757 | 1.0 |
| Quantile Regression | 106.382 ± 2.70593 | 251.346 ± 8.67954 | 0.590792 ± 0.0115523 | 1.0 |
| Gaussian + Selective GMM | 105.417 ± 1.15178 | 395.661 ± 174.897 | 0.708767 ± 0.0274691 | 0.849668 ± 0.0189464 |
| Gaussian Ensemble + Selective Ens Var | 100.639 ± 0.464183 | 351.961 ± 70.4308 | 0.704273 ± 0.0198608 | 0.782116 ± 0.0109776 |

Table A20: Miscoverage rate $\alpha = 0.2$: Results on the HISTOLOGYNUCLEIPIXELS dataset.

| Method | val MAE (↓) | val Interval Length (↓) | test Coverage (≥ 0.80) | test Prediction Rate (↑) |
|---|---|---|---|---|
| Conformal Prediction | 217.887 ± 3.71766 | 688.269 ± 13.6761 | 0.744155 ± 0.0107073 | 1.0 |
| Gaussian | 211.795 ± 10.0239 | 902.884 ± 295.983 | 0.454345 ± 0.0986975 | 1.0 |
| Gaussian Ensemble | 196.785 ± 2.14454 | 840.287 ± 113.005 | 0.412528 ± 0.0544736 | 1.0 |
| Quantile Regression | 221.136 ± 9.7918 | 679.117 ± 30.6895 | 0.580503 ± 0.0463751 | 1.0 |
| Gaussian + Selective GMM | 211.795 ± 10.0239 | 902.884 ± 295.983 | 0.462248 ± 0.105935 | 0.901985 ± 0.0400294 |
| Gaussian Ensemble + Selective Ens Var | 196.785 ± 2.14454 | 840.287 ± 113.005 | 0.417977 ± 0.0572495 | 0.954477 ± 0.0256246 |

Table A21: Miscoverage rate $\alpha = 0.2$: Results on the AERIALBUILDINGPIXELS dataset.

| Method | val MAE ($\downarrow$) | val Interval Length ($\downarrow$) | test Coverage ($\geq 0.80$) | test Prediction Rate ($\uparrow$) |
|---|---|---|---|---|
| Conformal Prediction | $235.417 \pm 7.16096$ | $685.829 \pm 31.3669$ | $0.52437 \pm 0.0610316$ | $1.0$ |
| Gaussian | $217.877 \pm 1.72493$ | $685.899 \pm 43.1749$ | $0.60129 \pm 0.0717427$ | $1.0$ |
| Gaussian Ensemble | $208.487 \pm 1.03581$ | $658.243 \pm 27.1206$ | $0.683599 \pm 0.0674139$ | $1.0$ |
| Quantile Regression | $229.243 \pm 6.21038$ | $662.104 \pm 18.5495$ | $0.635733 \pm 0.0350343$ | $1.0$ |
| Gaussian + Selective GMM | $217.877 \pm 1.72493$ | $685.899 \pm 43.1749$ | $0.679246 \pm 0.0462652$ | $0.65018 \pm 0.0999173$ |
| Gaussian Ensemble + Selective Ens Var | $208.487 \pm 1.03581$ | $658.243 \pm 27.1206$ | $0.743714 \pm 0.0354023$ | $0.571928 \pm 0.0402727$ |

Table A22: Miscoverage rate $\alpha = 0.05$: Results on the CELLS dataset.

| Method | val MAE ($\downarrow$) | val Interval Length ($\downarrow$) | test Coverage ($\geq 0.95$) | test Prediction Rate ($\uparrow$) |
|---|---|---|---|---|
| Conformal Prediction | $4.03121 \pm 2.78375$ | $22.3973 \pm 12.7194$ | $0.95612 \pm 0.00460235$ | $1.0$ |
| Gaussian | $3.61704 \pm 1.13624$ | $17.0459 \pm 4.98381$ | $0.95402 \pm 0.00275928$ | $1.0$ |
| Gaussian Ensemble | $2.78757 \pm 1.16951$ | $20.5445 \pm 5.00176$ | $0.9484 \pm 0.00331843$ | $1.0$ |
| Quantile Regression | $3.5478 \pm 1.25834$ | $17.8244 \pm 3.57608$ | $0.94858 \pm 0.00565346$ | $1.0$ |
| Gaussian + Selective GMM | $3.61704 \pm 1.13624$ | $17.0459 \pm 4.98381$ | $0.955558 \pm 0.0032579$ | $0.95212 \pm 0.00276362$ |
| Gaussian Ensemble + Selective Ens Var | $2.78757 \pm 1.16951$ | $20.5445 \pm 5.00176$ | $0.946085 \pm 0.00337562$ | $0.95156 \pm 0.00215277$ |

Table A23: Miscoverage rate $\alpha = 0.05$: Results on the CELLS-TAILS dataset.

| Method | val MAE ($\downarrow$) | val Interval Length ($\downarrow$) | test Coverage ($\geq 0.95$) | test Prediction Rate ($\uparrow$) |
|---|---|---|---|---|
| Conformal Prediction | $3.56733 \pm 1.0818$ | $16.3831 \pm 4.80611$ | $0.59048 \pm 0.0307226$ | $1.0$ |
| Gaussian | $4.05446 \pm 1.33153$ | $17.5443 \pm 5.3978$ | $0.59066 \pm 0.0417946$ | $1.0$ |
| Gaussian Ensemble | $2.40691 \pm 0.580524$ | $13.1205 \pm 1.81604$ | $0.67218 \pm 0.0308246$ | $1.0$ |
| Quantile Regression | $3.34375 \pm 0.642763$ | $14.5539 \pm 2.01592$ | $0.5666 \pm 0.0295055$ | $1.0$ |
| Gaussian + Selective GMM | $4.05446 \pm 1.33153$ | $17.5443 \pm 5.3978$ | $0.941551 \pm 0.0164539$ | $0.53636 \pm 0.0100057$ |
| Gaussian Ensemble + Selective Ens Var | $2.40691 \pm 0.580524$ | $13.1205 \pm 1.81604$ | $0.720221 \pm 0.0374625$ | $0.74512 \pm 0.0438128$ |

Table A24: Miscoverage rate $\alpha = 0.05$: Results on the CHAIRANGLE dataset.

| Method | val MAE ($\downarrow$) | val Interval Length ($\downarrow$) | test Coverage ($\geq 0.95$) | test Prediction Rate ($\uparrow$) |
|---|---|---|---|---|
| Conformal Prediction | $0.289127 \pm 0.081643$ | $1.27437 \pm 0.18759$ | $0.953087 \pm 0.000729425$ | $1.0$ |
| Gaussian | $0.376692 \pm 0.171928$ | $1.59112 \pm 0.367248$ | $0.949381 \pm 0.00143361$ | $1.0$ |
| Gaussian Ensemble | $0.361834 \pm 0.165348$ | $1.38384 \pm 0.481345$ | $0.95633 \pm 0.0016841$ | $1.0$ |
| Quantile Regression | $0.888707 \pm 0.472522$ | $4.04185 \pm 1.50356$ | $0.951608 \pm 0.00184683$ | $1.0$ |
| Gaussian + Selective GMM | $0.376692 \pm 0.171928$ | $1.59112 \pm 0.367248$ | $0.949963 \pm 0.00265095$ | $0.97208 \pm 0.00166628$ |
| Gaussian Ensemble + Selective Ens Var | $0.361834 \pm 0.165348$ | $1.38384 \pm 0.481345$ | $0.9541 \pm 0.00167267$ | $0.951359 \pm 0.00453556$ |

Table A25: Miscoverage rate $\alpha = 0.05$: Results on the CHAIRANGLE-GAP dataset.

| Method | val MAE ($\downarrow$) | val Interval Length ($\downarrow$) | test Coverage ($\geq 0.95$) | test Prediction Rate ($\uparrow$) |
|---|---|---|---|---|
| Conformal Prediction | $0.35034 \pm 0.161819$ | $1.56264 \pm 0.582954$ | $0.697817 \pm 0.00918537$ | $1.0$ |
| Gaussian | $0.454516 \pm 0.280174$ | $2.39892 \pm 1.02472$ | $0.733595 \pm 0.0367327$ | $1.0$ |
| Gaussian Ensemble | $0.226352 \pm 0.0677413$ | $1.59703 \pm 0.159795$ | $0.780615 \pm 0.0141415$ | $1.0$ |
| Quantile Regression | $1.43807 \pm 0.99796$ | $5.57113 \pm 1.71783$ | $0.759929 \pm 0.034465$ | $1.0$ |
| Gaussian + Selective GMM | $0.454516 \pm 0.280174$ | $2.39892 \pm 1.02472$ | $0.95774 \pm 0.00281998$ | $0.64882 \pm 0.00276083$ |
| Gaussian Ensemble + Selective Ens Var | $0.226352 \pm 0.0677413$ | $1.59703 \pm 0.159795$ | $0.844587 \pm 0.0165508$ | $0.7711 \pm 0.0191172$ |

Table A26: Miscoverage rate $\alpha = 0.05$: Results on the VENTRICULARVOLUME dataset.

| Method | val MAE ($\downarrow$) | val Interval Length ($\downarrow$) | test Coverage ($\geq 0.95$) | test Prediction Rate ($\uparrow$) |
|---|---|---|---|---|
| Conformal Prediction | $11.2471 \pm 0.201399$ | $70.211 \pm 1.37665$ | $0.776176 \pm 0.00853621$ | $1.0$ |
| Gaussian | $12.7238 \pm 1.52197$ | $67.1562 \pm 6.31002$ | $0.837931 \pm 0.0396691$ | $1.0$ |
| Gaussian Ensemble | $10.1141 \pm 0.180661$ | $51.9065 \pm 3.94789$ | $0.876646 \pm 0.0140944$ | $1.0$ |
| Quantile Regression | $12.0793 \pm 0.130032$ | $61.623 \pm 3.04835$ | $0.843574 \pm 0.0243659$ | $1.0$ |
| Gaussian + Selective GMM | $12.7238 \pm 1.52197$ | $67.1562 \pm 6.31002$ | $0.854302 \pm 0.0279176$ | $0.710188 \pm 0.0225912$ |
| Gaussian Ensemble + Selective Ens Var | $10.1141 \pm 0.180661$ | $51.9065 \pm 3.94789$ | $0.855053 \pm 0.0125036$ | $0.686207 \pm 0.0431835$ |

Table A27: Miscoverage rate $\alpha = 0.05$: Results on the BRAINTUMOURPIXELS dataset.

| Method | val MAE ($\downarrow$) | val Interval Length ($\downarrow$) | test Coverage ($\geq 0.95$) | test Prediction Rate ($\uparrow$) |
|---|---|---|---|---|
| Conformal Prediction | $21.3163 \pm 0.45997$ | $123.719 \pm 2.07495$ | $0.921113 \pm 0.00123648$ | $1.0$ |
| Gaussian | $21.0625 \pm 0.358012$ | $123.587 \pm 3.3168$ | $0.910141 \pm 0.00537523$ | $1.0$ |
| Gaussian Ensemble | $20.5336 \pm 0.211421$ | $113.592 \pm 1.37398$ | $0.905182 \pm 0.00359141$ | $1.0$ |
| Quantile Regression | $24.6897 \pm 2.49388$ | $126.788 \pm 7.00165$ | $0.915995 \pm 0.00956349$ | $1.0$ |
| Gaussian + Selective GMM | $21.0625 \pm 0.358012$ | $123.587 \pm 3.3168$ | $0.919455 \pm 0.0105219$ | $0.973672 \pm 0.0186442$ |
| Gaussian Ensemble + Selective Ens Var | $20.5336 \pm 0.211421$ | $113.592 \pm 1.37398$ | $0.910995 \pm 0.00267557$ | $0.975368 \pm 0.00323397$ |

Table A28: Miscoverage rate $\alpha = 0.05$: Results on the SKINLESIONPIXELS dataset.

| Method | val MAE ($\downarrow$) | val Interval Length ($\downarrow$) | test Coverage ($\geq 0.95$) | test Prediction Rate ($\uparrow$) |
|---|---|---|---|---|
| Conformal Prediction | $107.514 \pm 1.87464$ | $845.109 \pm 31.2572$ | $0.822753 \pm 0.0104666$ | $1.0$ |
| Gaussian | $105.417 \pm 1.15178$ | $691.768 \pm 232.071$ | $0.866844 \pm 0.0166304$ | $1.0$ |
| Gaussian Ensemble | $100.639 \pm 0.464183$ | $582.719 \pm 95.3488$ | $0.881009 \pm 0.00686587$ | $1.0$ |
| Quantile Regression | $112.07 \pm 5.13986$ | $627.938 \pm 68.3371$ | $0.827977 \pm 0.017595$ | $1.0$ |
| Gaussian + Selective GMM | $105.417 \pm 1.15178$ | $691.768 \pm 232.071$ | $0.889891 \pm 0.00940058$ | $0.849225 \pm 0.0194809$ |
| Gaussian Ensemble + Selective Ens Var | $100.639 \pm 0.464183$ | $582.719 \pm 95.3488$ | $0.881749 \pm 0.00534917$ | $0.782116 \pm 0.0109776$ |

Table A29: Miscoverage rate $\alpha = 0.05$: Results on the HISTOLOGYNUCLEIPIXELS dataset.

| Method | val MAE ($\downarrow$) | val Interval Length ($\downarrow$) | test Coverage ($\geq 0.95$) | test Prediction Rate ($\uparrow$) |
|---|---|---|---|---|
| Conformal Prediction | $217.887 \pm 3.71766$ | $1282.78 \pm 25.9915$ | $0.896603 \pm 0.0113207$ | $1.0$ |
| Gaussian | $211.795 \pm 10.0239$ | $1491.62 \pm 472.378$ | $0.680018 \pm 0.0860417$ | $1.0$ |
| Gaussian Ensemble | $196.785 \pm 2.14454$ | $1358.12 \pm 174.189$ | $0.66749 \pm 0.0522544$ | $1.0$ |
| Quantile Regression | $251.562 \pm 12.244$ | $1191.96 \pm 65.3122$ | $0.754566 \pm 0.0379023$ | $1.0$ |
| Gaussian + Selective GMM | $211.795 \pm 10.0239$ | $1491.62 \pm 472.378$ | $0.68897 \pm 0.0895173$ | $0.901191 \pm 0.0405698$ |
| Gaussian Ensemble + Selective Ens Var | $196.785 \pm 2.14454$ | $1358.12 \pm 174.189$ | $0.673945 \pm 0.0533738$ | $0.954477 \pm 0.0256246$ |

Table A30: Miscoverage rate $\alpha = 0.05$: Results on the AERIALBUILDINGPIXELS dataset.

| Method | val MAE ($\downarrow$) | val Interval Length ($\downarrow$) | test Coverage ($\geq 0.95$) | test Prediction Rate ($\uparrow$) |
|---|---|---|---|---|
| Conformal Prediction | $235.417 \pm 7.16096$ | $1474.93 \pm 53.6301$ | $0.731414 \pm 0.070291$ | $1.0$ |
| Gaussian | $217.877 \pm 1.72493$ | $1181.55 \pm 41.5859$ | $0.773522 \pm 0.0548329$ | $1.0$ |
| Gaussian Ensemble | $208.487 \pm 1.03581$ | $1108.87 \pm 21.3551$ | $0.88874 \pm 0.0490213$ | $1.0$ |
| Quantile Regression | $294.181 \pm 19.9268$ | $1281.37 \pm 51.1001$ | $0.888175 \pm 0.028968$ | $1.0$ |
| Gaussian + Selective GMM | $217.877 \pm 1.72493$ | $1181.55 \pm 41.5859$ | $0.831501 \pm 0.0374173$ | $0.650797 \pm 0.0987431$ |
| Gaussian Ensemble + Selective Ens Var | $208.487 \pm 1.03581$ | $1108.87 \pm 21.3551$ | $0.899838 \pm 0.0263409$ | $0.571928 \pm 0.0402727$ |

Table A31: Method variation results on the CELLS dataset.

| Method | val MAE ($\downarrow$) | val Interval Length ($\downarrow$) | test Coverage ($\geq 0.90$) | test Prediction Rate ($\uparrow$) |
|---|---|---|---|---|
| Gaussian + Selective GMM | $3.61704 \pm 1.13624$ | $14.5492 \pm 4.44927$ | $0.905241 \pm 0.00448315$ | $0.95216 \pm 0.00290558$ |
| Gaussian + Selective GMM, $k = 2$ | $3.61704 \pm 1.13624$ | $14.5492 \pm 4.44927$ | $0.9044 \pm 0.00481955$ | $0.952 \pm 0.00270259$ |
| Gaussian + Selective GMM, $k = 8$ | $3.61704 \pm 1.13624$ | $14.5492 \pm 4.44927$ | $0.904928 \pm 0.00469093$ | $0.9528 \pm 0.00331662$ |
| Gaussian + Selective GMM, Spectral Norm | $4.80289 \pm 1.89759$ | $18.5596 \pm 9.13391$ | $0.908 \pm 0.00703097$ | $0.95118 \pm 0.00220672$ |
| Gaussian + Selective kNN | $3.61704 \pm 1.13624$ | $14.5492 \pm 4.44927$ | $0.900069 \pm 0.00677637$ | $0.94656 \pm 0.00470472$ |
| Gaussian + Selective kNN, $k = 5$ | $3.61704 \pm 1.13624$ | $14.5492 \pm 4.44927$ | $0.900146 \pm 0.00671325$ | $0.94736 \pm 0.00395049$ |
| Gaussian + Selective kNN, $k = 20$ | $3.61704 \pm 1.13624$ | $14.5492 \pm 4.44927$ | $0.900048 \pm 0.00691667$ | $0.94624 \pm 0.00248564$ |
| Gaussian + Selective kNN, L2 | $3.61704 \pm 1.13624$ | $14.5492 \pm 4.44927$ | $0.906295 \pm 0.00579854$ | $0.95166 \pm 0.00422734$ |

Table A32: Method variation results on the CELLS-TAILS dataset.

| Method | val MAE ($\downarrow$) | val Interval Length ($\downarrow$) | test Coverage ($\geq 0.90$) | test Prediction Rate ($\uparrow$) |
|---|---|---|---|---|
| Gaussian + Selective GMM | $4.05446 \pm 1.33153$ | $15.4321 \pm 4.98796$ | $0.889825 \pm 0.0193021$ | $0.53654 \pm 0.0101012$ |
| Gaussian + Selective GMM, $k = 2$ | $4.05446 \pm 1.33153$ | $15.4321 \pm 4.98796$ | $0.891933 \pm 0.0175265$ | $0.525 \pm 0.00829409$ |
| Gaussian + Selective GMM, $k = 8$ | $4.05446 \pm 1.33153$ | $15.4321 \pm 4.98796$ | $0.893648 \pm 0.0182098$ | $0.53516 \pm 0.00975635$ |
| Gaussian + Selective GMM, Spectral Norm | $4.91114 \pm 2.46961$ | $17.3945 \pm 7.3177$ | $0.881351 \pm 0.021587$ | $0.5331 \pm 0.0157885$ |
| Gaussian + Selective kNN | $4.05446 \pm 1.33153$ | $15.4321 \pm 4.98796$ | $0.859179 \pm 0.0255173$ | $0.56692 \pm 0.0127107$ |
| Gaussian + Selective kNN, $k = 5$ | $4.05446 \pm 1.33153$ | $15.4321 \pm 4.98796$ | $0.855387 \pm 0.0261636$ | $0.57206 \pm 0.0129226$ |
| Gaussian + Selective kNN, $k = 20$ | $4.05446 \pm 1.33153$ | $15.4321 \pm 4.98796$ | $0.862429 \pm 0.0264602$ | $0.56236 \pm 0.0122017$ |
| Gaussian + Selective kNN, L2 | $4.05446 \pm 1.33153$ | $15.4321 \pm 4.98796$ | $0.898932 \pm 0.00921808$ | $0.51486 \pm 0.00474662$ |

Table A33: Method variation results on the CELLS-GAP dataset.

| Method | val MAE (↓) | val Interval Length (↓) | test Coverage (≥ 0.90) | test Prediction Rate (↑) |
|---|---|---|---|---|
| Gaussian + Selective GMM | $3.53089 \pm 1.0619$ | $15.6396 \pm 6.23458$ | $0.890569 \pm 0.00953089$ | $0.49372 \pm 0.0025926$ |
| Gaussian + Selective GMM, $k = 2$ | $3.53089 \pm 1.0619$ | $15.6396 \pm 6.23458$ | $0.892265 \pm 0.00867262$ | $0.48902 \pm 0.00420828$ |
| Gaussian + Selective GMM, $k = 8$ | $3.53089 \pm 1.0619$ | $15.6396 \pm 6.23458$ | $0.88967 \pm 0.0111239$ | $0.50196 \pm 0.00492041$ |
| Gaussian + Selective GMM, Spectral Norm | $2.81679 \pm 0.42988$ | $12.194 \pm 1.10907$ | $0.883644 \pm 0.0167566$ | $0.4991 \pm 0.00761236$ |
| Gaussian + Selective kNN | $3.53089 \pm 1.0619$ | $15.6396 \pm 6.23458$ | $0.874032 \pm 0.0432364$ | $0.53646 \pm 0.0139864$ |
| Gaussian + Selective kNN, $k = 5$ | $3.53089 \pm 1.0619$ | $15.6396 \pm 6.23458$ | $0.874822 \pm 0.0424219$ | $0.5368 \pm 0.0137332$ |
| Gaussian + Selective kNN, $k = 20$ | $3.53089 \pm 1.0619$ | $15.6396 \pm 6.23458$ | $0.873876 \pm 0.0443446$ | $0.536 \pm 0.014724$ |
| Gaussian + Selective kNN, L2 | $3.53089 \pm 1.0619$ | $15.6396 \pm 6.23458$ | $0.887052 \pm 0.0144573$ | $0.49576 \pm 0.00474578$ |

Table A34: Method variation results on the CHAIRANGLE dataset.

| Method | val MAE (↓) | val Interval Length (↓) | test Coverage (≥ 0.90) | test Prediction Rate (↑) |
|---|---|---|---|---|
| Gaussian + Selective GMM | $0.376692 \pm 0.171928$ | $1.37757 \pm 0.382191$ | $0.902482 \pm 0.00436054$ | $0.972739 \pm 0.00191733$ |
| Gaussian + Selective GMM, $k = 2$ | $0.376692 \pm 0.171928$ | $1.37757 \pm 0.382191$ | $0.902972 \pm 0.00540708$ | $0.96392 \pm 0.00121759$ |
| Gaussian + Selective GMM, $k = 8$ | $0.376692 \pm 0.171928$ | $1.37757 \pm 0.382191$ | $0.901822 \pm 0.00328732$ | $0.981238 \pm 0.00113892$ |
| Gaussian + Selective GMM, Spectral Norm | $0.516487 \pm 0.227015$ | $1.82929 \pm 0.809339$ | $0.903828 \pm 0.00534803$ | $0.970993 \pm 0.000466666$ |
| Gaussian + Selective kNN | $0.376692 \pm 0.171928$ | $1.37757 \pm 0.382191$ | $0.90222 \pm 0.00459694$ | $0.975465 \pm 0.00201785$ |
| Gaussian + Selective kNN, $k = 5$ | $0.376692 \pm 0.171928$ | $1.37757 \pm 0.382191$ | $0.902561 \pm 0.00342563$ | $0.983804 \pm 0.00152586$ |
| Gaussian + Selective kNN, $k = 20$ | $0.376692 \pm 0.171928$ | $1.37757 \pm 0.382191$ | $0.902137 \pm 0.00457917$ | $0.966592 \pm 0.00129222$ |
| Gaussian + Selective kNN, L2 | $0.376692 \pm 0.171928$ | $1.37757 \pm 0.382191$ | $0.903183 \pm 0.00504343$ | $0.977247 \pm 0.00149241$ |

Table A35: Method variation results on the CHAIRANGLE-TAILS dataset.

| Method | val MAE (↓) | val Interval Length (↓) | test Coverage (≥ 0.90) | test Prediction Rate (↑) |
|---|---|---|---|---|
| Gaussian + Selective GMM | $0.241214 \pm 0.091736$ | $1.09417 \pm 0.382907$ | $0.901946 \pm 0.00382993$ | $0.655448 \pm 0.001058$ |
| Gaussian + Selective GMM, $k = 2$ | $0.241214 \pm 0.091736$ | $1.09417 \pm 0.382907$ | $0.903353 \pm 0.00471599$ | $0.645452 \pm 0.00155062$ |
| Gaussian + Selective GMM, $k = 8$ | $0.241214 \pm 0.091736$ | $1.09417 \pm 0.382907$ | $0.902103 \pm 0.00382668$ | $0.660472 \pm 0.00115112$ |
| Gaussian + Selective GMM, Spectral Norm | $0.375118 \pm 0.24509$ | $1.24343 \pm 0.611561$ | $0.910147 \pm 0.00315697$ | $0.65518 \pm 0.00176257$ |
| Gaussian + Selective kNN | $0.241214 \pm 0.091736$ | $1.09417 \pm 0.382907$ | $0.860311 \pm 0.00617031$ | $0.703038 \pm 0.00691227$ |
| Gaussian + Selective kNN, $k = 5$ | $0.241214 \pm 0.091736$ | $1.09417 \pm 0.382907$ | $0.88625 \pm 0.00498864$ | $0.683742 \pm 0.00547083$ |
| Gaussian + Selective kNN, $k = 20$ | $0.241214 \pm 0.091736$ | $1.09417 \pm 0.382907$ | $0.809551 \pm 0.00764886$ | $0.743287 \pm 0.0107676$ |
| Gaussian + Selective kNN, L2 | $0.241214 \pm 0.091736$ | $1.09417 \pm 0.382907$ | $0.89682 \pm 0.00698641$ | $0.662076 \pm 0.00289246$ |

Table A36: Method variation results on the CHAIRANGLE-GAP dataset.

| Method | val MAE (↓) | val Interval Length (↓) | test Coverage (≥ 0.90) | test Prediction Rate (↑) |
|---|---|---|---|---|
| Gaussian + Selective GMM | $0.454516 \pm 0.280174$ | $2.09212 \pm 0.933756$ | $0.91215 \pm 0.00604745$ | $0.649372 \pm 0.00334919$ |
| Gaussian + Selective GMM, $k = 2$ | $0.454516 \pm 0.280174$ | $2.09212 \pm 0.933756$ | $0.912708 \pm 0.00541793$ | $0.642138 \pm 0.00143869$ |
| Gaussian + Selective GMM, $k = 8$ | $0.454516 \pm 0.280174$ | $2.09212 \pm 0.933756$ | $0.911374 \pm 0.00475979$ | $0.654824 \pm 0.00368979$ |
| Gaussian + Selective GMM, Spectral Norm | $0.294511 \pm 0.0861797$ | $1.19904 \pm 0.193646$ | $0.910836 \pm 0.00442911$ | $0.649996 \pm 0.00233428$ |
| Gaussian + Selective kNN | $0.454516 \pm 0.280174$ | $2.09212 \pm 0.933756$ | $0.911574 \pm 0.00376608$ | $0.673764 \pm 0.0113432$ |
| Gaussian + Selective kNN, $k = 5$ | $0.454516 \pm 0.280174$ | $2.09212 \pm 0.933756$ | $0.91224 \pm 0.00426519$ | $0.670646 \pm 0.00691764$ |
| Gaussian + Selective kNN, $k = 20$ | $0.454516 \pm 0.280174$ | $2.09212 \pm 0.933756$ | $0.907543 \pm 0.00204724$ | $0.680107 \pm 0.0162927$ |
| Gaussian + Selective kNN, L2 | $0.454516 \pm 0.280174$ | $2.09212 \pm 0.933756$ | $0.913383 \pm 0.00539352$ | $0.658744 \pm 0.00274677$ |

Table A37: Method variation results on the ASSETWEALTH dataset.

| Method | val MAE (↓) | val Interval Length (↓) | test Coverage (≥ 0.90) | test Prediction Rate (↑) |
|---|---|---|---|---|
| Gaussian + Selective GMM | $0.367501 \pm 0.0416437$ | $1.597 \pm 0.207599$ | $0.850824 \pm 0.047533$ | $0.93838 \pm 0.0170443$ |
| Gaussian + Selective GMM, $k = 2$ | $0.367501 \pm 0.0416437$ | $1.597 \pm 0.207599$ | $0.850232 \pm 0.0469534$ | $0.933687 \pm 0.0176348$ |
| Gaussian + Selective GMM, $k = 8$ | $0.367501 \pm 0.0416437$ | $1.597 \pm 0.207599$ | $0.85067 \pm 0.0456954$ | $0.932879 \pm 0.0102523$ |
| Gaussian + Selective GMM, Spectral Norm | $0.354647 \pm 0.00973911$ | $1.55594 \pm 0.0421589$ | $0.872742 \pm 0.0250168$ | $0.941812 \pm 0.0101895$ |
| Gaussian + Selective kNN | $0.367501 \pm 0.0416437$ | $1.597 \pm 0.207599$ | $0.852107 \pm 0.0474734$ | $0.933586 \pm 0.0203541$ |
| Gaussian + Selective kNN, $k = 5$ | $0.367501 \pm 0.0416437$ | $1.597 \pm 0.207599$ | $0.85256 \pm 0.0469571$ | $0.933232 \pm 0.02149$ |
| Gaussian + Selective kNN, $k = 20$ | $0.367501 \pm 0.0416437$ | $1.597 \pm 0.207599$ | $0.851914 \pm 0.0460655$ | $0.933636 \pm 0.0200087$ |
| Gaussian + Selective kNN, L2 | $0.367501 \pm 0.0416437$ | $1.597 \pm 0.207599$ | $0.853365 \pm 0.0410864$ | $0.897149 \pm 0.0162693$ |

Table A38: Method variation results on the VENTRICULARVOLUME dataset.

| Method | val MAE (↓) | val Interval Length (↓) | test Coverage (≥ 0.90) | test Prediction Rate (↑) |
|---|---|---|---|---|
| Gaussian + Selective GMM | $12.7238 \pm 1.52197$ | $51.566 \pm 3.52739$ | $0.752046 \pm 0.0529087$ | $0.707994 \pm 0.0208741$ |
| Gaussian + Selective GMM, $k = 2$ | $12.7238 \pm 1.52197$ | $51.566 \pm 3.52739$ | $0.745271 \pm 0.0582068$ | $0.790752 \pm 0.0307666$ |
| Gaussian + Selective GMM, $k = 8$ | $12.7238 \pm 1.52197$ | $51.566 \pm 3.52739$ | $0.747625 \pm 0.0584117$ | $0.656583 \pm 0.0258853$ |
| Gaussian + Selective GMM, Spectral Norm | $11.6311 \pm 0.483357$ | $45.5475 \pm 0.747647$ | $0.734907 \pm 0.00853342$ | $0.719279 \pm 0.0147984$ |
| Gaussian + Selective kNN | $12.7238 \pm 1.52197$ | $51.566 \pm 3.52739$ | $0.735105 \pm 0.0612413$ | $0.911599 \pm 0.0447475$ |
| Gaussian + Selective kNN, $k = 5$ | $12.7238 \pm 1.52197$ | $51.566 \pm 3.52739$ | $0.735857 \pm 0.0604531$ | $0.916614 \pm 0.0419073$ |
| Gaussian + Selective kNN, $k = 20$ | $12.7238 \pm 1.52197$ | $51.566 \pm 3.52739$ | $0.734752 \pm 0.0616982$ | $0.909404 \pm 0.0469634$ |
| Gaussian + Selective kNN, L2 | $12.7238 \pm 1.52197$ | $51.566 \pm 3.52739$ | $0.740812 \pm 0.049802$ | $0.740439 \pm 0.0222073$ |

Table A39: Method variation results on the BRAINTUMOURPIXELS dataset.

| Method | val MAE (↓) | val Interval Length (↓) | test Coverage (≥ 0.90) | test Prediction Rate (↑) |
|---|---|---|---|---|
| Gaussian + Selective GMM | $21.0625 \pm 0.358012$ | $93.6284 \pm 2.29916$ | $0.883515 \pm 0.0138279$ | $0.973576 \pm 0.0187252$ |
| Gaussian + Selective GMM, $k = 2$ | $21.0625 \pm 0.358012$ | $93.6284 \pm 2.29916$ | $0.881874 \pm 0.0118346$ | $0.977863 \pm 0.0145103$ |
| Gaussian + Selective GMM, $k = 8$ | $21.0625 \pm 0.358012$ | $93.6284 \pm 2.29916$ | $0.880823 \pm 0.01273$ | $0.973417 \pm 0.01856$ |
| Gaussian + Selective GMM, Spectral Norm | $22.0729 \pm 1.25273$ | $95.0605 \pm 3.40124$ | $0.890506 \pm 0.0112498$ | $0.9738 \pm 0.00857087$ |
| Gaussian + Selective kNN | $21.0625 \pm 0.358012$ | $93.6284 \pm 2.29916$ | $0.891264 \pm 0.00734602$ | $0.947185 \pm 0.0178434$ |
| Gaussian + Selective kNN, $k = 5$ | $21.0625 \pm 0.358012$ | $93.6284 \pm 2.29916$ | $0.891103 \pm 0.00680399$ | $0.949392 \pm 0.0173229$ |
| Gaussian + Selective kNN, $k = 20$ | $21.0625 \pm 0.358012$ | $93.6284 \pm 2.29916$ | $0.891759 \pm 0.00804523$ | $0.94453 \pm 0.0188456$ |
| Gaussian + Selective kNN, L2 | $21.0625 \pm 0.358012$ | $93.6284 \pm 2.29916$ | $0.879639 \pm 0.0084247$ | $0.981862 \pm 0.0068585$ |

Table A40: Method variation results on the SKINLESIONPIXELS dataset.

| Method | val MAE (↓) | val Interval Length (↓) | test Coverage (≥ 0.90) | test Prediction Rate (↑) |
|---|---|---|---|---|
| Gaussian + Selective GMM | $105.417 \pm 1.15178$ | $535.139 \pm 215.446$ | $0.821515 \pm 0.0137705$ | $0.849579 \pm 0.0206181$ |
| Gaussian + Selective GMM, $k = 2$ | $105.417 \pm 1.15178$ | $535.139 \pm 215.446$ | $0.825054 \pm 0.0122698$ | $0.844799 \pm 0.0191092$ |
| Gaussian + Selective GMM, $k = 8$ | $105.417 \pm 1.15178$ | $535.139 \pm 215.446$ | $0.815696 \pm 0.010454$ | $0.862683 \pm 0.0259293$ |
| Gaussian + Selective GMM, Spectral Norm | $107.531 \pm 2.11824$ | $607.435 \pm 407.758$ | $0.821797 \pm 0.0350093$ | $0.837273 \pm 0.0205022$ |
| Gaussian + Selective kNN | $105.417 \pm 1.15178$ | $535.139 \pm 215.446$ | $0.813027 \pm 0.0306586$ | $0.927047 \pm 0.00830718$ |
| Gaussian + Selective kNN, $k = 5$ | $105.417 \pm 1.15178$ | $535.139 \pm 215.446$ | $0.812763 \pm 0.0309198$ | $0.928375 \pm 0.0100103$ |
| Gaussian + Selective kNN, $k = 20$ | $105.417 \pm 1.15178$ | $535.139 \pm 215.446$ | $0.812419 \pm 0.0306294$ | $0.929172 \pm 0.0109976$ |
| Gaussian + Selective kNN, L2 | $105.417 \pm 1.15178$ | $535.139 \pm 215.446$ | $0.808667 \pm 0.0112832$ | $0.831784 \pm 0.0181658$ |

Table A41: Method variation results on the HISTOLOGYNUCLEIPIXELS dataset.

| Method | val MAE (↓) | val Interval Length (↓) | test Coverage (≥ 0.90) | test Prediction Rate (↑) |
|---|---|---|---|---|
| Gaussian + Selective GMM | $211.795 \pm 10.0239$ | $1211.83 \pm 396.946$ | $0.59554 \pm 0.0956742$ | $0.90569 \pm 0.0453555$ |
| Gaussian + Selective GMM, $k = 2$ | $211.795 \pm 10.0239$ | $1211.83 \pm 396.946$ | $0.597998 \pm 0.094508$ | $0.912836 \pm 0.0323147$ |
| Gaussian + Selective GMM, $k = 8$ | $211.795 \pm 10.0239$ | $1211.83 \pm 396.946$ | $0.595718 \pm 0.0973458$ | $0.943449 \pm 0.0409511$ |
| Gaussian + Selective GMM, Spectral Norm | $206.492 \pm 7.82601$ | $896.779 \pm 61.4921$ | $0.65672 \pm 0.0824111$ | $0.855668 \pm 0.0632114$ |
| Gaussian + Selective kNN | $211.795 \pm 10.0239$ | $1211.83 \pm 396.946$ | $0.608929 \pm 0.0805954$ | $0.846228 \pm 0.0482832$ |
| Gaussian + Selective kNN, $k = 5$ | $211.795 \pm 10.0239$ | $1211.83 \pm 396.946$ | $0.610025 \pm 0.0809455$ | $0.842258 \pm 0.046231$ |
| Gaussian + Selective kNN, $k = 20$ | $211.795 \pm 10.0239$ | $1211.83 \pm 396.946$ | $0.607876 \pm 0.0817183$ | $0.858491 \pm 0.0496696$ |
| Gaussian + Selective kNN, L2 | $211.795 \pm 10.0239$ | $1211.83 \pm 396.946$ | $0.588514 \pm 0.0914286$ | $0.975562 \pm 0.0190356$ |

Table A42: Method variation results on the AERIALBUILDINGPIXELS dataset.

| Method | val MAE (↓) | val Interval Length (↓) | test Coverage (≥ 0.90) | test Prediction Rate (↑) |
|---|---|---|---|---|
| Gaussian + Selective GMM | $217.877 \pm 1.72493$ | $929.562 \pm 47.6606$ | $0.76535 \pm 0.0388677$ | $0.652082 \pm 0.0990489$ |
| Gaussian + Selective GMM, $k = 2$ | $217.877 \pm 1.72493$ | $929.562 \pm 47.6606$ | $0.751721 \pm 0.0454772$ | $0.634602 \pm 0.122618$ |
| Gaussian + Selective GMM, $k = 8$ | $217.877 \pm 1.72493$ | $929.562 \pm 47.6606$ | $0.850213 \pm 0.025791$ | $0.617738 \pm 0.0879951$ |
| Gaussian + Selective GMM, Spectral Norm | $220.763 \pm 7.23119$ | $1014.77 \pm 164.155$ | $0.817517 \pm 0.0505544$ | $0.638149 \pm 0.046255$ |
| Gaussian + Selective kNN | $217.877 \pm 1.72493$ | $929.562 \pm 47.6606$ | $0.651867 \pm 0.0771154$ | $0.840103 \pm 0.0463989$ |
| Gaussian + Selective kNN, $k = 5$ | $217.877 \pm 1.72493$ | $929.562 \pm 47.6606$ | $0.654256 \pm 0.0771566$ | $0.846067 \pm 0.0456245$ |
| Gaussian + Selective kNN, $k = 20$ | $217.877 \pm 1.72493$ | $929.562 \pm 47.6606$ | $0.647779 \pm 0.0795474$ | $0.832596 \pm 0.0542738$ |
| Gaussian + Selective kNN, L2 | $217.877 \pm 1.72493$ | $929.562 \pm 47.6606$ | $0.815075 \pm 0.0278439$ | $0.641851 \pm 0.11347$ |

