# OpenReview forum: "How Reliable is Your Regression Model's Uncertainty Under Real-World Distribution Shifts?"
_TMLR — Accepted by TMLR_

### Review · Reviewer_d2Ld · 2023-03-17

**Summary Of Contributions:**

This paper proposes several benchmark datasets to study a key research problem of quantifying the uncertainty of out-of-domain data in regression tasks. Some baseline methods are applied, but none of them consistently achieves good performance on both synthetic and real-world datasets.

**Audience:**

Yes

**Broader Impact Concerns:**

No ethical concerns.

**Claims And Evidence:**

No

**Requested Changes:**

I suggest the authors to better explain the evaluation metrics and conduct the experiment more systematically. Also, more insights of the distributional shift of each dataset can be interesting to readers.

**Strengths And Weaknesses:**

Strengths:

- Plenty of benchmark datasets can be used to facilitate the research on the uncertainty of OOD regression tasks.


Weakness:

- The metrics are confusing. Some plots compare test coverage and interval length, while some others compare prediction rates. It is unclear the motivation for using different metrics.

- The comparison of test coverage under different prediction rates is tricky. It is claimed in the paper that Gaussian + Selective GMM is well-calibrated in all synthetic datasets. However, in fact, the prediction rate is quite low. It doesn't mean that it can perform well even on synthetic datasets.

- There are few insights or analyses on why a certain type of distributional shift can be solved by some assumptions. More analyses on this perspective can make the benchmark datasets more representative.

---

> ### Author Response · Authors · 2023-05-06
> **Response to Reviewer d2Ld (part 1/3)**
>
> Thank you for reviewing our paper and providing this feedback.
>
> We address your requested changes and raised concerns (weaknesses) below. We will update the paper in the coming days, and will notify you once this is completed. Please let us know if there are any remaining concerns which are not addressed by this response or by our planned paper revision.
>
>
> ***
>
>
> **_"The metrics are confusing. Some plots compare test coverage and interval length, while some others compare prediction rates. It is unclear the motivation for using different metrics"_:** \
> We compare two sets of uncertainty estimation methods. For the first set of methods (Conformal Prediction, Ensemble, Gaussian, Gaussian Ensemble, Quantile Regression), the model outputs a prediction interval $C_{\alpha}(x)$ and a predicted target $\hat{y}(x) \in C_{\alpha}(x)$ for each input $x$. The average interval length is then a natural secondary metric, measuring the quality/sharpness of the prediction intervals. If two different methods both are perfectly calibrated, i.e. $\mathrm{Coverage}(C_\alpha) = 1 - \alpha$, the method producing smaller prediction intervals is preferred. Therefore, Figure 2 and Figure 3 compare this first set of methods in terms of both test coverage and interval length.
>
> For the second set of methods (Gaussian + Selective GMM, Gaussian + Selective kNN, Gaussian + Selective Variance, Gaussian Ensemble + Selective GMM, Gaussian Ensemble + Selective Ensemble Variance), a selective prediction mechanism (with uncertainty function $\kappa_f: \mathcal{X} \rightarrow \mathbb{R}$) is added on top of either the Gaussian or Gaussian Ensemble method. The prediction and corresponding prediction interval of this underlying method is thus output only for certain inputs $x$, i.e. if and only if $\kappa_f(x) \leq \tau$. The proportion of inputs for which a prediction is output -- the prediction rate -- is then another natural secondary metric. A complete evaluation for this second set of methods should thus include both interval length and prediction rate, together with the main metric test coverage. The added selective prediction mechanism does however not modify the intervals of the underlying Gaussian or Gaussian Ensemble method (which already have been evaluated in terms of interval length in Figure 2 and Figure 3), it just gives the option to abstain from outputting these predictions for certain inputs. In Figure 4, Figure 5, Figure A1 and Figure A2, we therefore compare this second set of methods only in terms of test coverage and prediction rate.
>
> We will clarify this in the revised paper. We will also add more detailed descriptions to the captions of the results figures (Figure 2  - Figure 5, Figure A1 and Figure A2), clarifying which methods are being compared and in terms of which metrics.

---

> > ### Author Response · Authors · 2023-05-06
> > **Response to Reviewer d2Ld (part 2/3)**
> >
> > **_"The comparison of test coverage under different prediction rates is tricky. It is claimed in the paper that Gaussian + Selective GMM is well-calibrated in all synthetic datasets. However, in fact, the prediction rate is quite low. It doesn't mean that it can perform well even on synthetic datasets"_:** \
> > For methods based on selective prediction, prediction intervals $C_{\alpha}(x^\star)$ are output only for certain test inputs $x^\star$ (iff $\kappa_f(x^\star) \leq \tau$), and the empirical interval coverage is computed only on this subset of test. The prediction intervals of such a method are thus perfectly calibrated if, only considering this subset of test, the proportion of inputs for which the prediction interval covers the corresponding target equals $1 - \alpha$. I.e., the model is given the option to abstain from outputting predictions for some inputs, but the prediction intervals that actually are output by the model should be calibrated. It is in this sense that the Gaussian + Selective GMM method is almost perfectly calibrated across all synthetic datasets (as stated in the second paragraph of Section 6.2). This is similar to the notion of "selective calibration" discussed for the classification setting in [1].
> >
> > While calibration (test coverage) is our main metric, we agree that this by itself not is sufficient for a method to be said to "perform well" in a general sense. As briefly discussed in Section 3.2 (Secondary Metrics paragraph), a perfectly calibrated model with a low prediction rate might not be particularly useful in practice. While even very low prediction rates likely would be tolerated in many safety-critical medical applications (as long as the method stays perfectly calibrated), one can imagine more low-risk settings where this calibration versus prediction rate trade-off is less clear. This is exactly the reason why we present all test coverage results together with the corresponding prediction rates in Figure 4, Figure 5, Figure A1 and Figure A2. Since not a single one of the evaluated methods was close to being perfectly calibrated across all datasets, we did however mainly focus on analyzing the test coverage. If multiple methods had performed well in terms of calibration, a more detailed analysis and discussion of the secondary metrics performance would have been necessary.
> >
> > We will clarify and discuss these points in more detail in the revised version.
> >
> > [1] Adam Fisch, Tommi S. Jaakkola, and Regina Barzilay. Calibrated Selective Classification. TMLR, 2022.
> >
> >
> > ***
> >
> >
> > **_"Also, more insights of the distributional shift of each dataset can be interesting to readers"_:** \
> > We agree that it would be interesting to relate the test coverage performance to a quantitative measure of distribution shift ("distance" between the train/val and test distributions), complementing our qualitative discussion in the final paragraph of Section 6.3. How to quantify the level of distribution shift in real-world datasets is however far from obvious, see e.g. Appendix E.1 in [2]. We will explore if the difference in regression accuracy (MAE) on val and test can be adopted as such a measure, extending the approach from [2] to our regression setting.
> >
> > Moreover, we will add at least one experiment to study varying levels of distribution shifts. We will explore creating test sets with increasing degrees of distribution shifts for the synthetic Cells-Tails and ChairAngle-Tails datasets (e.g. for Cellls-Tails, by increasing the target range from $]50, 150]$ to $[1, 200]$ in steps). We will also explore creating a version of the AerialBuildingPixels dataset with a more limited shift between train/val and test, by evaluating on the images from Kitsap County, WA or Vienna, Austria (instead of West Tyrol, Austria).
> >
> > [2] Florian Wenzel et al. Assaying Out-Of-Distribution Generalization in Transfer Learning. NeurIPS, 2022.

---

> > > ### Author Response · Authors · 2023-05-06
> > > **Response to Reviewer d2Ld (part 3/3)**
> > >
> > > **_"There are few insights or analyses on why a certain type of distributional shift can be solved by some assumptions. More analyses on this perspective can make the benchmark datasets more representative"_:** \
> > > While we agree that e.g. explaining why such an obvious performance difference was observed between synthetic and real-world datasets (as discussed in Section 6.3) is an important problem, we argue that it is beyond the scope of the current paper. Instead, we consider this a very interesting direction for future work.
> > >
> > > Our main contribution is constructing a benchmark of large-scale yet convenient datasets, containing different types of challenging distribution shifts, testing the reliability of uncertainty estimation methods -- specifically for the image-based regression setting. It consists solely of publicly available datasets, and our complete implementation will be made available. We also employ our benchmark to uncover important limitations of popular uncertainty estimation methods (the failure mode of the conformal prediction methods, despite their commonly promoted theoretical guarantees, is an especially important finding), but addressing these limitations is beyond the scope of our paper. Our goal is for the benchmark to serve as a challenge to the community, starting a line of research that moves towards a solution to the ultimate problem of how to develop truly reliable methods. We will clarify this in the revised version.

---

### Review · Reviewer_c8Ct · 2023-03-19

**Summary Of Contributions:**

This work conducts a benchmark study for (deep) regression models under realistic distributional shifts. Authors defines calibration as the coverage probability of 90% confidence intervals, and examined the calibration and selective performance of 10 variants of canonical uncertainty techniques and their combinations (conformal prediction, quantile regression, frequentest Gaussian regression, approximate Bayesian regression). Author considered 8 vision benchmarks covering medical imaging, CAD design, and remote sensing, each equipped with corresponding OOD splits. Author find that the methods such as quantile regression and conformal prediction under-perform despite their i.i.d. theoretical guarantees, and the performance of all methods (even those designed for OOD detection) deteriorate under real-world distributional shifts.

**Audience:**

Yes

**Claims And Evidence:**

No

**Requested Changes:**

* **Investigate the relationship between accuracy, calibration and sharpness (interval length).** In this work, authors investigated 3 aspects of method performance: accuracy (measured by MAE), calibration (measured by CI coverage) and sharpness (measured by CI length). However, as we know both intuitively and revealed in classification literature, these 3 aspects are not independent. For example, worse prediction accuracy essentially leads to a more difficulty trade-off between calibration and sharpness, in that the CI length needs to be larger to attain the correct confidence interval. Consequently, to get a deeper understanding of the model performance under distributional shift, it might be interesting to conduct some additional accuracy v.s. uncertainty studies to plot the pairwise association between the three metrics.

* **Study different calibration performance at different $\alpha$ levels**. In this work, the authors only investigated the coverage probability of 90% confidence intervals, which may not fully capture the method performance at different confidence levels. I would encourage authors to consider adding some additional results for other commonly-considered confidence level (e.g., 95%), or specify a calibration metric that measure the difference between nominal v.s. empirical coverage across a range of confidence levels. Even if all methods do not perform well at some levels (e.g., 95%), it is still a valuable message that is suitable for TMLR and worthwhile to inform the community.

* **Study performance deterioration at different degrees of distributional shift**. To fully illustrate the method behavior under distributional shift, it is a standard practice of this literature (e.g., [Ovadia et al (2019)](https://proceedings.neurips.cc/paper/2019/file/8558cb408c1d76621371888657d2eb1d-Paper.pdf) and [Krishnan and Tickoo (2020)](https://proceedings.neurips.cc/paper/2020/hash/d3d9446802a44259755d38e6d163e820-Abstract.html)) to study method performance under controlled test sets of different degrees of distributional shift. Can authors consider adding at least one additional study to illustrate this? It should be easily doable for the synthetic Cells and Chair data. For real-world shifts, you can consider e.g., those remote sensing datasets by creating test sets with different degrees of geographical affinity with respect to training data.

Minor:

* Section 3.2. "Secondary Metrics".  If my understanding is correct, the secondary metrics authors proposed (interval length and prediction rate) are measuring the "sharpness" of the probabilistic predictions. Consequently, the evaluation framework authors proposed echoes the conceptual framework is put forward by Gneiting, Balabdaoui, and Raftery (2007, which authors already cited) "optimizing sharpness while controlling for calibration". It might be worthwhile to point out this connection by adding a finishing comment at the end of Section 3.2.

**Strengths And Weaknesses:**

Strength:

 * Novelty & Significance: This work conducted a comprehensive study for well-known regression techniques on relevant ML settings (deep regression for vision recognition under real-world distributional shift). The performance axes measured (CI coverage, CI interval width, selective prediction rate) are relatively comprehensive. The conclusion draw from the empirical studies (esp the failure mode of popular methods such as conformal / quantile regression) is worthwhile to put out there for the community.
  * Clarity: The paper is clearly written and with experiment results neatly summarized in few key figures.

Weakness:

  * Experiment Evaluation: Currently, author measured the three performance axes (CI coverage, CI interval width, selective prediction rate) independently without investigating their tradeoff relationships. Furthermore, the distributional shift evaluation is done on test splits with fixed degree of shift. Therefore it is difficult to understand how method performance deteriorates as the degree of distributional shift increases. The experiment evaluation can be made much stronger by considering these aspects (please see "Requested Chanegs").

---

> ### Author Response · Authors · 2023-05-06
> **Response to Reviewer c8Ct**
>
> Thank you for the comprehensive and very insightful review. This is valuable feedback which we think will help further strengthen the paper.
>
> We address your four requested changes in turn below. We will update the paper in the coming days, and will notify you once this is completed. Please let us know if there are any remaining concerns which are not addressed by this response or by our planned paper revision.
>
> ***
>
> **Investigate the relationship between accuracy, calibration and sharpness (interval length):** \
> We agree that this would help gain a deeper understanding of the method performance, thank you for the suggestion. We will compare the performance of the five common uncertainty estimation methods (Conformal Prediction, Ensemble, Gaussian, Gaussian Ensemble, Quantile Regression) by plotting test coverage error (difference between empirical and expected interval coverage) vs. val MAE, test coverage error vs. val interval length, and val MAE vs. val interval length.
>
> ***
>
> **Study different calibration performance at different $\alpha$ levels:** \
> Please note that we already report results also for $\alpha = 0.2$ and $\alpha = 0.05$. While the main paper only contains results for 90\% prediction intervals ($\alpha = 0.1$), we repeat most of the evaluation for two alternative miscoverage rates $\alpha$ in Appendix B. Specifically, we redo the evaluation of 6/10 methods on 9/12 datasets with 80\% ($\alpha = 0.2$) and 95\% ($\alpha = 0.05$) prediction intervals. The results for 80\% prediction intervals are given in Table A13 - Table A21, and the results for 95\% are given in Table A22 - Table A30. This is just briefly mentioned in the beginning of Section 6, and we agree that it should be described in more detail also in the main paper. To better highlight the results of this additional investigation, we will also present them in figures of the same type as Figure 2 - Figure 5 (instead of just presenting them in Table A13 - Table A30).
>
> ***
>
> **Study performance deterioration at different degrees of distributional shift:** \
> Thank you for the suggestion, we agree that this would be interesting. We will add at least one such study. We will explore creating test sets with increasing degrees of distribution shifts for the Cells-Tails and ChairAngle-Tails datasets (e.g. for Cells-Tails, by increasing the target range from $]50, 150]$ to $[1, 200]$ in steps). We will also explore creating a version of the AerialBuildingPixels dataset with a more limited shift between train/val and test, by evaluating on the images from Kitsap County, WA or Vienna, Austria (instead of West Tyrol, Austria).
>
> ***
>
> **Minor (Section 3.2, Secondary Metrics):** \
> We will add a reference to (Gneiting et al., 2007) also in this paragraph.

---

### Review · Reviewer_ztGJ · 2023-04-25

**Summary Of Contributions:**

This work benchmarks a variety of methods that compute prediction intervals on a collection of synthetic and real-world regression tasks. The primary evaluation criteria is "Coverage" which is what proportion of predictions of predictions are within the prediction interval on a held-out dataset.

Statements/Conclusions drawn by this work:
> "while all methods are well calibrated when there is no distribution shift, they become highly overconfident on many of the benchmark datasets"

> "not a single method actually reaches the desired 90% test coverage on any of these real-world datasets"


**Audience:**

Yes

**Claims And Evidence:**

No

**Requested Changes:**

Can you contrast the achieved performance of the models trained for this work with published SOTA numbers. Perhaps as another row in the Appendix tables that show the MAE scores. Or a scatter plot in the main text?

Can you add more actionable conclusions? The paper states that these methods don't perform well under distributional shift, have the experiments yielded any more conclusions that you can enumerate in the conclusion section?


**Strengths And Weaknesses:**

Strengths:
The datasets and methods evaluated are well organized and diverse. The text is well written and is organized well overall.


Weaknesses:
It is unclear how the model MAE performance compares to the SOTA models for these datasets. Perhaps the performance conclusions on real world datasets is because these models are not representative of models trained by experts in those domains. However this does not change the fact that the uncertainty should have been predicted higher given any performance of the model but it would be nice to present it here anyway.

There are not many specific claims made by the paper. I find it hard to enumerate the takeaways. There is a lot of discussion but the statements there are very light and hard to conclude from. For example "feature-space methods perform well relative to common baselines, the resulting selective prediction methods are still overconfident in many cases" maybe adding quantitative differences here can make it more actionable. Also, hypothesizing why this is the case given your experience working with these models.

The paper presents one main experiment that is repeated across different datasets. This is limiting because we just see one perspective of a potentially very dynamic space of models/methods. It might provide more concrete evidence to see the distributional shift vary based on some parameter and then look at the response of the uncertainty methods.

---

> ### Author Response · Authors · 2023-05-06
> **Response to Reviewer ztGJ (part 1/2)**
>
> Thank you for providing this valuable feedback.
>
> We address your requested changes and raised concerns (weaknesses) below. We will update the paper in the coming days, and will notify you once this is completed. Please let us know if there are any remaining concerns which are not addressed by this response or by our planned paper revision.
>
>
> ***
>
>
> **_"Can you contrast the achieved performance of the models trained for this work with published SOTA numbers. Perhaps as another row in the Appendix tables that show the MAE scores. Or a scatter plot in the main text?"_:** \
> While the Cells and ChairAngle datasets are taken directly from Ding et al. (2021; 2020), the authors propose and use these datasets for generative image modelling purposes, not for training regression models. All other datasets utilized in our benchmark are custom datasets, which we construct by modifying publicly available datasets from the literature. For the AssetWealth dataset, the original images are resized from size $224 \times 224$ to $64 \times 64$. For VentricularVolume, a video dataset is converted into an image-based regression dataset. The four remaining datasets (BrainTumourPixels, SkinLesionPixels, HistologyNucleiPixels and AerialBuildingPixels) are all converted image segmentation datasets. Constructing these custom datasets is one of our main contributions. This is what enables us to propose an extensive benchmark of large-scale yet convenient datasets, containing different types of challenging distribution shifts, specifically for image-based regression. Because we propose a new benchmark consisting of custom datasets, we are however not able to compare the MAE of our models with any published results. We will clarify this in the revised paper.
>
> We choose to train all models based on a ResNet34 backbone network because it is widely used across various applications, yet simple to implement and quite computationally inexpensive. The proposed benchmark and the evaluated uncertainty estimation methods are however entirely independent of this specific choice of network architecture. Exploring the use of other more powerful models and evaluating how this affects the reliability of uncertainty estimation methods is an interesting direction. Due to time constraints, we will however leave this for future work. By only using publicly available datasets, and making available our complete implementation, we also hope that the proposed benchmark will be utilized by many other researchers in the future. Our goal is for the benchmark to serve as a challenge to the community, starting a line of research on various aspects of developing truly reliable regression models.
>
>
> ***
>
>
> **_"The paper presents one main experiment that is repeated across different datasets. This is limiting because we just see one perspective of a potentially very dynamic space of models/methods. It might provide more concrete evidence to see the distributional shift vary based on some parameter and then look at the response of the uncertainty methods"_:** \
> Please note that we evaluate 10 methods across multiple diverse datasets, both synthetic and real-world, featuring different types of distribution shifts and of different levels. Thus, our evaluation is not limited to just one perspective. We do however agree that it would be interesting to conduct controlled experiments to further study varying levels of distribution shifts and how this affects the performance. We will add at least one such experiment. We will explore creating test sets with increasing degrees of distribution shifts for the Cells-Tails and ChairAngle-Tails datasets (e.g. for Cellls-Tails, by increasing the target range from $]50, 150]$ to $[1, 200]$ in steps). We will also explore creating a version of the AerialBuildingPixels dataset with a more limited shift between train/val and test, by evaluating on the images from Kitsap County, WA or Vienna, Austria (instead of West Tyrol, Austria).

---

> > ### Author Response · Authors · 2023-05-06
> > **Response to Reviewer ztGJ (part 2/2)**
> >
> > **_"Can you add more actionable conclusions? The paper states that these methods don't perform well under distributional shift, have the experiments yielded any more conclusions that you can enumerate in the conclusion section?"_:** \
> > We agree that adding more actionable insights and takeaways to Section 7 would help further strengthen the paper, hopefully spurring more follow-up work by other researchers. We will analyze the results again and update the corresponding discussion in Section 6.3, and then add a subsection that clearly states our main actionable takeaways.
> >
> > Preliminary list of actionable takeaways:
> >
> > - All methods are well calibrated on baseline datasets with no distribution shift, but become highly overconfident in many realistic scenarios. Uncertainty estimation methods must therefore be evaluated using sufficiently challenging benchmarks. Otherwise, one might be lead to believe that methods will be more reliable during real-world deployment than they actually are.
> >
> > - Conformal prediction methods have commonly promoted theoretical coverage guarantees, but these depend on an assumption (exchangeable data points) that is unlikely to hold in many practical applications. Consequently, also these methods can become highly overconfident in realistic scenarios. If the underlying assumptions are not examined critically by practitioners, such theoretical guarantees risk instilling a false sense of security -- making these models even less suitable for safety-critical deployment.
> >
> > - The clear performance difference between synthetic and real-world datasets observed for selective prediction methods based on feature-space density (Gaussian + Selective GMM, Gaussian + Selective kNN, Gaussian Ensemble + Selective GMM) is a very interesting direction for future work. If the reasons for this performance gap can be understood, a method that stays well calibrated across all datasets could potentially be developed.
> >
> > - Selective prediction methods based on feature-space density perform well relative to other methods (as expected based on recently demonstrated state-of-the-art OOD detection performance), but are also overconfident in many cases. Only comparing the relative performance of different methods is therefore not sufficient. To track if actual progress is being made towards the ultimate goal of truly reliable uncertainty estimation methods, benchmarks must also evaluate method performance in an absolute sense.

---

### Author Response · Authors · 2023-05-08
**Revised Paper**

We have now uploaded a revised version of the paper.

Changes:
- We moved Section 6.3 Discussion to a separate section (Section 7 Discussion), and then also added a list of four main actionable takeaways at the end of this section.
- We added Figure A3 - Figure A10 at the end of the appendix.

We will provide more details on the revision within 14 hours.

---

> ### Author Response · Authors · 2023-05-09
> **Revised Paper - Further Details**
>
> Due to time-constraints we did not manage to include everything from our reviewer responses in the revised paper, but it does include all promised new experiments and results (see Figure A3 - Figure A10). We prioritized completing these over some of the clarifications and other textual updates.
>
> If the paper is accepted, we will incorporate the remaining changes in the camera-ready version. In a camera-ready version, we would also describe and analyze the new experiments/results in more detail, and move some of Figure A3 - Figure A10 to the main paper.
>
> We thank the reviewers again for their comments and very valuable feedback. Please let us know if there are any remaining concerns which are not addressed in the revised paper or in our previous responses, we would be happy to provide further details or clarifications.

---

### Decision · Action_Editors · 2023-06-23

**Recommendation:** Accept with minor revision

**Comment:**

This paper presents a benchmark of vision-based regression problems, with a focus on evaluating uncertainty estimation under data distribution shift.  There has been extensive work on this in the classification setting, but not as much in the regression setting, so it explores a useful void in the literature.  The initial reviews were leaning towards reject, with reviewers appreciating the contribution of the benchmark but were left wanting more in terms of analysis, discussion of related work and overall claims / contribution.  In their response, the authors have added more experiments and presented some useful and interesting take-away analysis of their results. Two of three reviewers recommended leaning reject and one leaning accept.  Also two of three found the claims were not supported by evidence.  However, one reviewer felt that the paper including the benchmarks and analysis were a useful contribution to the community.  In my own reading of the work, and in discussion with the reviewer, I tend to agree with that reviewer.   The additions to the paper after the author response period seem to help strengthen the message and the author response seems to help.  I found the take-away messages interesting and useful. Although maybe it's expected, I found it interesting to observe that conformal prediction performed very well in some scenarios but also so poorly under data distribution shift.

Overall, there doesn't seem to be anything specifically problematic or wrong about the paper and it presents an interesting and useful contribution to the community.  Therefore I would recommend an accept but request that the authors take some time to carefully prepare the camera ready, addressing the reviewer comments.  A common criticism from the reviewers was that some of the details that should be in the main text were lost somewhat in the appendix.  Also the authors have promised to add additional experiments including at least one where the amount of distribution shift is carefully controlled.  It would be useful if the authors could integrate these into the main text for the camera ready.

**Audience:**

Yes, uncertainty in deep learning is a popular subject and becoming more and more important as models are deployed in real-world scenarios.  This helps provide some insights into regression scenarios where one can and can't trust the quantification of uncertainty of a particular appoach.

**Claims And Evidence:**

Two of the three reviewers found that the claims were not supported by evidence.  However, it seemed like a major criticism from the reviewers was that there was a lack of claims to accompany the evidence.  The paper is somewhat of a pile of experiments with perhaps a lack of strong narrative around what the experiments show.  The revision after the author response seems to have done a lot to address this point and presented some interesting analysis.  I don't personally see strong claims in the paper that are unsubstantiated, so I tend to agree with the reviewer that found the claims and evidence ok.

---

> ### Author Response · Authors · 2023-07-11
> **Thank you**
>
> Thank you for the thorough evaluation and detailed comments, it is highly appreciated.
>
> We will add a camera-ready version in the coming weeks, and will make sure to include the remaining promised changes.